# Deep learning subgrid-scale parametrisations for short-term forecasting of sea-ice dynamics with a Maxwell-elasto-brittle rheology

Tobias Sebastian Finn[1], Charlotte Durand[1], Alban Farchi[1], Marc Bocquet[1], Yumeng Chen[2], Alberto Carrassi[2,3], and Véronique Dansereau[4]

[1]CEREA, École des Ponts and EDF R&D, Île-de-France, France
[2]Dept. of Meteorology and NCEO, University of Reading, Reading, United Kingdom
[3]Dept of Physics and Astronomy "Augusto Righi", University of Bologna, Italy
[4]Université Grenoble Alpes, CNRS, Grenoble INP, Laboratoire 3SR, Grenoble, France

**Correspondence:** Tobias Sebastian Finn (tobias.finn@enpc.fr)

**Abstract.** We introduce a proof-of-concept to parameterise the unresolved subgrid-scale of sea-ice dynamics with deep learning techniques. Instead of parameterising single processes, a single neural network is trained to correct all model variables at the same time. This data-driven approach is applied to a regional sea-ice model that accounts exclusively for dynamical processes with a Maxwell-elasto-brittle rheology. Driven by an external wind forcing in a $40 \, \text{km} \times 200 \, \text{km}$ domain, the model generates examples of sharp transitions between unfractured and fully-fractured sea ice. To correct such examples, we propose a convolutional U-Net architecture which extracts features at multiple scales. We test this approach in twin experiments: the neural network learns to correct forecasts from low-resolution simulations towards high-resolution simulations for a lead time of about 10 minutes. At this lead time, our approach reduces the forecast errors by more than $75 \, \%$, averaged over all model variables. As most important predictors, we identify the dynamics of the model variables. Furthermore, the neural network extracts localised and directional dependent features, which points towards the shortcomings of the low-resolution simulations. Applied to correct the forecasts every 10 minutes, the neural network is run together with the sea-ice model. This improves the short-term forecasts up to an hour. These results consequently show that neural networks can correct model errors from the subgrid-scale for sea-ice dynamics. We therefore see this study as an important first step towards hybrid modelling to forecast sea-ice dynamics on an hourly to daily timescale.

## 1   Introduction

Sea-ice models with elasto-brittle rheologies (e.g., Rampal et al., 2016) simulate the dynamics of sea ice with an unprecedented accuracy for Arctic-wide simulations in the mesoscale with horizontal resolutions of around $10 \, \text{km}$ (Rabatel et al., 2018; Bouchat et al., 2022; Boutin et al., 2022). These models reproduce the observed temporal and spatial scale-invariance of the sea-ice deformation and drift across multiple scales, up to the resolution of a single grid cell (Dansereau et al., 2016; Rampal et al., 2019; Ólason et al., 2021). Elasto-brittle rheologies parametrise the unresolved subgrid-scale processes associated with brittle fracturing through a progressive damage framework (Tang, 1997; Amitrano et al., 1999; Girard et al., 2011). Such framework

connects the elastic modulus of the material at the grid cell level to the degree of fracturing at the subgrid-scale. Comprised between 0, undamaged, and 1, completely damaged material, the fracturing is represented by the level of *damage*. When the internal stress exceeds a given damage criterion locally, the level of damage increases and the elastic modulus decreases, thereby reducing the local effective stress. Excessive stress is elastically redistributed throughout the material, causing overcritical stress elsewhere. Hence, the damage is highly localised and progressively propagated through the material, which also leads to a strong localisation of the deformation. The Maxwell-elasto-brittle rheology (Dansereau et al., 2016) adds to this framework the concept of an "apparent" viscosity. Coupled to the level of damage, the added viscosity allows accounting for the relaxation of stresses by permanent deformations within a fractured sea-ice cover. Although models with such rheologies successfully reproduce the observed scaling properties of sea-ice deformation, they locally underestimate very high convergence and shear rates in some instances (Ólason et al., 2022). Thus, some important, possibly subgrid-scale, processes are still unresolved at resolutions of around 10 km or unrepresented in elasto-brittle rheologies and their damage parametrisations.

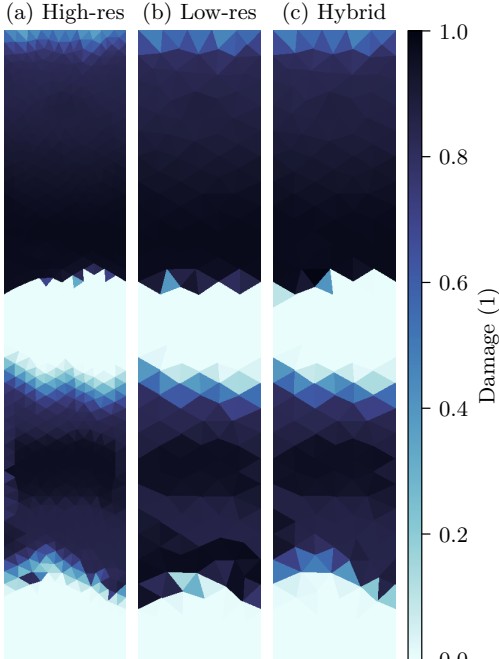

**Figure 1.** Snapshot of sea-ice damage for a one-hour forecast with the here-used regional sea-ice model. Shown are the high-resolution simulations (a, 4 km resolution) and low-resolution forecasts (b, c). To initialise the low-resolution forecasts, the initial conditions of the high-resolution are projected into a low-resolution space with 8 km resolution. Started from these projected initial conditions, the low-resolution forecast (b) generates too much damage compared to the high-resolution field. Running the low-resolution model together with our learned model error correction (c) leads to a better representation of the damaging process, which improves the forecast by 62 % in this example.

To exemplify the impact of these unresolved subgrid-scale processes on the sea-ice dynamics, and to see how deep learning can remedy these issues, we perform twin experiments with a regional sea-ice model that depicts exclusively the dynamics in a Maxwell-elasto-brittle rheology (Dansereau et al., 2016, 2017, 2021). In a $40\,\mathrm{km} \times 200\,\mathrm{km}$ ($x \times y$-direction) domain, we impose an external wind forcing with a sinusoidal velocity in $y$-direction. This forcing generates sharp transitions from unfractured to almost completely fractured sea ice. Such an instance of sharp transitions is exemplary shown in Fig. 1a for a simulation with a $4\,\mathrm{km}$ horizontal resolution and a lead time of one hour. Initialised with the same but projected initial conditions, a simulation at a $8\,\mathrm{km}$ horizontal resolution leads to a different trajectory, Fig. 1b. Such different instances of sea-ice dynamics are caused by differently integrated processes. Consequently, the sea-ice damage can already significantly differ after one hour of simulation. Here, in the transition zones, the low-resolution simulation fractures the sea ice too strongly compared to the high-resolution. In this study, we introduce a baseline deep learning approach to correct the missing processes. By parametrising the subgrid-scale, the hybrid model can better reproduce the temporal evolution of high-resolution simulations at the lower resolution, Fig. 1c.

Subgrid-scale parametrisations with machine learning have already been proved useful for other Earth system components (Brenowitz and Bretherton, 2018; Beucler et al., 2021; Irrgang et al., 2021). In the atmosphere, cloud processes can be learned from emulating super-parametrised or super-resolved models within a lower-resolution model (Gentine et al., 2018; Rasp et al., 2018; Seifert and Rasp, 2020). Additionally, machine learning can parametrise turbulent dynamics in the atmosphere (Beck and Kurz, 2021; Cheng et al., 2022) and in the ocean (Zanna and Bolton, 2020; Guillaumin and Zanna, 2021).

To predict the sea-ice concentration, purely data-driven surrogate models can replace geophysical models at daily (Liu et al., 2021) and seasonal forecast horizons (Andersson et al., 2021). Furthermore, small neural networks can emulate granular simulations of ocean-sea-ice interactions, allowing to parametrise the effect of ocean waves onto the sea ice (Horvat and Roach, 2022). In this study, we take another point of view and show more generally that subgrid-scale processes for sea-ice dynamics can be parametrised with deep learning, correcting all prognostic model variables at the same time.

The dynamics of sea-ice impose hereby new challenges for neural networks (NNs) that should parametrise the subgrid-scale:

- Current sea-ice models represent leads in a band of few pixels, and sharp transition zones can appear as a non-continuous step function within the data. For such discrete-continuous mixture data distributions, NNs that simply learn to regress into the future tend to diffuse and blur the target (Ayzel et al., 2020; Ravuri et al., 2021), if trained by a pixel-wise loss function. A correct representation of sharp transitions can thus induce problems within the training of the NN, resulting into a diffusion of the normally concentrated transition zones.

- In elasto-brittle models, the handling of the internal stress depends on the fragmentation of sea ice. This dependency also leads to different forecast error distributions for different fragmentation levels, even for variables only indirectly related to the stress, like the sea-ice thickness. Consequently, for model error correction, a NN has to be trained across a range of fragmentation levels and should be able to output multimodal predictions in the best case.

- As sea ice is scale-invariant up to the kilometre-scale, fragmentation of sea-ice propagates from small, unresolved, scales to the larger, resolved, scales. Because the small scales are unresolved, the appearance of linear kinematic features seems

to be stochastic from the resolved macro-scale point of view. Furthermore, such features are inherently multifractal and propagate in an anisotropic medium (Wilchinsky and Feltham, 2006, 2011).

Finally, the found subgrid-scale parametrisation approach should be scalable to a range of resolutions, from regional models used in this study to Arctic-wide models, like neXtSIM (Rampal et al., 2016; Ólason et al., 2022).

As a first step towards solving these challenges for NNs and giving a proof-of-concept, we present the aforementioned twin experiments with a regional model. Our goal is to train NNs to correct the output of simulations with a $8\,\mathrm{km}$ horizontal resolution towards simulation with a $4\,\mathrm{km}$ resolution. As the low-resolution model setup resolves fewer processes than the high-resolution setup, the NN has to account for the unresolved subgrid-scale processes to correct model errors. For this goal, we have found a baseline deep learning architecture, based on the U-Net approach (Ronneberger et al., 2015) and with applied tricks, e.g., from the ConvNeXt architecture (Liu et al., 2022). The NNs are trained to correct all nine prognostic model variables for a lead time of $10\,\mathrm{min}$ and $8\,\mathrm{s}$ (a multiplier of our $16\,\mathrm{s}$ model time step). During forecasting, the so-trained NN can be applied every $10\,\mathrm{min}$ and $8\,\mathrm{s}$ to continuously correct the model output. Based on this approach, we present first promising results for short-term forecasting (up to $60\,\mathrm{min}$), as showcased in Fig. 1c.

We introduce the problem that we try to solve, the regional sea-ice model, and our strategy to train the NNs in Sect. 2. The NN for the model error correction is briefly explained in Sect. 3. Results are given in Sect. 5, summary and discussion in Sect. 6, and final, concise, conclusions in Sect. 7. A more rigorous introduction of the model can be found in Appendix A and a more technical description of the NN in Appendix B.

## 2 Twin experiments for deep learning a model error correction

Our goal is to make a proof-of-concept that subgrid-scale processes can be parameterised by neural networks (NNs). We hereby parametrise subgrid-scale processes with a NN that corrects model errors. As testbed, we use a regional sea-ice model that depicts sea-ice dynamics in a Maxwell-elasto-brittle rheology. To train the neural networks, we use twin experiments, where we compare low-resolution forecast to a known high-resolved truth, simulated with the same sea-ice model.

### 2.1 Problem formulation

Our goal is to parametrise unresolved processes of the forecast model $\mathcal{M}(\cdot)$ that maps an initial state $\boldsymbol{x}_{t-1}^{\mathrm{in}}$ at time $t-1$ to a forecast $\boldsymbol{x}_t^{\mathrm{f}}$ at time $t$,

$$\boldsymbol{x}_t^{\mathrm{f}} = \mathcal{M}(\boldsymbol{x}_{t-1}^{\mathrm{in}}), \tag{1}$$

to simplify the notation, time has been discretised, $t \in \mathbb{N}$. Normally, parametrisations for single processes are integrated together with the forecast model. Instead, we learn a model error correction that has to parametrise subgrid-scale processes and correct all prognostic model variables at the same time.

The correction is represented by the output of a NN, $f(\boldsymbol{x}_{t-1}^{\text{in}}, \boldsymbol{x}_t^{\text{f}}, \boldsymbol{\phi})$, which makes use of the initial state and the forecast as input and combines them with its parameters $\boldsymbol{\phi}$. The NN is trained to predict the residual $\Delta\boldsymbol{x}_t = \boldsymbol{x}_t^t - \boldsymbol{x}_t^{\text{f}}$ between the truth $\boldsymbol{x}_t^t$ and the forecasted state.

To apply the model error correction for continuous forecasting, the predicted residual is added to the forecast, resulting into the corrected forecast $\boldsymbol{x}_t^{\text{c}}$. This corrected forecast can be then used as subsequent initial state for the forecast model,

$$100 \quad \boldsymbol{x}_t^{\text{c}} = \boldsymbol{x}_t^{\text{f}} + f(\boldsymbol{x}_{t-1}^{\text{in}}, \boldsymbol{x}_t^{\text{f}}, \boldsymbol{\phi}), \qquad\qquad\qquad \boldsymbol{x}_t^{\text{in}} = \boldsymbol{x}_t^{\text{c}}. \qquad\qquad (2)$$

Applied to correct the model variables in this way, the neural network can be used together with the sea-ice model.

## 2.2   Testbed with a regional sea-ice model

The model depicts the dynamical processes of sea ice with a Maxwell-elasto-brittle rheology (Dansereau et al., 2016). The thermodynamics consist of only redistribution of sea-ice *thickness*, handled as tracer variable similarly to the sea-ice *area*. The
elasto-brittle rheology introduces a *damage* variable that parameterises subgrid-scale processes and represents the fragmentation level of the sea ice on a grid-box level. Depending on the state of the sea ice and especially the *cohesion*, the sea-ice deformation, represented as *stress*, is converted into permanent damage. In this model, the stress and the sea-ice *velocity* are driven by the atmospheric surface wind as only external forcing. In total, The model has nine prognostic variables, which will be all corrected by the model error correction. We refer to Appendix A for a more technical and complete description of the
regional sea-ice model.

The model's equation are spatially discretised by a first-order continuous Galerkin scheme for the sea-ice velocity components, and a zeroth-order discontinuous Galerkin scheme for all other model variables. The model is integrated in time with a first-order Eulerian implicit scheme, and a semi-implicit fixed point scheme iteratively solves the equations for the velocities, the stress, and the damage. The model area spans $40\,\text{km} \times 200\,\text{km}$ in $x$- and $y$-direction, respectively (Fig. 2a), and we
run the model at two different resolutions, at a $4\,\text{km}$ and a coarsened $8\,\text{km}$ resolution. The integration time step is $8\,\text{s}$ for the high-resolution setup, and $16\,\text{s}$ for the low-resolution.

As external wind forcing, depending on the spatial $x$- and $y$-position and the temporal $t$-position, we impose a surface wind defined by the velocity $u_{\text{a}}(x, y, t)$ in $y$-direction,

$$u_{\text{a}}(x, y, t) = A \cdot \sin\left[\frac{2\pi}{\lambda}(\phi + y + t \cdot \nu)\right] + u_0. \qquad\qquad (3)$$

Given base velocity $u_0$, the wind velocity is sinusoidal with amplitude $A$, wave length $\lambda$, phase $\phi$, and advection velocity $\nu$. To generate different situations in our experiments, the forcing parameters are randomly drawn (cf. Sect. 4), resulting into a velocity field such as depicted in Fig. 2b. As a consequence of such a forcing, the sea ice experiences deformations in localised zones (Fig. 2c), lead to quick transitions between unfractured and completely fractured sea ice (Fig. 2d).

We use von-Neumann boundary conditions, and an inflow of undamaged sea ice. With this model setup, the simulations can
be generally seen as zoomed-in region within an undamaged sea-ice field.

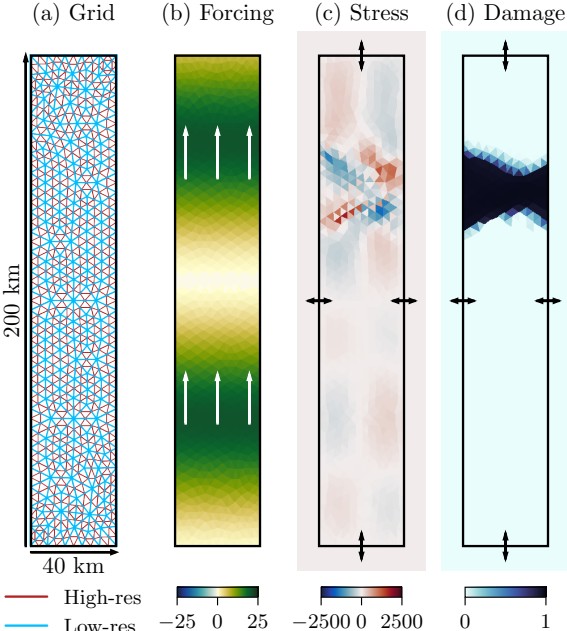

**Figure 2.** (a) The model domain with the high- (red) and low-resolution (blue) grid; (b) snapshot of the surface wind velocity in $y$-direction in $\mathrm{m\,s^{-1}}$, used as wind forcing for the shown case, the white arrows indicate the main movement direction; (c) snapshot of the stress, $\sigma_{xy}$ in $\mathrm{Pa}$, where the arrows correspond to von Neumann boundary conditions on all four sides; (d) snapshot of the damage, where the arrows correspond to an inflow of undamaged sea ice on all four sides. All snapshots are taken at an arbitrary time and represent a typically encountered case in our dataset.

## 2.3 Twin experiments

In our twin experiments, we have two kinds of simulations, as depicted in Fig. 3: we define the low-resolution model setup as our forecast model, which we want to correct towards high-resolution setup as true model. The initial conditions at the high-resolution are integrated with the true model to simulate the truth at the target lead time, in our case $10\,\mathrm{min}$ and $8\,\mathrm{s}$.

To initialise the forecast that should be corrected towards the truth, we project the true initial conditions from the high-resolution to the low-resolution. As projection operator, we make use of the interpolation defined by first-order continuous Galerkin and zeroth-order Galerkin elements, corresponding to Lagrange interpolation with (linear) barycentric and nearest neighbour interpolation, respectively.

To generate the forecast, the initial conditions at the low-resolution are integrated to the target lead time with the forecast model. As we want to reinitialise the forecast model with the corrected model fields later, the model error correction has to be estimated at the low-resolution. To consequently match the resolution of the forecast with the truth, we project the truth at the target lead time to the low-resolution with our previously defined projection operator.

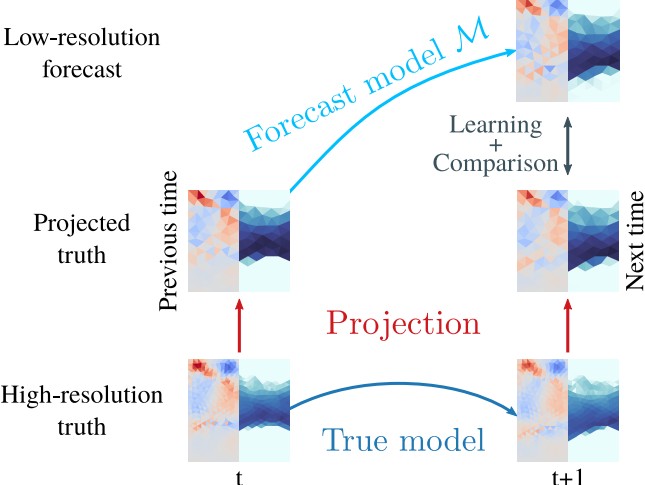

**Figure 3.** In our twin experiments, the high-resolution state with a 4 km resolution is propagated from time $t$ to time $t+1$ with the high-resolution true model; one discrete time step corresponds here to a lead time of 10 min and 8 s. The high-resolution truth at time $t$ is projected by Lagrange interpolation into low-resolution space (4 km resolution), acting as initial state for the forecast. The forecast is performed by the low-resolution forecast model $\mathcal{M}$, which propagates the state from time $t$ to time $t+1$ in the low-resolution space. The model error correction is learned by comparing the low-resolution forecast at time $t+1$ to the truth at the same time, projected into low-resolution space.

The neural network targets the difference between truth and forecast at the low-resolution (cf. Sect. 2.1). Using this strategy and an ensemble of initial conditions and forcing parameters, we generate our training dataset, then used to learn the model error correction. Additionally, we evaluate the performance of the learned model error correction on a similar but independent test dataset.

## 3  A convolutional U-Net baseline

The neural network (NN) should learn to relate the input predictors to the output targets. The inputs and targets are spatially discretised as finite elements, and the NN should directly act on this triangular model grid. Moreover, the NN architecture should be scalable from regional models as used in this study to Arctic-wide models, like neXtSIM. As we expect that the model errors from the sea-ice dynamics have an anisotropic behaviour, we additionally want to directly encode the extraction of localised features with a directional-dependent weighting into the NN. Therefore, as depicted in Fig. 4, we use a NN based on a convolutional U-Net architecture (Ronneberger et al., 2015). For a more technical description of this NN architecture, we refer to Appendix B.

Convolutional NNs are optimised for their use on Cartesian spaces, where they can easily exploit spatial autocorrelations. The model variables are additionally defined on different positions at the triangles: the velocities are defined on the nodes of the triangles, whereas all other variables are constant across a triangle. Consequently, we project from triangular space into

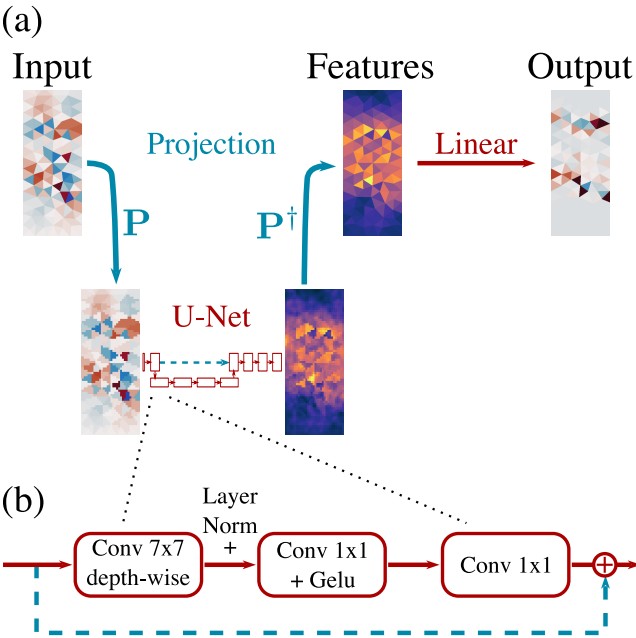

**Figure 4.** In our deep learning approach (a), the input fields are projected by a fixed linear projection operator $\mathbf{P}$ from their triangular space into a Cartesian space that has a higher resolution, where the learnable U-Net extracts features. Back-projected into triangular space by the pseudo-inverse of the linear projection operator $\mathbf{P}^{\dagger}$, these features are combined by learnable linear functions to obtain the output. In our case, the U-Net consists of multiple ConvNeXt-like blocks (b) that have a branch path and a fixed skip connection (i.e., the output of an identity function): in the branch path, a learnable convolutional layer extracts depth-wise, i.e. without mixing the channels, spatial features with a kernel size of $7 \times 7$. The resulting features are layer-normalised and combined by two consecutive learnable convolutions with a $1 \times 1$ kernel and a Gaussian error linear unit (Gelu) activation function in-between. In the end, the features are added to the output of the skip connection. Throughout the Figure, blue coloured connections indicate a fixed function, red colours a learnable function, and dotted lines in the U-Net and ConvNeXt block represent skip connections.

Cartesian space, where the convolutional NN is applied to extract features. As in the projection step from high-resolution model grid to low-resolution grid, we again use Lagrange interpolation with a Barycentric and nearest neighbour interpolation, as in our twin experiments (cf. Sec. 2). To mitigate a possible loss of information by the projection step, we define a Cartesian space with a much higher resolution than the original triangular space.

The U-Net uses convolutional filters and shares its weights across all grid points. This way, the U-Net extracts shift-invariant and localised features which represent common motifs. To learn features at different scales, the features are coarse-grained once in the encoding part of the U-Net (left part of the U-Net in Fig. 4a), and upscaled in the decoding part of the U-Net (right part of the U-Net in Fig. 4a), giving the U-Net its distinct name. We implement the coarse-graining using strided convolutions (Springenberg et al., 2015), where grid points are skipped, and the upscaling with bilinear interpolation followed by a convolution layer (Odena et al., 2016). To retain fine-grained features, the upscaled information is combined with information from

the finer scale by a skip connection (i.e., the output of identity functions), as indicated in Fig. 4a by the horizontal blue dashed line. This allows the U-Net to extract localised features across two scales.

Instead of commonly-used convolutional blocks with standard convolutional filters, followed by a normalisation and non-linear activation function, we make use of blocks inspired from the ConvNeXt architecture (Liu et al., 2022), as shown in Fig. 4b. In these blocks, the feature extraction is split into extraction of features from spatial correlations and correlations across features. This makes the U-Net computationally more efficient and shows empirically an improved performance (cf. Appendix C). After the U-Net has extracted the features, the features are pushed through a rectified linear unit (relu, $x_{\mathsf{out}} = \max(0, x_{\mathsf{in}})$) non-linearity to introduce a discontinuity in the features, which empirically helps the NN to represent sharp transitions in the level of damage (cf. Appendix D4).

The extracted features are projected back from the Cartesian space into the triangular space. Because the projection operator is purely linear, the back-projection operator can be analytically estimated by the pseudo-inverse of the projection matrix. As the Cartesian space is higher-resolved, the back-projection averages the features of several Cartesian elements into features of single triangular elements.

Back in the triangular space, the extracted features are combined by learnable linear functions. These linear functions process each element-defining grid point independently but using the same weights across all grid points. To estimate their own model error correction out of the features, each of the nine model variables has its own linear function.

In total, by projecting the input into a Cartesian space, the convolutional U-Net extracts features, which are then the basis for the estimation of the output in the original triangular space. The use of the U-Net allows us to extract localised features and an efficient implementation, even for Arctic-wide models. The extraction of features at a higher resolution bundled with their combination in triangular space makes the NN directly applicable for finite-element models.

## 4   Data generation and training

We train and test different NNs with twin experiments, using the regional sea-ice model, as described in Sect. 2. We simulate high-resolution truth trajectories with a resolution of $4\,\mathrm{km}$ and an integration step of $8\,\mathrm{s}$, and low-resolution forecasts with a $8\,\mathrm{km}$ resolution and a $16\,\mathrm{s}$ step. The NNs are trained to correct these low-resolution forecasts for a lead time of $10\,\mathrm{min}$ and $8\,\mathrm{s}$.

We train the NNs on an ensemble of 100 trajectories. The NN hyperparameters, like the depth of the network or the number of channels, are tuned against a distinct validation dataset with 20 trajectories. Finally, the scores are estimated using an independent test dataset with 50 trajectories.

All high-resolution are initialised with a randomly chosen cohesion field and randomly drawn forcing parameters, as specified in Table 1. These parameters are chosen such that most trajectories have fractured sea ice in different regions of the simulated domain. The low-resolution setup uses the same forcing parameters, whereas the cohesion field is one of the prognostic model variables and, hence, projected to the low-resolution.

**Table 1.** The random ensemble parameters and their distribution, $U(a,b)$ specifies a random variable drawn from a continuous uniform distribution with its two boundaries $a$ and $b$. The cohesion is independently drawn for each grid point and ensemble member, whereas each ensemble member has one set of forcing parameters.

| Description | Value |
| --- | --- |
| Cohesion $C$ | $U(5 \times 10^3 \,\mathrm{Pa}, 1 \times 10^4 \,\mathrm{Pa})$ |
| Amplitude $A$ | $U(8 \,\mathrm{m\,s}^{-1}, 20 \,\mathrm{m\,s}^{-1})$ |
| Wave length $\lambda$ | $U(50 \,\mathrm{km}, 200 \,\mathrm{km})$ |
| Phase $\phi$ | $U(-100 \,\mathrm{km}, 100 \,\mathrm{km})$ |
| Advection $\nu$ | $U(-0.5 \,\mathrm{m\,s}^{-1}, 0.5 \,\mathrm{m\,s}^{-1})$ |
| Base velocity $u_0$ | $\max(20 \,\mathrm{m\,s}^{-1} - A, U(0 \,\mathrm{m\,s}^{-1}, 10 \,\mathrm{m\,s}^{-1}))$ |

Defining the truth trajectories, the high-resolution simulations are run for three days of simulation time. The forcing is linearly increased to its full strength as in Dansereau et al. (2016) during the first day of simulation, which is consequently treated as spin-up and omitted from the evaluation. Over the subsequent two days, the truth trajectories are hourly sliced to obtain the initial conditions. Projected into low-resolution, the initial conditions are integrated with the forecast model until the forecast lead time of $10 \,\mathrm{min}$ and $8 \,\mathrm{s}$. To generate the datasets for the training of the NNs, these forecasts are compared to the
projected truth fields at the same lead time.

    These datasets contain input-target pairs. The inputs for the NNs consist of 20 fields: nine forecast model fields and one forcing field for the initial conditions and the forecast lead time. The targets are the difference between the projected truth and the forecasted state at the forecast lead time, and consist of nine fields. The inputs and targets are normalised by a global per-variable mean and standard deviation, both estimated from the training dataset.

The hourly slicing gives us $48$ samples per trajectory, resulting into $4800$, $960$, and $2400$ samples for the training, validation, and test dataset, respectively. In total, the training dataset has $12.3 \times 10^6$ degrees-of-freedom (number of samples $\times$ number of variables $\times$ number of grid points). The NN configuration used in our experiments (cf. Table B1 in Appendix B) has $1.2 \times 10^6$ parameters; an order of magnitude smaller than the degrees of freedom in the training dataset. During training, the NNs experience no overfitting, even if only $10\,\%$ of the training data is used, as shown in Appendix D1.

We train the NNs by minimising a loss function proportional to a weighted mean absolute error (MAE), a more rigorous treatment of the loss function can be found in Appendix B3. The MAE is estimated for each variable independently. To average these MAEs across all variables, the individual MAEs are weighted by a per-variable weight. The weights are learned alongside the NN and can be seen as uncertainty estimate from the training dataset. In our case, the weighted MAE empirically performs better than a weighted mean-squared error loss function and if the weighting is automatically learned from data (Appendix D3).

If not otherwise specified, all NNs are trained for 1000 epochs with a batch size of $64$. To optimise the NNs, we use Adam (Kingma and Ba, 2017) with a learning rate of $\gamma = 3 \times 10^{-4}$, $\beta_1 = 0.9$, and $\beta_2 = 0.999$. We refrain of learning rate decay or early stopping, as such methods would make the experiments harder to compare.

All experiments are performed on the CNRS/IDRIS (French National Centre for Scientific Research) Jean Zay supercomputer, using a single NVIDIA Tesla V100 GPU or NVIDIA Tesla A100 GPU per experiment. The NNs are implemented in PyTorch (Paszke et al., 2019) with PyTorch lightning (Falcon et al., 2020) and Hydra (Yadan, 2019). The code is publicly available under https://github.com/cerea-daml/hybrid_nn_meb_model.

## 5 Results

We propose a baseline architecture based on the U-Net, as described in Sect. 3, in the following simply called U-NeXt. We have selected the parameters of the U-NeXt architecture (cf. Table B1 in Appendix B) after a randomised hyperparameter screening in the validation dataset with 200 different network configurations.

We evaluate our trained NNs on the test dataset with the mean absolute error (MAE) in the low-resolution. To get comparable performances across the nine model variables, we normalise their errors by their expected MAE in the training dataset. Note that this normalisation results into a constant weighting, differing from the adaptive weighting used during the training process, which depends on the training trajectory. Furthermore, this normalisation allows us to estimate the performance of the NNs with a single metric, averaged over all model variables. The NNs are trained ten times with different random seeds ($s \in [0, 9]$), and all results are averaged over the ten trained networks.

As baseline method, we use a persistence forecast with the initial conditions as constant prediction. We additionally compare the forecasts corrected by the NN to the uncorrected forecasts from our sea-ice model.

In the following, we discuss the results on the test dataset in Sect. 5.1, what we can learn about the residuals by analysing the sensitivity of the NN to its inputs in Sect. 5.2, and how we can combine the NN with the geophysical model for lead times up to one hour in Sect. 5.3.

### 5.1 Performance on the test dataset

In a first step, we evaluate the performance of our model error correction on the test dataset, without applying the correction together with the geophysical model, Table 2. For additional results, we refer to Appendix C and Appendix D, where we among other things compare with other NN architectures, other loss functions, and other activation functions.

The NN corrects the model forecasts across all variables. This results in an averaged gain of the hybrid model over $75\%$ compared to the sea-ice model. For the stress, damage, and area, the persistence forecast performs better than the sea-ice model, as the model forecast drifts towards the attractor of the low-resolution model setup, as discussed in Sect. 5.3. Since the NN uses the initial conditions as input, the hybrid model surpasses the performance of persistence, even for variables where persistence is better than the sea-ice model. In Appendix C, we show that the model error of the sea-ice model is mostly driven by a dynamical error such that simply correcting the bias has almost no impact on the performance. In total, the NN consistently improves the forecast on the test dataset.

To apply CNNs to the raw data of our finite-elements-based sea-ice model, we project from triangular to Cartesian space, where the features are extracted. The number of elements in the Cartesian space determines its effective resolution and, thus,

**Table 2.** Normalised MAE on the test dataset, estimated in low-resolution, and averaged over ten NNs trained with different seeds. Reported are the errors for the velocity component in $y$-direction $v$, for the stress component $\sigma_{yy}$, the damage $d$, and the area $A$. The mean $\overline{\Sigma}$ is the error averaged over all nine model variables, including the non-shown ones. A score of one would correspond to the MAE of the sea-ice model in the training dataset. Bold scores are the best scores in a column. For a table with standard deviation across seeds, we refer to Table C1.

| Name | $v$ | $\sigma_{yy}$ | d | A | $\overline{\Sigma}$ |
|---|---|---|---|---|---|
| Persistence | 0.37 | 0.29 | 0.60 | 2.37 | 0.79 |
| Sea-ice model | 1.14 | 0.91 | 1.09 | 0.94 | 1.03 |
| Hybrid model | **0.23** | **0.17** | **0.38** | **0.33** | **0.24** |

the finest scale on which the NN can extract features. To demonstrate the effect of different resolutions on the result, we perform three different experiments, where we change the grid size, while keeping the NN architecture the same (Table 3).

**Table 3.** Normalised MAE on the test dataset for different Cartesian grid sizes, $x$-direction $\times$ $y$-direction. The error components are estimated as in Table 2. The training loss is estimated as the expected Laplace negative log-likelihood, averaged over the training dataset, variables, and ten NNs trained from different seeds. The bold grid size is the used grid size and bold scores are the best scores in a column.

| Grid size | Loss | $v$ | $\sigma_{yy}$ | d | A | $\overline{\Sigma}$ |
|---|---|---|---|---|---|---|
| $8 \times 32$ | -9.31 | 0.29 | 0.42 | 0.64 | 0.72 | 0.50 |
| $16 \times 64$ | **-18.58** | 0.26 | 0.18 | 0.41 | 0.43 | 0.28 |
| $\mathbf{32 \times 128}$ | -18.47 | **0.23** | **0.17** | **0.38** | **0.33** | **0.24** |

The training loss, here the negative Laplace log-likelihood, measures how well a NN can be fitted towards the training dataset. Although its resolution is higher than the original resolution of $8\,\mathrm{km}$, the back-projection for the $8 \times 32$ grid is underdetermined, as the mapping is non-surjective, degrading the performance of the NN. Starting at the $16 \times 64$ grid, the Cartesian space covers all triangular grid points, and all NNs have a similar predictive power with similar training losses. Nevertheless, the MAE of the finest $32 \times 128$ grid is the lowest for all variables. As we keep the architecture the same for all resolutions, the higher the resolution, the smaller the receptive field of the NN. At the highest resolution, the NN is thus forced to extract more localised features. Such localised features seem to better represent the processes needed for the prediction of the residuals and for parametrising the subgrid-scale; this improvement by using a finer Cartesian space will be discussed more in detail in Sect. 6.

In Fig. 5, we visualise a typical output of the U-Net before it gets projected back into triangular space and linearly combined. The higher the resolution, the sharper and more fine-grained the feature map. Sharper features can better represent anisotropy and discrete processes in sea ice. Exhibiting more fine-grained motifs, in the highest resolution, Fig. 5c, the network can extract features along the $x$- and $y$-direction and can even represent small-scale structures in diagonal directions. These fine-grained features indicate an ability to parametrise the effect of the subgrid-scale onto the resolved scales. Moreover, as a consequence

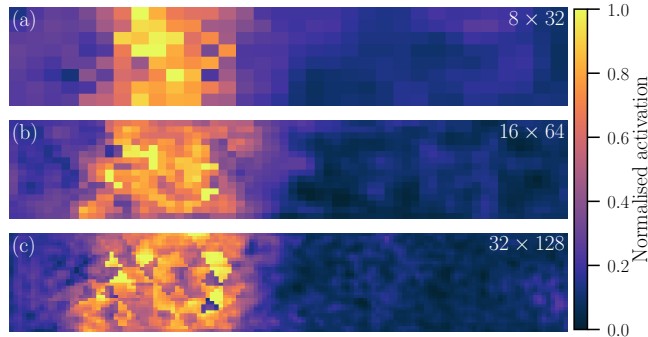

**Figure 5.** Normalised feature map in Cartesian space for grid sizes of (a) $8 \times 32$, (b) $16 \times 64$, and (c) $32 \times 128$. The feature map is estimated based on the same sample in the test dataset. The specific feature maps are selected such that the extracted features qualitatively match for all three resolutions. As the maps might have different order of magnitudes, they are normalised by their 99-th percentile for visualisation purpose, the colours are thus proportional to the activation.

of the extraction of more localised features for finer spaces, the NN also localises the background noise such that the field appears to be much noisier in the case of inactive zones, where the activation is low.

## 5.2 Sensitivity to the input

The inputs of the NN have a crucial impact on the performance of the model error correction. In the following, we evaluate the sensitivity of the NN with respect to its input variables. In a first step, we alter the input and measure the resulting performance of the NN with the normalised MAE, Table 4.

**Table 4.** Normalised MAE on the test dataset for different input sets. The error components are estimated as in Table 2. $n_{in}$ corresponds to the number of input channels. The bold scores are the best scores in a column.

| Name | $n_{in}$ | $v$ | $\sigma_{yy}$ | d | A | $\overline{\Sigma}$ |
|---|---|---|---|---|---|---|
| Initial only | 10 | 0.63 | 0.63 | 0.77 | 0.34 | 0.60 |
| Forecast only | 10 | 0.66 | 0.62 | 0.75 | 0.35 | 0.60 |
| Both | 20 | 0.23 | 0.17 | 0.38 | 0.33 | 0.24 |
| W/o forcing | 18 | 0.24 | 0.18 | 0.37 | 0.33 | 0.25 |
| Difference only | 10 | 0.19 | **0.15** | 0.37 | 0.30 | 0.23 |
| + initial state | 20 | **0.17** | **0.15** | **0.33** | **0.26** | **0.21** |
| + forecasted state | 20 | **0.17** | **0.14** | **0.33** | **0.26** | **0.21** |

Usually, only the initial conditions are used for a neural-network-based model error correction (Farchi et al., 2021a). As sea-ice dynamics is a first-order Markov system, the results are very similar when using only the initial conditions or only the

forecast state as input. Compared to input from a single time, using both times as input improves the prediction by around 60 %. In this case, the NN learns to correct the model error based on the difference between the forecast and initial conditions, representing the sea-ice dynamics. If only a single time is used as input, the NN has to internally learn an emulator of the dynamics. Explicitly giving the difference to the NN instead of raw states improves the correction, although the number of predictors is halved. With the difference, the network has directly access to the model dynamics. Further adding the initial conditions or forecasted state to the difference improves the correction; the network has then access to relative and absolute values.

In the second step, we analyse how the input variables influence the output of the NN. As we want to quantify the impact of the dynamics on the output, we base the analysis on the previous "Initial + Difference" experiment from Table 4. As global measure, we use the permutation feature importance (Breiman, 2001): the NN is applied several times, each time, another input variables is shuffled across the samples. By shuffled an input variable, its information is destroyed, and the output of the NN is changed. This possibly changes the prediction error compared to the unperturbed original output. Focussing on active regions, we measure the errors with the RMSE, estimated over the whole test dataset. The higher the RMSE for a shuffled input variable, the higher the importance for this variable onto the errors, as summarised in Table 5. Because the information of only single variables are destroyed, the permutation feature importance is sensitive to correlated input variables (Appendix D5). Consequently, the inter-variable importance in Table 5 is likely underestimated.

All model variables are highly sensitive to their own dynamics. Furthermore, the feature importance reflects the relations inherited by the model equations (cf. Sect. 2.2, Dansereau et al., 2017). For instance, caused by the dependence of the thickness upon the sea-ice area, they are linked together in the input-output relation. The wind forcing externally drives and influences the sea-ice velocity in $y$-direction, $v$. The $v$-velocity, however, advects and mixes the cohesion, area, and thickness. By modulating the momentum equation and mechanical parameters, respectively, the area and thickness influence the velocity and stress components. In total, for each model variable, their dynamics are in fact the single most important input variable, on which basis the neural network extracts features.

As local measure, we move to the sensitivity $\frac{\partial f(\boldsymbol{x},\boldsymbol{\theta})}{\partial \boldsymbol{x}}$ of the NN output to its input fields (Simonyan et al., 2013), again for the "Initial + Difference" experiment. To showcase what the NN has learned in spatial meanings, we concentrate here on a single grid point in a single prediction for the sea-ice area. The initial conditions, the dynamics, the forecast error, and the NN prediction for the sea-ice area is shown in Fig. 6a–d. To smooth the sensitivity and reduce its noise, we perturb the input variables 128 times with noise drawn from $\mathcal{N}(0, 0.1^2)$, and average the sensitivity over these noised versions (Smilkov et al., 2017). The resulting saliency maps (Fig. 6e–h) show which grid points influence the selected output. The larger its amplitude, the more sensitive is the output to that grid point.

For the selected grid point, the prediction is especially sensitive to the area itself and the thickness, in absolute values, Fig. 6e, and their dynamics, Fig.6f. This underlines the already mentioned relation between the sea-ice area and thickness, and confirms the global results of the permutation feature importance in Table 5. The sensitivity additionally exhibits a strong localisation for the damage dynamics, Fig. 6g, and is directional dependent to the velocity dynamics, Fig. 6h. Hence, the NN seems to rely on localised and anisotropic features to predict the residual.

**Table 5.** The permutation feature importance of the RMSE for the given output variable with respect to the input variable for "Initial + Difference" as input, estimated over the whole test dataset. The numbers show the multiplicatively RMSE increase of a specific output variable (row) if a given input variable (column) is permuted, a higher number corresponds to a higher feature importance. The colours are normalised by the highest feature importance for a given row (output variable) and proportionally to the feature importance. The "Difference" variables specify the difference of the forecast state to the initial conditions as input into the NN.

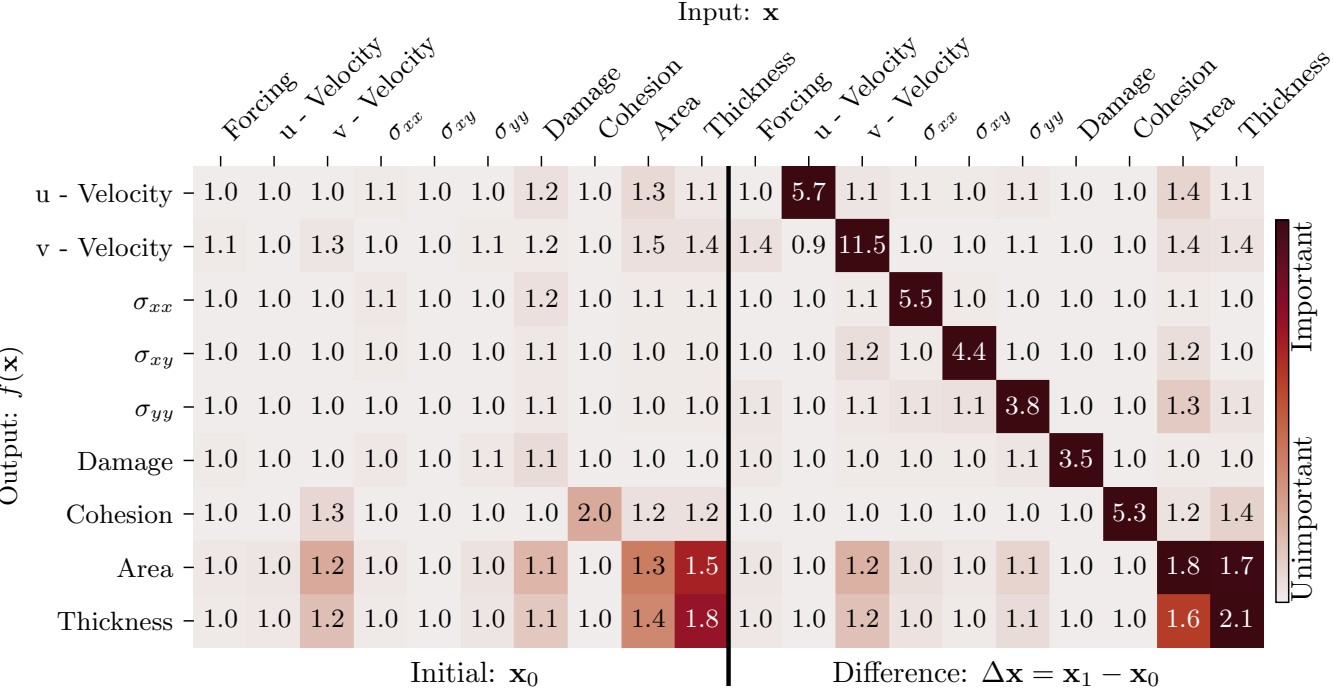

Based on these sensitivities, we can interpret what features the NN has learned, guiding us towards a physical meaning of the model errors. The diametral impacts of the thickness and area in absolute values and dynamics indicate that the sea-ice model tends to overestimate the effect of the dynamics, whereas the initial conditions have a stronger persisting influence than predict by the model. However, the connectivity between grid points is underestimated by the model, as seen in Fig.6f. In general, the model overestimates the fracturing process leading to a mean error of $2.31 \times 10^{-3}$ for the damage in the training dataset. This overestimation of fracturing could also explain the very localised impact of the damage dynamics, Fig.6g. The directional dependency on the velocity dynamics, Fig.6h, additionally indicates an overestimation of the effects of the velocity divergence; if fracturing processes are induced by divergent stresses, the NN tries to decrease their impact on the sea-ice area.

In general, this analysis has shown that the NN relies not only on a single time step as predictor but on how the fields develop in time, indicating that the dynamics themself are the biggest source of model error between different resolutions. Additionally, the network extracts localised and anisotropic features, which are physically interpretable and point towards general shortcomings in the dynamics of the sea-ice model.

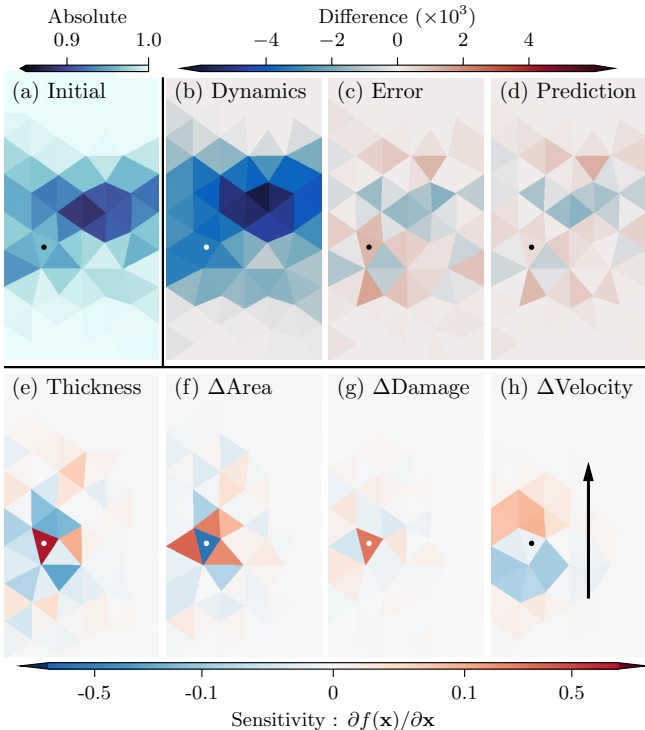

**Figure 6.** Snapshots at an arbitrary time of (a) sea-ice area at initial time, (b) the difference in area between forecast time and initial time, (c) difference in area between projected truth and forecast at forecast time, and (d) prediction of the NN for the area. In the lower part, we show the sensitivity of the prediction for the area at a chosen grid point, indicated by a white or black dot, on (e) the thickness at initial time, (f) the difference in area, (g) the difference in damage, and (h) the difference in velocity in $y$-direction. The black arrow in (h) indicates the main sea-ice movement direction.

## 5.3 Forecasting with model error correction

After establishing the importance of the dynamics for the error correction, we use the error correction together with the low-resolution forecast model for short-term forecasting. As trained for a forecast horizon of $10$ min and $8$ s, we apply the NN to correct the forecasted states every $10$ min and $8$ s. Because the prognostic sea-ice thickness is represented as a ratio between the actual sea-ice thickness and the area, its error distribution can have very fat tails and can be non-well-behaved. Thus, we predict as output the actual sea-ice thickness, then, as post-processing step, translated into the prognostic sea-ice thickness. We additionally enforce physical bounds on all variables, by limiting the values to physical reasonable bounds after error correction. We change the performance metric to be the RMSE, a commonly used metric to evaluate forecast performances. We evaluate the performance across all $2400$ hourly time slices on the test dataset. For forecasting purpose, the NN with the initial and forecasted fields as input performs generally better than the NN with initial and difference fields (Appendix D6); for simplification in the following, we use only the NN with the initial and forecasted fields, calling it again "Hybrid model".

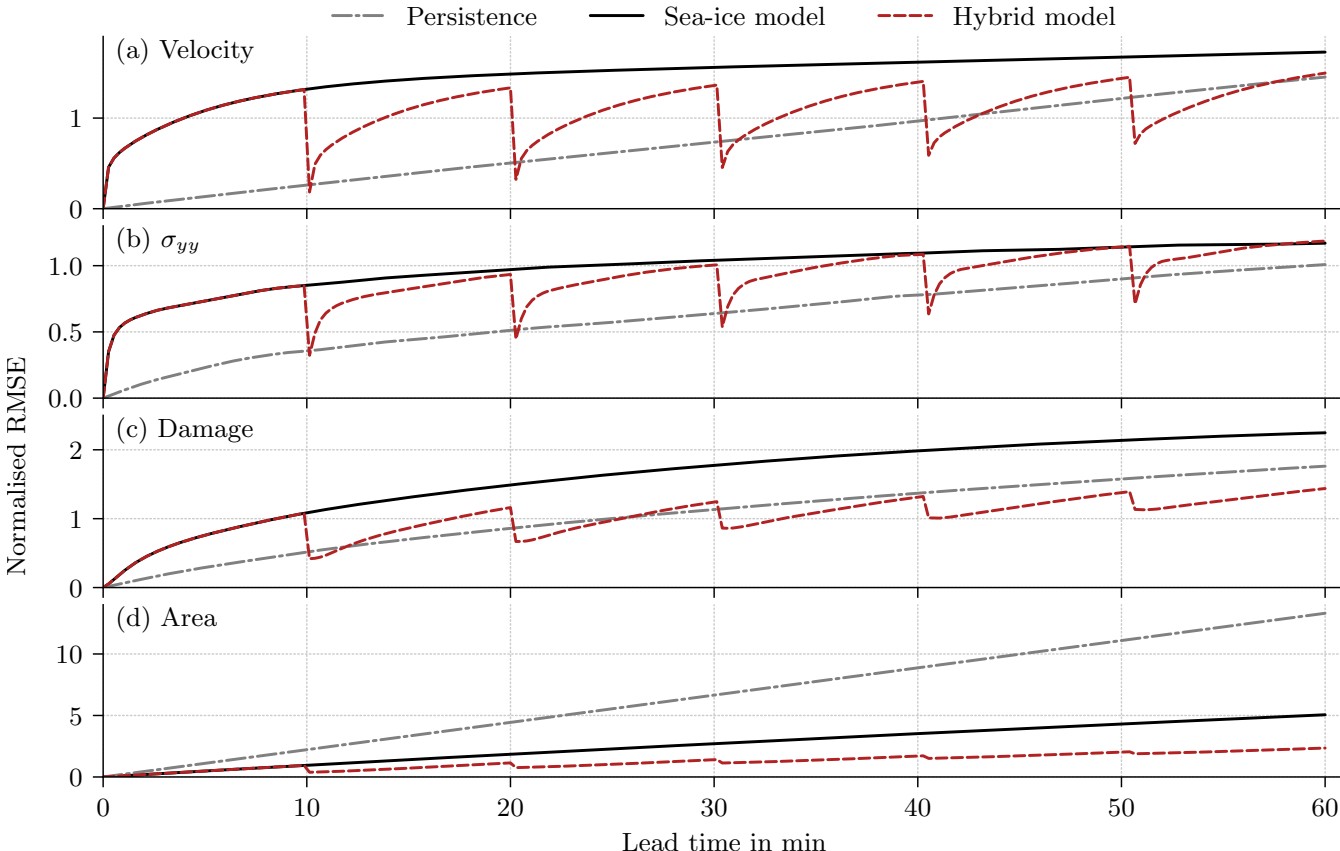

**Figure 7.** Normalised RMSE for (a) the velocity in $y$-direction, (b) the divergent stress in $y$-direction, (c) the damage, and (d) the sea-ice area as function of lead time on the test dataset, normalised by the expected RMSE on the training dataset for a lead time of $10$ min and $8$ s. In the hybrid model, the forecast is corrected every $10$ min and $8$ s, and the performance is averaged over all ten networks trained with different random seeds.

Overall, the hybrid models surpass the performance of the original geophysical model (Fig. 7). However, for the forecast with the sea-ice model and the hybrid model, a strong drift is evident. As correcting the bias has almost no impact on the performance in the test dataset (Appendix C), this drift is not caused by model biases. Instead, the projected initial state lays not on the attractor of the forecast model, the forecast drifts towards the model attractor; a behaviour similar to what is typical in seasonal or decadal climate predictions initialised with observed data. This results in large deviations between geophysical forecast and projected truth, and the persistence forecast is better than the forecast model for the velocity, the stress, and the damage. And yet, correcting the model states with NNs improves the forecasts at correction time, even compared to persistence. Even so, the correction nudges the forecast towards the projected truth and out of the attractor. Consequently, between two consecutive updates, the forecast drifts again towards the attractor when the model runs freely, which leads to a decreased

impact of the error correction. Nevertheless, the accumulated model error correction results into an improved forecast for a lead time of $60$ min (Table 6), especially for the sea-ice area and damage, even if the last correction is already $9$ min ago.

**Table 6.** Normalised RMSE on the test dataset for a lead time of $60$ min. The last update in the hybrid models was at a lead time of $50$ min and $40$ s. The errors are normalised by the expected standard deviation for a lead time of $60$ min on the training dataset. The symbolic representation of the variables has the same meaning as in Table 2. The bold scores are the best scores in a column.

| Name | $v$ | $\sigma_{yy}$ | d | A | $\overline{\Sigma}$ |
|---|---|---|---|---|---|
| Persistence | **1.13** | **0.81** | 0.83 | 2.58 | 1.19 |
| Sea-ice model | 1.34 | 0.93 | 1.06 | 0.98 | 1.06 |
| Hybrid model | 1.16 | 0.95 | **0.68** | **0.46** | **0.81** |

    The forecast error generally increases with lead time, but the error reduction gets smaller with each update, especially for the sea-ice area. Since the NN correction is imperfect, the error during the next forecast cycle is an interplay between the errors from the initial conditions and from the model. The NN is trained with perfect initial conditions to correct the model error

only. As the influence of the initial condition error increases with each update, the error distribution shifts, and the statistical relationship between input and residual changes with lead time; the network can correct less and less forecast errors. This effect has an larger impact on the forecast if the lead time between two corrections with the NN is further reduced (Appendix D2).

    To show the effect of this error distribution shift, we analyse the differences between the first and fifth update step with the centred spatial pattern correlation (Houghton et al., 2001, p. 733) between the NN prediction and the true residual: we centre all

fields by removing their mean, and estimate Pearson's correlation coefficient between the prediction and the residual in space. By centring the fields, we omit the influence of the amplitudes upon the performance of the NN. The higher the correlation, the higher the similarity in the patterns between the prediction and the residual, and a correlation of $1$ would indicate a perfect pattern correlation.

    The correlations are estimated over space for each test sample and variable independently and averaged via a Fisher z-

transformation (Fisher, 1915): the single correlations are transformed by the inverse hyperbolic tangent function. In transformed space, the values are approximately Gaussian distributed, and we average them across samples. The average is transformed back by the hyperbolic tangent function.

**Table 7.** Centred pattern correlation on the test dataset between the updates and the residuals for the first update and fifth update. The symbols of the variables are the same as in Table 2.

| Update | $v$ | $\sigma_{yy}$ | d | A | $\overline{\Sigma}$ |
|---|---|---|---|---|---|
| First update | 0.94 | 0.99 | 0.93 | 0.92 | 0.98 |
| Fifth update | 0.70 | 0.89 | 0.59 | 0.28 | 0.76 |

Since they are trained for this, the NN can almost perfectly predict the residual patterns for the first update. At the fifth update, larger parts of the residual patterns are unpredictable for our trained NN. Especially the sea-ice area has a longer memory for error corrections such that the predicted patterns are almost unrelated to the residual patterns for the fifth update. Caused by the drift towards the attractor, the sea-ice model forgets parts of the previous error correction for the velocity and divergent stress component, and these forgotten parts get corrected again in the fifth update. However, the pattern correlation is also decreased for these dynamical variables for the fifth update. Based on these results, the error distribution shift is one of the main challenges for the application of such model error corrections for forecasting.

Our proposed parametrisation is deterministic and is designed to target the median value. On the resolved scale, sea-ice dynamics can look stochastically noised, with suddenly appearing strains and linear kinematic features, as discussed in the introduction. We show the effect of the seemingly stochastic behaviour in Fig. 8 with the temporal development of damage and total deformation for the high-resolution simulation, the forecast model without parametrisation, and the parametrised hybrid model.

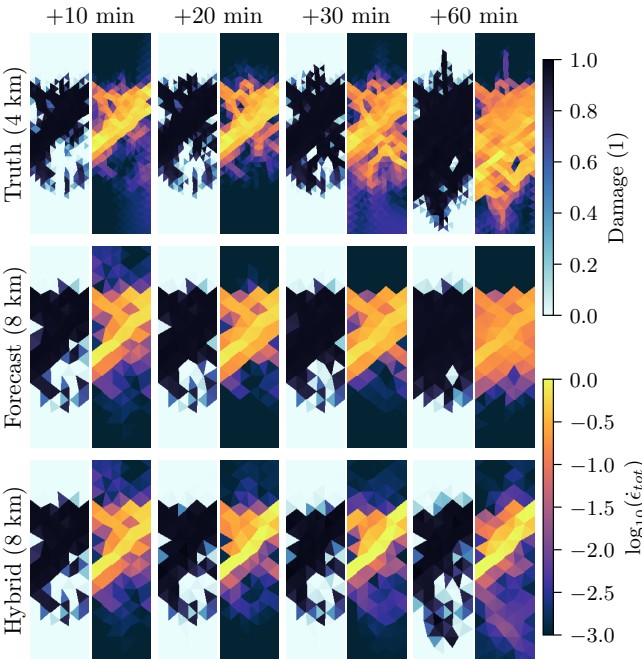

**Figure 8.** Snapshots of damage (left) and total deformation (right), showing their temporal evolution, in the high-resolution simulation (top), for the low-resolution forecast model (middle), and the low-resolution hybrid model (bottom).

The initial state exhibits damaged sea ice in the centre, corresponding to a diagonal main strain in the total deformation. In the high-resolution simulation, the damaging process continues, leading to more widespread damaging of sea ice. Related to new strains, the damage is extended towards the south. The low-resolution forecast model only diffuses the deformation without remaining main strain in the already damaged sea ice. As a result, the model misses the southward-extending strain

and damaging process. Furthermore, the model extends the damage and deformation southwards, although the newly developed
strain is weaker than in the high-resolution. The parametrisation can represent widespread damaging of sea ice. However, the
parametrisation misses the development of new strains and positions the main strain at the wrong place. This problem can
especially occur on longer forecasting time scales, where the damage field is further developed compared to its initial state.
Therefore, we see the need for parametrisations that can also reproduce the stochastic effects of subgrid-scales onto the resolved
scales.

## 6 Summary and discussion

We have introduced an approach to parametrise subgrid-scale dynamical processes in sea-ice models based on deep learning
techniques. Using twin experiments with a model of sea-ice dynamics that implements a Maxwell-elasto-brittle rheology, the
NN learns to correct low-resolution forecasts towards high-resolution simulations for a forecast lead time of $10$ min and $8$ s.

Our results show that NNs are able to correct model errors related to the sea-ice dynamics and can thus parametrise the
unresolved subgrid-scale processes as for other Earth system components. In addition, we are able to directly transfer recent
improvements in deep learning, like ConvNeXt blocks (Liu et al., 2022), to ameliorate the representation of the subgrid-scale.
Instead of parametrising single processes, we correct all model variables at the same time with one big NN, here with $1.5 \times 10^6$
parameters.

For feature extraction, we map from the triangular model space into a Cartesian space with a higher resolution to preserve
correlations of the input data. Our results show hereby that higher-resolved Cartesian spaces improve the parametrisation; the
network can then extract more information about the subgrid-scale. In the Cartesian space, a convolutional U-Net architecture
extracts localised and anisotropic features on two scales. Mapped back into the original triangular space, the extracted features
are linearly combined to predict the residuals, which parametrises the effect of the subgrid-scale upon the resolved scales.
Therefore, using a mapping into Cartesian space, we can apply CNNs to Arctic-wide models with unstructured grids, like
neXtSIM.

Our results suggest that the finer the Cartesian space resolution, the better the performance of the NN. This improvement
could emerge from our type of twin experiments, where the main difference in the resolved processes is a result of different
model resolutions. Consequently, extracting features at a higher resolution than the forecast model might be needed to represent
the processes of the higher-resolution simulations; the NN would act as an emulator for these processes. In this case, the
resolution needed for the projection would be linked to the resolution of the targetted simulations. However, in the light of our
results, this link seems to be unlikely: the performance of the finer $32 \times 128$ grid is higher than the $16 \times 64$ grid, although the
latter one has already a higher resolution than the grid from our targetted simulations. Additionally, the link cannot explain the
increased training loss but decreased test errors for the finer grid.

The gain likely results from an inductive bias in the NN for Cartesian spaces with higher resolutions. We keep the NN
architecture and its hyperparameters, like the size of the convolutional kernels, the same, independent of the resolution in the
Cartesian space. Consequently, viewed from the original triangular space, the receptive field of the NN is reduced by increasing

the resolution. The function space representable by such NN is more restricted, and, as the fitting power is reduced, the training loss increases again. The NN is biased towards more localised features. These localised features help the network to represent previously unresolved processes better. This better representation improves the generalisation of the NN, resulting into lower test errors. However, as this study is performed with twin experiments in very specific settings, it remains unknown to us if the projection into a space that has a higher resolution is advantageous for subgrid-scale parametrisations in general.

The permutation feature importance as global feature importance and sensitivity maps as local importance help us to explain the learned NN by physical reasoning. The sensitivity map has additionally shown that the convolutional U-Net can extract anisotropic and localised features, depending on the relation between input and output. We see such an analysis as especially relevant for subgrid-scale parametrisations learned from observations, as the feature importance can be utilised to find the sources of model errors and guide model developments.

Applying the NN correction together with the forecast model improves the forecasts up to one hour. Since the error correction is imperfect, the initial condition errors accumulate for longer forecast horizons. The longer the forecast horizon, the less are the targetted residuals in the training data representative for the true residuals. Such issues would be solved in online training of the NNs (Rasp, 2020; Farchi et al., 2021a), which could be nevertheless too costly for real-world applications. Offline reinforcement learning additionally tackles similar issues (Levine et al., 2020; Lee et al., 2021; Prudencio et al., 2022) and can be thus a way to partially solve them.

Although the here-learned NNs can make continuous corrections, they represent only deterministic processes. As the evolution of sea ice propagates from the subgrid-scale to larger scales, unresolved processes can appear like stochastic noise from the resolved point of view. Consequently, the deterministic model error correction is unable to parametrise such stochastic-like processes, which can lead for example to wrongly-positioned strains and linear kinematic features. Generative deep learning (Tomczak, 2022) can offer a solution to such problems and could introduce a form of stochasticity into the subgrid-scale parametrisation, e.g. by meanings of generative adversarial networks (Goodfellow et al., 2014) or denoising diffusion models (Sohl-Dickstein et al., 2015). Such techniques can also be used to learn the loss metric, circumventing issues by defining a loss function for training.

Because of missing subgrid-scale processes in the low-resolution forecast model, the high-resolution simulations, projected into the low-resolution, are far off the low-resolution attractor. Consequently, when the forecast is run freely, it drifts toward its own attractor, resulting into large deviations to the projected high-resolution states. This difficult forecast setting is indeed quite realistic, as models miss also in reality subgrid-scale processes (Bouchat et al., 2022; Ólason et al., 2022), such that empirical free-drift or even persistence forecasts are difficult to beat with forecast models (Schweiger and Zhang, 2015; Korosov et al., 2022). As the attractor of the forecast models does not match the attractor of the observations or the projected high-resolution state, also finding the best state on the model attractor would not necessarily lead to an improved forecast (e.g. Stockdale, 1997; Carrassi et al., 2014). The only way is therefore to improve the forecast model, thereby changing its attractor, e.g., by directly parametrising the subgrid-scale processes with a tendency correction.

A subgrid-scale parametrisation can be generally seen as a kind of forcing. Here, we use a resolvent correction, where we correct the forecast model with NNs at integrated time steps; the parametrisation is like Euler integrated in time. Our results

show that the NN needs access to the dynamics of the model to correct tendencies related to the drift towards the wrong attractor, at least at correction time. One strategy can be thus to increase the update frequency or to distribute the correction over an update window, similarly to incremental analysis update in data assimilation (Bloom et al., 1996). Another strategy is to use tendency corrections (Bocquet et al., 2019; Farchi et al., 2021a), where the parametrising NN is directly incorporated as an external forcing term into the model equation. As the tendency correction is included into the model itself, it also changes and possibly corrects the attractor. Needed to train such a tendency correction (Farchi et al., 2021a), the adjoint model is typically unavailable for large-scale sea-ice models. To remedy such needs, one could train the NN as a resolvent correction and scale the correction to a tendency correction.

This study and its experiments are designed to be a proof-of-concept. The NN is able to correct model errors, our results nevertheless indicate shortcomings and challenges towards an operational application of such subgrid-scale parametrisations. Our sea-ice model exhibits a strong drift towards its own attractor, which leads to large difference between simulations at different resolutions. It is yet unknown for us if this strong drift is only evident in our model or if it prevails also for other sea-ice models. Nevertheless, the NN should ideally take the models's attractor into account such that the corrected states stay on this attractor.

Additionally, the NN is trained to correct forecasts for a specific model setup and a specific model resolution. Normally, the NN has to be retrained for other setups, and especially other resolutions. However, we might be lucky in correcting model errors from sea-ice dynamics: as sea-ice dynamics are temporally and spatially scale-invariant for resolutions up to at least $1\,\mathrm{km}$, we might be able to apply the same model error correction for different resolutions. In any case, the NN trained for one resolution could be used as starting point to fine-tune it towards another resolution.

In our case, we apply twin experiments, where we train the NN to correct forecasts with perfectly known initial conditions towards a high-resolution simulation. Although such training is simple and in our case sufficient, the NN suffers from an error distribution shift. Applying twin experiments for the training of subgrid-scale parametrisations, the NN learns to emulate processes of the high-resolution simulations. Such an emulation could allow us to achieve a similar performance with low-resolution simulations as with high-resolution simulations, which would speed-up simulations. However, in this case, the NN learns instantiations of already known processes.

Instead, subgrid-scale parametrisations should be ideally learned by incorporating observations into the forecast. This way, the parametrisation could learn to incorporate processes which might be unknown yet. Such learning from sparsely-distributed observations can be enabled by combining machine learning with data assimilation (Bocquet et al., 2019, 2020; Brajard et al., 2020, 2021; Farchi et al., 2021b; Geer, 2021). Therefore, we see this combination as one of the next step towards the goal of using observations to learn data-driven parametrisations for sea-ice models.

## 7 Conclusions

Based on our results for twin experiments with a sea-ice dynamics-only model in a channel-setup, we conclude the following:

- Deep learning can correct forecast errors and can thus parametrise unresolved subgrid-scale processes related to the sea-ice dynamics. For its trained forecast horizon, the neural network can reduce the forecast errors by more than $75\,\%$, averaged over all model variables. This error correction makes the forecast better than persistence for all model variables at correction time.

- A single big neural network can parametrise processes related to all model variables at the same time. The needed weighting parameters can be hereby spatially-shared and learned with a maximum likelihood approach. A Laplace likelihood improves the extracted features compared to a Gaussian likelihood and is better suited to parametrise the sea-ice dynamics.

- Convolutional neural networks with a U-Net architecture can represent important processes for sea-ice dynamics by extracting localised and anisotropic features from multiple scales. For sea-ice models defined on a triangular or unstructured grid, such scalable convolutional neural networks can be applied for feature extraction by mapping the input data into a Cartesian space that has a higher resolution than the original space. The finer Cartesian space keeps hereby correlations from the input data intact and enables the network to extract better features related to subgrid-scale processes.

- Because forecast errors in the sea-ice dynamics are likely linked to errors of the forecast model attractor, we have to apply the model error correction as a post-processing step and to input into the neural network the initial and forecasted state. This way, the neural network has access to the model dynamics and can correct them. Consequently, the dynamics of the forecast model variables are the most important predictors in a model error correction for sea-ice dynamics.

- Although only trained for correction at the first update step, applying the error correction together with the forecast model improves the forecast, tested up to one hour. The accumulation of uncorrected errors results into a distribution shift in the forecast errors, making the error correction less efficient for longer forecast horizons. Online training or techniques borrowed from offline reinforcement learning would be needed to remedy this distribution shift.

- The deterministic model error correction leads to an improved representation of the fracturing processes. Nevertheless, the unresolved subgrid-scale in the sea-ice dynamics can have seemingly stochastic effects on the resolved scales. These stochastic effects can result in wrongly positioned strains and fracturing processes for a deterministic error correction. To properly parametrise such effects, we would need generative neural networks.

## Appendix A: The regional sea-ice model with a Maxwell-elasto-brittle rheology

In the following paragraphs, we will describe the most important properties of the regional sea-ice model used in this study. For a more technical presentation of the model, we refer the reader to Dansereau et al. (2016, 2017). Our chosen model parameters are given in Table A1.

**Table A1.** The parameters for the regional sea-ice model that depicts the sea-ice dynamics (Dansereau et al., 2016, 2017) used in this study.

| Parameter | | Values |
|---|---|---|
| Poisson s ratio | $\nu$ | 0.3 |
| Internal friction coefficient | $\mu$ | 0.7 |
| Ice density | $\rho$ | $900\ \mathrm{kg\,m^{-3}}$ |
| Velocity of the elastic shear in ice | $c$ | $500\ \mathrm{m\,s^{-1}}$ |
| Undamaged elastic modulus | $E_0$ | $5.85 \times 10^8\ \mathrm{Pa}$ |
| Undamaged apparent viscosity | $\eta_0$ | $5.85 \times 10^{15}\ \mathrm{Pa\,s}$ |
| Undamaged relaxation time | $\lambda_0$ | $1 \times 10^7\ \mathrm{s}$ |
| Damage exponent | $\alpha$ | 4 |
| Characteristic time for damage | $t_\mathrm{d}$ | 16 s |
| Characteristic time for healing | $t_\mathrm{h}$ | $5 \times 10^5\ \mathrm{s}$ |
| Average grid resolution | $\Delta x$ | 4 km (high-res) |
| | | 8 km (low-res) |
| Integration time step | $\Delta t$ | 8 s (high-res) |
| | | 16 s (low-res) |
| Air drag coefficient | $C_{\mathrm{d_a}}$ | $1.5 \times 10^{-3}$ |
| Air density | $\rho_\mathrm{a}$ | $1.3\ \mathrm{kg\,m^{-3}}$ |
| Water drag coefficient | $C_{\mathrm{d_w}}$ | $5.5 \times 10^{-3}$ |
| Water density | $\rho_\mathrm{w}$ | $1 \times 10^3\ \mathrm{kg\,m^{-3}}$ |

The characteristic time in the damaging process is chosen to be no source of forecast error.

Compared to Arctic and pan-Arctic sea-ice models, like neXtSIM (Rampal et al., 2016; Ólason et al., 2022), this model is a regional standalone model that accounts exclusively for dynamical processes. Like most sea-ice models, it is two-dimensional

and based on a plane stress approximation. Nine variables constitute its prognostic state vector: sea-ice velocity in $x$- and $y$-direction, the three stress components, level of damage, cohesion, thickness, and concentration.

Atmospheric wind stress is the sole external mechanical forcing, whereas the ocean beneath the sea ice is assumed to be at rest. Given the small horizontal extent of our simulation domain (see Fig. 2), we also neglect the Coriolis force.

The Maxwell-elasto-brittle rheology from Dansereau et al. (2016) specifies the constitutive law of the model. It combines

elastic deformations, with an associated elastic modulus, and permanent deformations, with an associated apparent viscosity.

The ratio of the viscosity to the elastic modulus defines the rate at which stresses are dissipated into permanent deformations. Both variables are coupled to the level of damage: deformations are strictly elastic over undamaged ice and completely irreversible over fully-damaged ice. The level of damage propagates in space and time due to damaging and healing. Ice is damaged, and thus the level of damage increases, when and where the stresses are overcritical according to a Mohr-Coloumb damage criteria (Dansereau et al., 2016). This mechanism parametrises the role of brittle failure processes from the subgrid-scale onto the mechanical weakening of ice at the mesoscale. Reducing the level of damage, ice is healed at a constant rate, which parameterises the effect of subgrid-scale refreezing of cracks onto the mechanical strengthening of the ice. By neglecting thermodynamical sources and sinks in the model, cohesion, thickness, and area are solely driven by advection and diffusion processes; the prognostic variable for the thickness is hereby the thickness of the ice-covered portion of a grid cell, defined as the ratio between thickness and area. For the prognostic sea-ice thickness and area, a simple volume-conserving scheme is introduced to represent the mechanical redistribution of the ice thickness associated with ridging (Dansereau et al., 2017).

The model equations are discretised in time using a first-order, Eulerian implicit scheme. Due to the coupling of the mechanical parameters to the level of damage, the constitutive law is non-linear, and a semi-implicit fixed point scheme is used to iteratively solve the momentum, the constitutive, and the damage equations. Within a model integration time step, these three fields are updated first. Cohesion, thickness, and area are updated secondly, using the already updated fields of sea-ice velocity and damage.

The equations are discretised in space by a discontinues Galerkin scheme. The velocity and forcing components are defined by linear, first-order, continuous finite elements. All other variables and derived quantities like deformation and advection are characterised by constant, zeroth-order, discontinuous elements. The model is implemented in C++ and uses the *Rheolef* library (Version 6.7, Saramito, 2022).

Our virtual area spans $40 \, \mathrm{km} \times 200 \, \mathrm{km}$: a channel-like setup, which is nevertheless anisotropy-allowing. The model is based on a triangular grid with an average triangle size of $8 \, \mathrm{km}$ for the low-resolution forecasts. The grid for the high-resolution truth trajectories is a refined version of the low-resolution with a spacing of $4 \, \mathrm{km}$ (Fig. 2a).

If not otherwise stated, we initialise the simulations with undamaged sea ice, the velocity and stress components are set to zero, and the area and thickness to one. The cohesion is initialised with a random field, drawn from a uniform distribution between $5 \times 10^3 \, \mathrm{Pa}$ and $1 \times 10^4 \, \mathrm{Pa}$. We use von-Neumann boundary conditions on all four sides (Fig. 2c), with an inflow of undamaged sea ice (Fig. 2d) and a random cohesion, again between $5 \times 10^3 \, \mathrm{Pa}$ and $1 \times 10^4 \, \mathrm{Pa}$. The model configuration thus simulates a zoom into an (almost) undamaged region of sea ice, e.g. in the centre of the Arctic.

For the atmospheric wind forcing, we impose a sinusoidal velocity in $y$-direction and no velocity in $x$-direction, see also Eq. (3). Because of the anisotropy, the sea ice can nevertheless move in $x$-direction. Depending on its length scale and amplitude, the sinusoidal forcing generates cases of rapid transitions between undamaged and fully-damaged sea ice. As spin-up for the high-resolution simulations, the wind forcing is linearly increased over the course of the first simulation day. The parameters of the wind forcing are randomly drawn, as described in Sect. 4. The wind forcing is updated at each model integration time step ($8 \, \mathrm{s}$ for the high-resolution simulations and $16 \, \mathrm{s}$ for the low-resolution simulations).

## Appendix B: UNeXt architecture

To represent spatial correlations and anisotropic features, we use convolutional NNs (CNNs). We train a model error correction as subgrid-scale parametrisation (cf. Sect. 2.1), applied in a post-processing step, after the model forecast is generated. Since the Maxwell-elasto-brittle model is spatially discretised on a triangular space (cf. Sect. 2.2), we introduce a linear projection operator $\mathbf{P}$ (Sect. B1), interpolating from the triangular space to a Cartesian space that has a higher resolution, and where convolutions can easily be applied. After this projection, we apply a U-Net (Sect. B2) to extracts features in Cartesian space from the projected input fields. These features are then projected back into the triangular space with the pseudo-inverse of the projection operator $\mathbf{P}^{\dagger}$. There, linear functions combine pixel-wise (i.e., processing each element-defining grid point independently) the extracted features to the predicted residual, one for each variable. Each linear function is learned and shared across all grid points. The NN predicts the residuals for all nine forecast model variables at the same time with one shared U-Net. By sharing the U-Net across tasks, the NN has to learn patterns and features for error correction of all variables. To weight the nine different loss functions, we make use of a maximum likelihood approach (Sect. B3). This proposed pipeline (visualised in Fig. 4) can be seen as a baseline that enables a subgrid-scale parametrisation with deep learning for sea-ice dynamics, correcting all model variables at the same time.

### B1  The projection operator

For the Cartesian space, we chose a discretisation of $32 \times 128$ elements in the $x$- and $y$-directions, defined by constant, zeroth-order, Cartesian elements evenly distributed in the $40 \text{ km} \times 200 \text{ km}$ domain. As each Cartesian element has a resolution of $1.25 \text{ km} \times 1.5625 \text{ km}$, the Cartesian space has a higher resolution than the original triangular space ($\sim 8 \text{ km}$). Using such a super-resolution mitigates the loss of information caused by the projection. Furthermore, the NN can learn interactions between variables on a smaller-scale than used in the model, which helps to parametrise the subgrid-scale, as we will see in Sect. 5.1.

As projection operator $\mathcal{P}$, we use Lagrange interpolation from the original triangular elements to the Cartesian ones. For the velocity and forcing components, defined as first-order elements, this interpolation corresponds to a (linear) Barycentric interpolation and to a nearest neighbour interpolation for all other variables, defined as zeroth-order elements; $\mathcal{P}$ thus reduces to a linear operator, hereafter written $\boldsymbol{P}$. Because of the higher resolution, there are multiple Cartesian elements per triangular element, and the inverse of the operator does not exist as the linear system is over-determined. Consequently, in order to define the backward projection from the Cartesian space to the triangular space, we use the Moore–Penrose pseudo-inverse $\boldsymbol{P}^{\dagger}$. Since the rank of $\boldsymbol{P}$ is by construction equal to the dimension of the triangular space, i.e. its column number, the pseudo-inverse is in our case equal to $\boldsymbol{P}^{\dagger} = (\boldsymbol{P}^{\top}\boldsymbol{P})^{-1}\boldsymbol{P}^{\top}$, where $\boldsymbol{P}^{\top}$ corresponds to the transposed operator. Note, for coarse Cartesian spaces, the mapping from Cartesian space to triangular space can be non-surjective, meaning that not all triangular elements are covered by at least one Cartesian element: the pseudo-inverse is in this case rank deficient.

In the case of zeroth-order discontinuous Galerkin elements, the projection operator assigns to each Cartesian element one triangular element. The back-projection operator then corresponds to an averaging of the Cartesian elements into their assigned

triangular element. This averaging can be seen as a type of ensembling the information from smaller, normally unresolved, scales to larger, resolved scales. We have implemented this projection operator as a NN layer with fixed weights in PyTorch.

## B2  The U-Net feature extractor

We use CNNs in Cartesian space. The feature extractor should be able to extract multiscale features, and to represent rapid spatial transitions, which might occur only on finer scales. Consequently, we have selected a deep NN architecture with a U-like representation, a so-called U-Net (Ronneberger et al., 2015). The encoding part (on Fig. 4a, the left side of the U-Net) extracts information on multiple scales (here on two), by cascading downsampling steps. The decoding part (on Fig. 4a, the right side of the U-Net) refines coarse-scale information up and combines them with information from finer scales, and outputs 585 the extracted features. Consequently, the U-Net architecture can extract features at multiple scales, mapped onto the finest scale.

**Table B1.** Proposed baseline U-NeXt architecture based on ConvNeXt-like blocks. "Down" and "Up" correspond to downsampling and upsampling blocks, respectively. Counting with the weights of the linear functions in triangular space, the architecture has in total around $1.2 \times 10^6$ parameters.

| Stage | Operation | Params | $n_{in}$ | $n_{out}$ | $n_x$ | $n_y$ |
|-------|-----------|--------|----------|-----------|-------|-------|
| Input | ConvNeXt | 23 056 | 20 | 128 | 32 | 128 |
| Down 1 | Down | 295 424 | 128 | 256 | 16 | 64 |
| | ConvNeXt | 145 152 | 256 | 256 | 16 | 64 |
| | ConvNeXt | 145 152 | 256 | 256 | 16 | 64 |
| Bottleneck | ConvNeXt | 145 152 | 256 | 256 | 16 | 64 |
| Up 1 | Up | 295 552 | 256 | 128 | 32 | 128 |
| | ConvNeXt | 95 744 | 128 | 128 | 32 | 128 |
| | ConvNeXt | 39 808 | 128 | 128 | 32 | 128 |
| Output | ConvNeXt | 39 808 | 128 | 128 | 32 | 128 |
| | relu | – | 128 | 128 | 32 | 128 |

Our typical U-Net architecture consists of 3 different blocks: Residual blocks, mainly inspired by ConvNeXt blocks (Liu et al., 2022), a downsampling block, and an upsampling block. Our complete U-net architecture has in total approximately $1.2 \times 10^6$ trainable parameters and consists of five stages, cf. Table B1. The rectified linear unit (relu) in the output stage, 590 $h_{out} = \max(0, h_{out-1})$, introduces a discontinuity into the features, which can help the NN to represent sharp transitions in the level of damage. The input fields projected into the Cartesian space are treated as input channels for the input stage and include nine state variables and one forcing field for both input time steps, resulting to in-total 20 input channels. The architecture is quite thick, with 128 output channels, to extract features for all model variables at the same time.

### B2.1 The ConvNeXt blocks

In our standard configuration, the processing blocks are mainly inspired by ConvNeXt blocks (Liu et al., 2022). The output $\boldsymbol{h}_l = f_l(\boldsymbol{h}_{l-1}) + g_l(\boldsymbol{h}_{l-1})$ of the $l$-th block is calculated based on the output of the previous block $\boldsymbol{h}_{l-1}$ by adding a residual connection $f_l(\boldsymbol{h}_{l-1})$ to a branch connection $g_l(\boldsymbol{h}_{l-1})$, as depicted in Fig. 4b.

The residual connection is an identity function $f_l(\boldsymbol{h}_{l-1}) = \boldsymbol{h}_{l-1}$ if the number of its output channels $n_{out}$ equals the number of its input channels $n_{in}$. Otherwise, a convolution with a $1 \times 1$ kernel, called in the following $1 \times 1$ convolution, combines the $n_{in}$ input channels to $n_{out}$ output channels as a linear pixel-wise-shared function.

In the branch connection, a single convolutional layer with a $7 \times 7$ kernel is applied depth-wise (i.e. on each input channel independently) to extract information about neighbouring pixels; before applying the convolution, the fields are zero padded by three pixels on all four sides, such that the output of the layer has the same size as the input. The output of this spatial extraction layer is normalised by layer normalisation (Ba et al., 2016) across all channels and grid points. Compared to batch normalisation (Szegedy et al., 2014), layer normalisation is independent of the number of samples per batch and performs on par in this type of block (Liu et al., 2022).

Afterwards, a convolution layer with a $1 \times 1$ kernel mixes the normalised channel information up. If not otherwise depicted, the output of this intermediate layer gets activated by a Gaussian error linear unit (Gelu, Hendrycks and Gimpel, 2020). The last $1 \times 1$ convolution linearly combines the activated channels into $n_{out}$ channels. The output of this branch connection is scaled by learnable factors $\boldsymbol{\gamma}$, one for each output channel, and initialised with $\gamma_i = 1 \times 10^{-6}$. This type of scaling improves the convergence for deeper networks with residual layers (Bachlechner et al., 2020; De and Smith, 2020).

### B2.2 The down- and upsampling

For the downsampling operation, in the encoding part of the U-Net, we use a layer normalisation, followed by zero padding of one pixel on all four sides, and a convolution with a kernel size of $3 \times 3$ and stride of $2 \times 2$, similar to Liu et al. (2022). As this operation halves the data sizes in $x$- and $y$-direction, the number of channels is doubled in the convolution. By replacing max-pooling operations with a strided convolution, the downsampling operation becomes learnable (Springenberg et al., 2015).

For the upsampling operation, in the decoding part of the U-Net, we use a sequence of bilinear interpolation, which doubles the spatial resolution, layer normalisation, zero padding of one pixel on all four sides, and a convolution with a $3 \times 3$ kernel, which halves the number of channels. A bilinear interpolation followed by a convolution avoids unwanted checker-board effects (Odena et al., 2016), which can occur when using transposed convolutions for upsampling. Before further processing, the output of the upsampling block is concatenated with the output of the encoding part at the same spatial resolution, indicated by the blue dotted line in Fig. 4a.

### B3 Learning via maximum likelihood

In our NN architecture, we want to predict a model error correction for all nine model variables at the same time, which causes nine different loss function terms, like nine different mean-squared errors or mean absolute errors (MAEs). As each of these

variables has its own error magnitude, variability, and issues to correct, we have to weight the loss functions against each other with parameters $\boldsymbol{\lambda} \in \mathbb{R}^9$, $\mathcal{L}_{total} = \sum_{i=1}^{9} \lambda_i \mathcal{L}_i$. To tune these parameters, we use a maximum likelihood approach, which relates the weighting parameters to the uncertainty of the nine different model variables (Cipolla et al., 2018).

In the maximum likelihood approach, a conditional probability distribution $p(\boldsymbol{\Delta x} \mid \boldsymbol{x}, \boldsymbol{\theta})$ parametrised by $\boldsymbol{\theta}$ is assumed to approximate the true, but unknown, data generating conditional probability distribution of the residuals $\boldsymbol{\Delta x}$ given the input $\boldsymbol{x}$ – note that for conciseness the initial state $\boldsymbol{x}^{\text{in}}$ and the forecasted state $\boldsymbol{x}^{\text{f}}$ have here been gathered in a single input vector $\boldsymbol{x}$. The parameters of this probability distribution are optimised such that the negative log-likelihood of the observed residuals $\boldsymbol{\Delta x}$ given the input $\boldsymbol{x}$ and parameters is minimised,

$$\boldsymbol{\theta}^{\star} \triangleq \underset{\boldsymbol{\theta}}{\operatorname{argmin}}[-\ln p(\boldsymbol{\Delta x} \mid \boldsymbol{x}, \boldsymbol{\theta})].$$

The log-likelihood factorises hereby as sum over multiple dimensions like the samples or variables.

We treat the output of our NN $f(\boldsymbol{x}, \boldsymbol{\phi})$ with its weights $\boldsymbol{\phi}$ as the median of a univariate approximated Laplace distribution. From the perspective of the NN, the negative log-likelihood is thus a weighted MAE loss function. As all data points are equally weighted, a Laplace distribution results into a more robust estimation against outliers than a Gaussian distribution. Contrary to the median predicted as field, we use a single scale parameter per variable, $b_i$, shared across all grid points. We optimise the nine scale parameters together with the NN by minimising the negative log-likelihood, averaged over $B$ data pairs $(\boldsymbol{x}_j, \boldsymbol{\Delta x}_j)$. As we utilise a variant of stochastic gradient descent for optimisation, the data pairs are drawn from the training dataset $\mathcal{D}$ at each iteration. Before summing all nine loss terms up, we average the negative log-likelihood per-variable across all grid points (here simplified denoted as average across $M$ grid points) as the velocity components have fewer data points than all other variables, caused by their spatial discretisation (c.f. Sect. 2.2),

$$\mathcal{L}_{total} = \frac{1}{BM} \sum_{i=1}^{9} \sum_{j=1}^{B} \sum_{k=1}^{M} \frac{1}{b_i} |\Delta x_{i,j,k} - f_{i,j,k}(\boldsymbol{x}, \boldsymbol{\phi})| + \ln(2b_i).$$

The factor in front of the absolute error, $\lambda_i = \frac{1}{b_i}$, is the weighting factor; the MAE can be recovered by setting $b_i = 1$ as constant. The additional term, $\ln(2b_i)$, origins from the normalisation of the Laplace distribution, is independent of the errors, and counteracts a too small $b_i$. This approach optimises $b_i$ to match the expected MAE (Norton, 1984) in the training dataset and can be seen as an uncertainty estimate, e.g. recently used in Cipolla et al. (2018); Rybkin et al. (2020). Compared to using a fixed climatological value, this approach adaptively weights the loss, depending on the error of the NN for the different variables. This adaptive weighting marginally improves the training of the NN, as shown in Sect. D3. Since we learn the scale parameters purely from data, this approach can be seen as type II maximum likelihood or empirical Bayes approach (Murphy, 2012).

## Appendix C: Screening of architectures

As our NN architecture accommodates multiple decisions, we will here explore how they influence the results on the test dataset. We show what would happen if we would use other CNN architectures (Table C1). Detailed NN configurations can be

found in Appendix C1. We have selected the parameters of the U-Net architecture after a randomised hyperparameter screening in the validation dataset with 200 different network configurations per architecture, like for the U-NeXt architecture.

**Table C1.** Normalised MAE on the test dataset for different NN architectures, shown are average and standard deviation across ten training seeds. Reported are the errors for the velocity component in $y$-direction $v$, for the stress component $\sigma_{yy}$, the damage $d$, and the area $A$. The mean $\overline{\Sigma}$ is the error averaged over all nine model variables, including the non-shown ones. A score of one would correspond to the performance of the raw forecast in the training dataset. Models in the first block are to baseline methods; models in the second block are the multiscale convolutional models and in the third block the U-Nets; models in the fourth block are like the "Conv (x1)" architecture, but trained for each variable independently and with the specified number of channels in the hidden layer. Bold scores correspond to best scores in that column.

| Name | Params ($\times 10^6$) | $v$ | $\sigma_{yy}$ | d | A | $\overline{\Sigma}$ |
|---|---|---|---|---|---|---|
| Persistence | - | $0.37 \pm 0.00$ | $0.29 \pm 0.00$ | $0.60 \pm 0.00$ | $2.37 \pm 0.00$ | $0.79 \pm 0.00$ |
| Raw forecast | - | $1.14 \pm 0.00$ | $0.91 \pm 0.00$ | $1.09 \pm 0.00$ | $0.94 \pm 0.00$ | $1.03 \pm 0.00$ |
| Bias-corrected forecast | - | $1.14 \pm 0.00$ | $0.90 \pm 0.00$ | $1.09 \pm 0.00$ | $0.94 \pm 0.00$ | $1.02 \pm 0.00$ |
| Conv ($\times 1$) | 0.05 | $0.36 \pm 0.02$ | $0.27 \pm 0.01$ | $0.48 \pm 0.01$ | $0.63 \pm 0.01$ | $0.36 \pm 0.01$ |
| Conv ($\times 5$) | 0.29 | $0.33 \pm 0.01$ | $0.24 \pm 0.01$ | $1.00 \pm 0.15$ | $0.35 \pm 0.01$ | $0.35 \pm 0.02$ |
| U-Net | 3.7 | $0.35 \pm 0.00$ | $0.24 \pm 0.00$ | $0.41 \pm 0.00$ | $\mathbf{0.33 \pm 0.00}$ | $0.28 \pm 0.00$ |
| U-NeXt | 1.2 | $\mathbf{0.23 \pm 0.00}$ | $\mathbf{0.17 \pm 0.00}$ | $\mathbf{0.38 \pm 0.01}$ | $\mathbf{0.33 \pm 0.00}$ | $\mathbf{0.24 \pm 0.00}$ |
| Independent Conv ($\times 1$, 16) | 0.05 | $0.35 \pm 0.01$ | $0.25 \pm 0.02$ | $0.46 \pm 0.01$ | $0.60 \pm 0.01$ | $0.35 \pm 0.00$ |
| Independent Conv ($\times 1$, 128) | 0.42 | $0.47 \pm 0.04$ | $0.36 \pm 0.05$ | $0.96 \pm 0.22$ | $0.70 \pm 0.05$ | $0.47 \pm 0.05$ |

The simplest approach to correct the model forecast is to estimate a global bias, one for each variable, in the training dataset and to correct the forecast by this constant. As we measure the MAE, we take as bias the median error instead of the mean error in the training dataset. Correcting the bias has almost no impact on the scores, and, consequently, the model error is dominated by dynamical errors.

As a next level of complexity, we introduce a shallow CNN architecture with one layer as feature extractor. Using dilation in the convolutional kernel, this layer can extract shallow multiscale spatial information for each grid point. This shallow architecture with only around $5 \times 10^4$ parameters constantly improves the forecast by around 65 % in average. Introducing a hierarchy of five convolutional layers in the "Conv ($\times 5$)" architecture increases the number of parameters to $2.9 \times 10^5$. However, the averaged metric is only marginally better than for the shallow CNN. Its multiscale capacity is limited, and the NN cannot scale with the depth of the network to extract more information. Caused by their limited capacity, the NNs have problems to converge, which harms the performance, like the damage in the case of "Conv ($\times 5$)". A shallow network with a single convolutional layer can be nevertheless an option to obtain a small and fast NN for a subgrid-scale parametrisation. Such a fast NN can be helpful if the additional latency time impacts its application in a sea-ice model.

An approach to extract multiscale information is to use a U-Net architecture that extracts and combines information from different levels of coarsened resolution. To make the approaches comparable, we use almost the same configuration as specified in Sect. B2 and Table B1, except that we replace the ConvNeXt blocks by simple convolutional layers with a kernel size of $3 \times 3$, followed by batch normalisation (Szegedy et al., 2014), and a Gaussian error linear unit activation function. Using such a U-Net decreases the forecast errors by more than $20\%$ compared to the basic CNN. Although the improvement is for some variables only small compared to the simpler networks, the U-Net improves the balance between different variables. Consequently, the metric for the U-Net is always better than for persistence, showing its capacity and potential to extract multiscale information.

Replacing the classical convolutional layers with ConvNeXt blocks as described in Sect. B2 gives an additional improvement in the performance of the NNs. Although the number of parameters for the U-NeXt is only a third of the number for the U-Net, the ConvNeXt blocks reduce the forecast errors by additional $14\%$. The blocks reduce hereby especially the errors in the damage and area. The ConvNeXt blocks are able to extract more information from existing data than convolutional layers. Because the U-NeXt is the best performing method also in the validation dataset, we use this architecture throughout the manuscript.

Training "Conv ($\times 1$)" for each model variable independently has only a marginal impact on the scores, although their latencies are much larger than for the shared NN. The convergence issues in the "Conv ($\times 5$)" and "Independent Conv ($\times 1$, 128)" architecture indicate that the improvement of the bigger neural networks is not only related to an increased number of parameters but also because their multiscale layout. These results signify that training one big neural network for model error correction of all model variables allows us to use bigger networks, which improves their general performance.

## C1 Neural network configurations

By mapping from triangular space into high-resolution Cartesian space, several Cartesian elements are caught in one triangular element. Consequently, a simple convolutional layer would have problems to extract information across multiple scales. To circumvent such problems, we apply in the case of the naively-stacked convolutional layers two convolutional layers at the same time – one local filter with a $3 \times 3$ kernel, and one larger-scale filter with $3 \times 3$ kernel and a dilation of $6 \times 7$, such that the filter sees the next triangular element – we call such a layer "MultiConv". Using zero padding, we keep the output of the layers the same. The output of both convolutional layers is averaged to get a single output. As usual for CNNs, we use batch normalisation instead of layer normalisation. We keep Gelu as activation function, as for the ConvNeXt blocks, except for the last layer, where we use relu. The "Conv ($\times 1$)" uses a single block (Table C2), whereas the "Conv ($\times 5$) stacks five blocks (Table C3) with increasing number of feature channels.

Our baseline "U-Net" (Ronneberger et al., 2015) has classical convolutional blocks instead of ConvNeXt blocks. Like in the case for the U-NeXt, we have optimised the hyperparameters for this U-Net with a random hyperparameter sweep over 200 different network configuration. Similarly to the U-Next (c.f. Table B1), the here-used configuration is based on one level of depth, where the fields are downsampled in the encoder and upsampled in the decoder part. Our convolutional blocks have one convolutional layer with a $3 \times 3$ kernel with zero padding, batch normalisation, and Gelu as activation layer. For the *downsampling* in the encoder, we sequentially use: one convolutional layer with a $3 \times 3$ kernel, stride of $2 \times 2$, and zero padding;

**Table C2.** "Conv ($\times 1$)" based on a single multiscale convolutional layer with "Batch norm" as batch normalisation.

| Operation | Params | $n_{in}$ | $n_{out}$ | $n_x$ | $n_y$ |
|---|---|---|---|---|---|
| MultiConv | 46 208 | 20 | 128 | 32 | 128 |
| Batch norm | 256 | 128 | 128 | 32 | 128 |
| relu | - | 128 | 128 | 32 | 128 |

**Table C3.** "Conv ($\times 5$)" based on five stages with multiscale convolutional layer with "Batch norm" as batch normalisation.

| Stage | Operation | Params | $n_{in}$ | $n_{out}$ | $n_x$ | $n_y$ |
|---|---|---|---|---|---|---|
| Stage 1 | MultiConv | 11 552 | 20 | 32 | 32 | 128 |
| | Batch norm | 64 | 32 | 32 | 32 | 128 |
| | Gelu | - | 32 | 32 | 32 | 128 |
| Stage 2 | MultiConv | 18 464 | 32 | 32 | 32 | 128 |
| | Batch norm | 64 | 32 | 32 | 32 | 128 |
| | Gelu | - | 32 | 32 | 32 | 128 |
| Stage 3 | MultiConv | 36 928 | 32 | 64 | 32 | 128 |
| | Batch norm | 128 | 64 | 64 | 32 | 128 |
| | Gelu | - | 64 | 64 | 32 | 128 |
| Stage 4 | MultiConv | 73 792 | 64 | 64 | 32 | 128 |
| | Batch norm | 128 | 64 | 64 | 32 | 128 |
| | Gelu | - | 64 | 64 | 32 | 128 |
| Stage 5 | MultiConv | 147 584 | 64 | 128 | 32 | 128 |
| | Batch norm | 256 | 128 | 128 | 32 | 128 |
| | relu | - | 128 | 128 | 32 | 128 |

batch normalisation, and Gelu as activation layer. For the *upsampling* in the decoder, we sequentially use: bilinear interpolation; one convolutional layer with a $3 \times 3$ kernel, stride of $2 \times 2$, and zero padding; batch normalisation, and Gelu as activation layer. The encoder and decoder are connected via a bottleneck and a shortcut connection at the non-scaled level. Because we use several convolutional layers which extract spatial information and mix the channel at the same time, the network has much more parameters than the U-NeXt (Table C4).

**Table C4.** "U-Net" with normal convolutional blocks, where down and up correspond to downsampling and upsampling operations, respectively. Each convolutional block is a sequence of a convolutional layer, batch normalisation and a Gelu activation function, which is skipped in the last Output Conv block.

| Stage | Operation | Params | $n_{in}$ | $n_{out}$ | $n_x$ | $n_y$ |
|---|---|---|---|---|---|---|
| Input | Conv | 23 296 | 20 | 128 | 32 | 128 |
| Down 1 | Down | 295 424 | 128 | 256 | 16 | 64 |
| | Conv | 590 336 | 256 | 256 | 16 | 64 |
| | Conv | 590 336 | 256 | 256 | 16 | 64 |
| | Conv | 590 336 | 256 | 256 | 16 | 64 |
| Bottleneck | Conv | 590 336 | 256 | 256 | 16 | 64 |
| Up 1 | Up | 295 168 | 256 | 128 | 32 | 128 |
| | Conv | 295 168 | 128 | 128 | 32 | 128 |
| | Conv | 147 712 | 128 | 128 | 32 | 128 |
| | Conv | 147 712 | 128 | 128 | 32 | 128 |
| Output | Conv | 147 712 | 128 | 128 | 32 | 128 |
| | relu | – | 128 | 128 | 32 | 128 |

## Appendix D: Additional results

In this section, we provide additional results, showing the influence of different choices in the training on the performance in the testing dataset.

### D1 Number of training samples

Large NNs have many parameters, in the case of the U-NeXt $1.2 \times 10^6$, and could fit functions with as many degrees-of-freedom. If the number of degrees-of-freedom in the training dataset is similar or even lower, the NN might perfectly fit and remember the training dataset. As there is noise and spurious correlations within the dataset, the NN would also learn these "features" and overfit towards the training dataset. In this overfitting case, the NN would fail to generalise and to give good predictions to data unseen during training.

In the following, we analyse the training behaviour of the NN and see what happens if we artificially train on a portion of data only (Fig. D1). For the validation dataset, over the course of the training, we show the negative log-likelihood (NLL) with a Laplace assumption and scaling parameters fixed to their climatological values. This loss is proportional to a mean absolute error (MAE) with a fixed weighting, where the weights are given by the inverse scaling parameters. Additionally, we analyse the weighted MAE in the testing dataset for the different NNs after training.

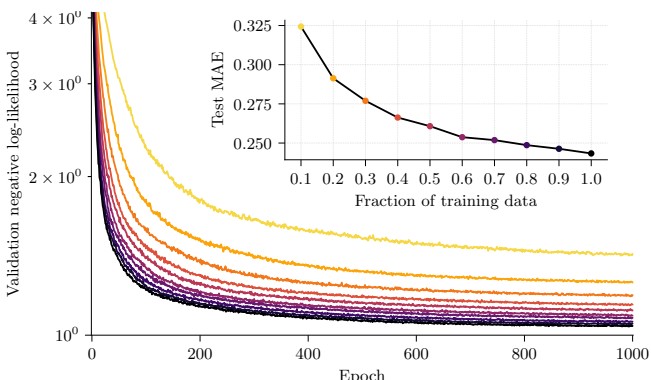

**Figure D1.** The negative log-likelihood for a Laplace distribution, proportional to the mean absolute error (MAE), with a fixed weighting in the validation dataset as function of epochs for different fractions of training data, the brighter the colour, the less training data is used. The smaller Figure shows the averaged MAE in the test dataset as function of the fraction of training data. A fraction of 1.0 corresponds to around $12.3 \times 10^6$ degrees-of-freedom in the training dataset.

For all fractions of training data, the validation NLL smoothly decreases over the course of training. Consequently, we see no overfitting, even with only $10\%$ training data. Additionally, the loss and the weighted MAE saturates as function of fraction in the training data: the gain training on more data is larger for small sample sizes than for larger sample sizes. Such a logarithmic data scaling behaviour is expected and can be even observed for very large language models (Kaplan et al., 2020). Given this scaling, we would need much more data to scale the performance further.

We can now wonder why the NN trained on only $10\%$ training data shows no overfitting, even though the NN has as roughly much degrees-of-freedom as the training data. We attribute this behaviour to our projection step or to fitting one NN on all model variables. The most NN parameters are stored within the feature extractor in Cartesian space. Caused by the back-projection step mapping from Cartesian to triangular space, the features are averaged across grid points. Consequently, we hypothesise that the true number of NN parameters as seen from the triangular space is much smaller than $1.2 \times 10^6$. Furthermore, fitting one NN on all model variables acts as kind of regularisation. To gain a balanced performance across all variables, the NN has to extract features shared across variables. Seen for a single variable alone, feature sharing reduces the capacity of the NN. Additionally, by sharing, the NN is encouraged to extract more generalisable features. In general, this would mean that training a single big NN for multiple variables could really improve data-driven forecasting, even for only a limited amount of data.

## D2 Lead time between two correction steps

One of our fixed parameters is the lead time for which the NN is trained and applied for forecasting. In the following, we will shortly discuss the impact of the lead time between two correction steps (hereafter correction time) on the forecasting performance, again measured by the normalised RMSE.

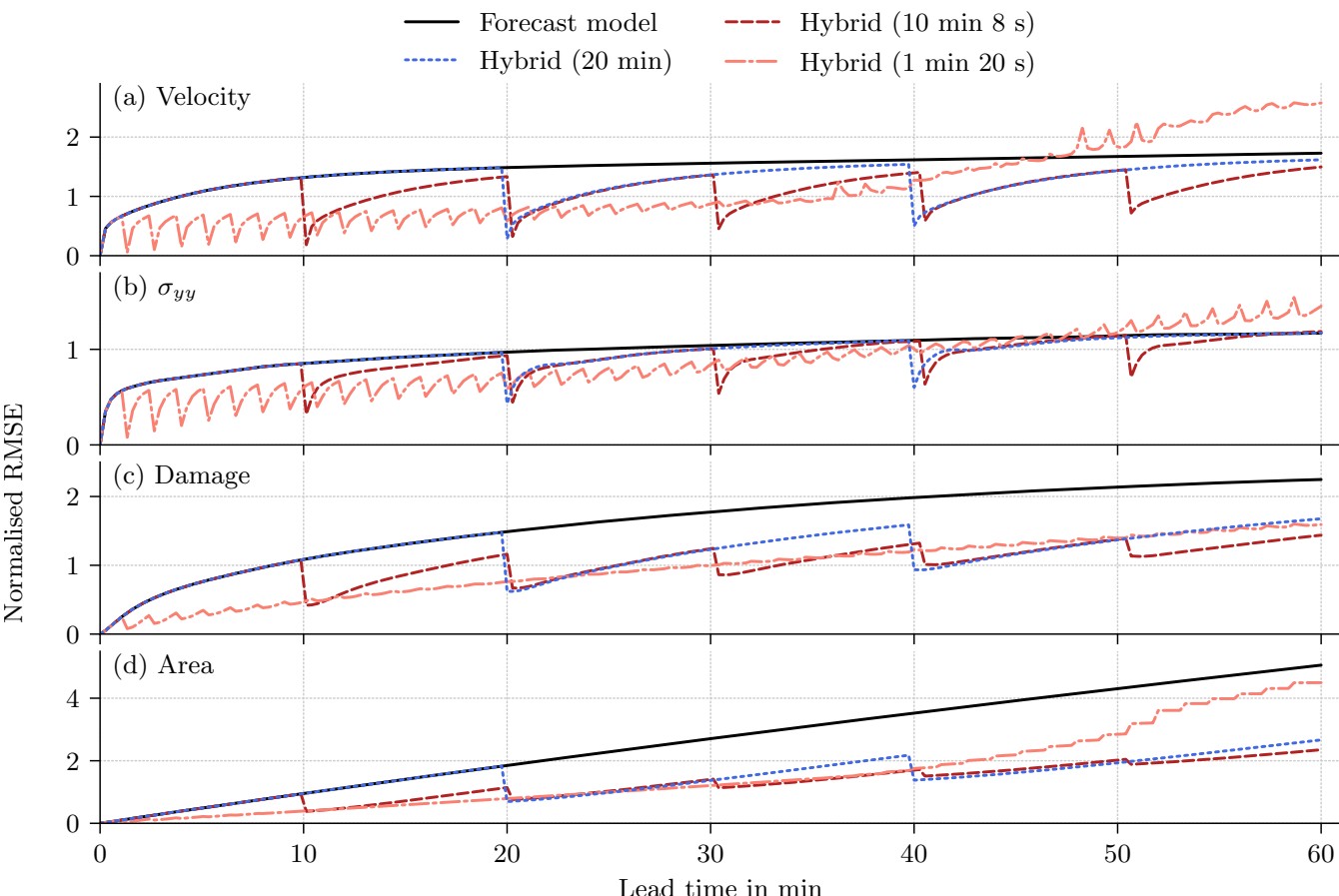

**Figure D2.** Normalised RMSE for (a) the velocity in $y$-direction, (b) the divergent stress in $y$-direction, (c) the damage, and (d) the sea-ice area as function of lead time on the test dataset, normalised by the expected RMSE on the training dataset for a lead time of $10\,\mathrm{min}$ and $8\,\mathrm{s}$. In the hybrid models, the forecast is corrected after each specified lead time in brackets.

Decreasing the correction time decreases how long the trajectory freely drifts towards the attractor of the sea-ice model. Model errors can be additionally earlier corrected, before they have a too large impact on the forecast. Consequently, we would expect the shorter the correction time, the better the forecasting performance. However, in our case, the forecasting performance is worse for a lead time of $80\,\mathrm{s}$ than for $10\,\mathrm{min}$ and $8\,\mathrm{s}$. Decreasing the correction time also increases the number of correction steps in a given time window. The more correction steps, the more the error distribution can shift. We have already identified the distribution shift as one of the main challenges towards the application of such model error corrections for forecasting. For a decreased correction time, the impact of the distribution shift outweighs the positive impact of earlier model error corrections. This results into a negative model error correction impact after a forecasting lead time of $60\,\mathrm{min}$, if the correction time is decreased.

For an increased correction time of $20\,\mathrm{min}$, we get a slightly improved performance at correction times compared to the shorter correction time of $10\,\mathrm{min}$ and $8\,\mathrm{s}$. Their performance is however generally comparable. Therefore, these results clearly point out again the negative impact of the distribution shift on the forecasting performance.

## D3 Loss functions

Optimising the Laplace log-likelihood corresponds to minimising the MAE, whereas an optimisation of a Gaussian log-likelihood minimises the mean-squared error. Thus, we report the averaged root-mean-squared error (RMSE) and MAE over all variables to measure the influence of the loss function on the performance of the NNs (Table D1). As the RMSE and MAE are normalised by their climatological values in the training dataset, the weighting between the model variables is fixed to their climatological values, favouring fitting networks with fixed climatological weighting.

**Table D1.** The average RMSE and MAE, normalised by their expected climatology, on the test dataset for different training loss functions. The bold loss function is the selected loss functions, and bold scores are the best scores in a column.

| Name | RMSE | MAE |
| --- | --- | --- |
| Gaussian (fixed) | 0.34 | 0.30 |
| Gaussian (trained) | 0.33 | 0.29 |
| Laplace (fixed) | 0.33 | 0.25 |
| **Laplace (trained)** | **0.32** | **0.24** |

Compared to a Gaussian log-likelihood with trainable variance parameters, the Laplace log-likelihood as loss function improves not only the MAE by around $17\,\%$, but also the RMSE by around $3\,\%$. Despite the fixed weighting, fitting the uncertainty parameters together with the NN marginally improves these metrics in both cases. Using adaptive uncertainty parameters modulates the gradient during training, and the optimisation benefits from this adaption, resulting into the shown error decrease.

The loss function influences the output of the NN and the learned features before they are linearly combined to the output (Fig. D3). In the learned features, a Laplace log-likelihood increases the contrast between highly-activated, active, regions and passive regions in the background with a low activation value, Fig. D3, (a) and (b). Here, we define the contrast of a feature map as the ratio between its spatially-averaged standard deviation $\sigma$ to its spatially-averaged mean value $\mu$. For the Laplace log-likelihood, the median contrast (1.15) is higher than for the Gaussian log-likelihood (0.93), as can be seen in Fig. D3, (c). The distribution for the Laplace log-likelihood is additionally more balanced, meaning that less extreme values appear on both ends. We attribute these differences to the different behaviour of the loss function (Hodson, 2022). As the Gaussian log-likelihood is more sensitive to larger errors in the training dataset, the NN has to learn specialised feature maps for these cases. The Laplace log-likelihood leads to a higher contrast in the feature maps and to more balanced feature maps. Based on its increased contrast, we hypothesise that the Laplace log-likelihood results into better linearly separable feature maps. On their basis, the linear functions can more easily combine the features to predict sharper and more localised residuals, improving the performance of the NN.

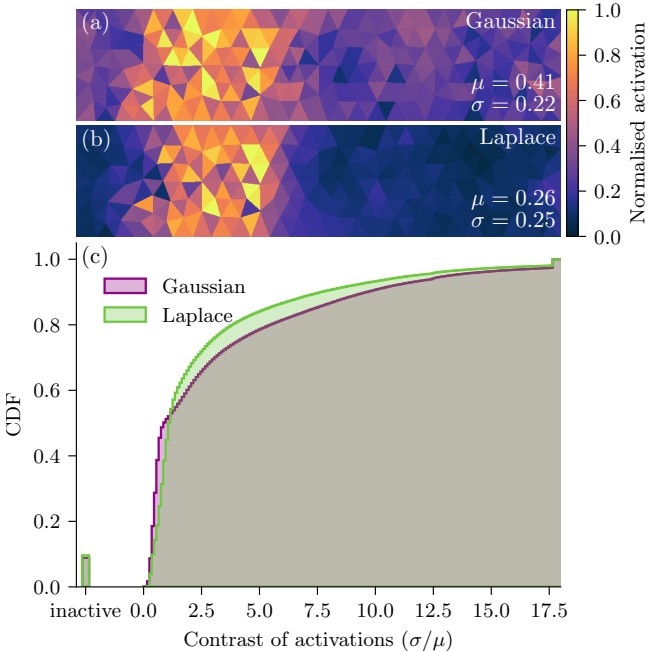

**Figure D3.** Two typical feature maps for a NN trained with either (a) a Gaussian log-likelihood or (b) a Laplace log-likelihood. For visualisation purpose, the feature maps are again normalised by their 99-th percentile, as in Fig. 5. The contrast (c) is estimated as the spatial standard deviation $\sigma$ divided by the spatial mean $\mu$ of each feature map, extracted from the test dataset. The number of inactivate features maps (=constant zero) is normalised by the same value as the CDF. A higher contrast indicates better linear separable features.

## D4    Activation functions

Another decision that we took in our architecture is to use the Gaussian error linear unit (Gelu) in the blocks and the rectified linear unit activation (relu) function as activation of the features, before they are projected back into triangular space and linearly combined. The Gelu activation function is recommended for use in a ConvNeXt block (Liu et al., 2022), but its performance seems to us to be on par with the relu activation function. Whereas the Gelu activation function is a smooth function inspired by dropout (Hendrycks and Gimpel, 2020), relu is a non-smooth function, which induces sparsity in the feature maps.

As similarly found in Liu et al. (2022), replacing the Gelu activation function with a relu activation function in the ConvNeXt blocks leads to almost the same results on the test dataset (Table D2). Furthermore, also the activation function for the extracted features at the end of the feature extractor has only a small influence. Even using no activation function at this position degrades the mean performance by 4 %, for some variables like the damage or area using no activation function leads to the best results. Because the Gelu activation function is state-of-the-art in many deep learning tasks and recommended Liu et al. (2022), we use the Gelu within the ConvNeXt blocks.

In the following, we show feature maps for different activation functions at the feature output of the U-Net as qualitative measure (Fig. D4). Using no activation function (Fig. D4, a), extracts continuous features. These features roughly follow a

**Table D2.** Normalised MAE on the test dataset for different activation functions in the ConvNeXt blocks and as feature activation (w/o: no activation function). The error components are estimated as in Table 2 and the same acronyms are used. The bold combination shows the selected activation functions, and bold scores are the best scores in a column.

| Activation | $v$ | $\sigma_{yy}$ | d | A | $\overline{\Sigma}$ |
|---|---|---|---|---|---|
| relu & relu | 0.24 | 0.17 | **0.37** | 0.34 | **0.24** |
| Gelu & Gelu | 0.23 | 0.17 | 0.39 | **0.33** | **0.24** |
| Gelu & w/o | **0.22** | **0.16** | 0.42 | 0.34 | 0.25 |
| **Gelu & relu** | 0.23 | 0.17 | 0.38 | **0.33** | **0.24** |

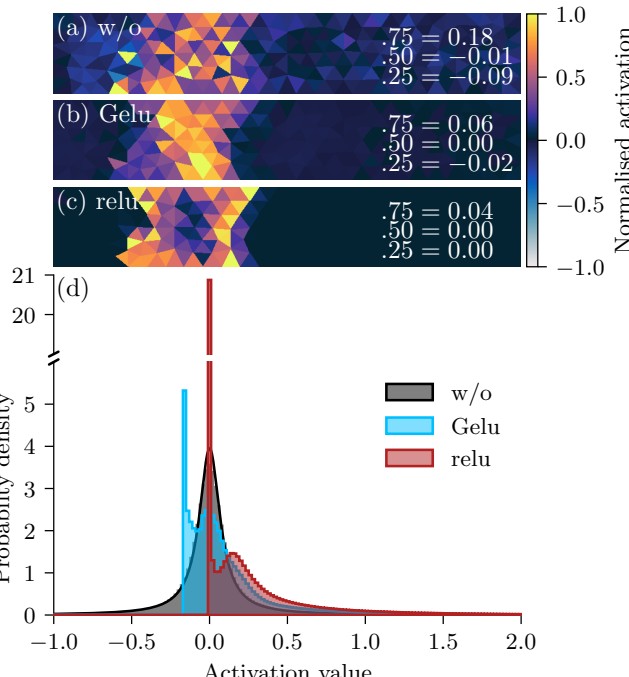

**Figure D4.** Snapshot of typical feature maps for (a) no feature activation (w/o), (b) the Gaussian error linear unit (Gelu), or (c) the rectified linear unit (relu). For visualisation purpose, the feature maps are normalised by their 99-th percentile. The numbers indicate the percentiles of the normalised feature maps. The histogram (d) represents the unnormalised feature activation values over the whole test dataset. As the histogram for the relu activation function have a large spike at 0 in (d), the y-axis is broken.

Cauchy distribution around 0 without enforcing sparsity (Fig. D4, d, the black contour line is the Cauchy distribution fitted via maximum likelihood). Caused by its weighting with the Gaussian error function, Gelu squashes the negative values of the activation values together, leading to a peak of the values around zero. Nevertheless, only few values are truly zero, as GELU do not set values explicitly to zero. In contrast, using relu enforces sparsity, and the NN can extract localised and "patchified" features (Fig. D4, c). Consequently, the relu activation generates many deactivated pixels. Although the performance of the

relu activation is the same as for the Gelu activation, we hypothesize that sparse features can improve the representation of subgrid-scale processes, as sea ice has a discrete character, especially at the marginal ice zone or around leads. Therefore, we use for our experiments the Gelu activation function as activation in the ConvNeXt blocks and the relu activation function as feature activation, before the features are linearly combined.

### D5 Permutation feature importance for variable groups

Using the permutation feature importance, we have analysed that all model variables are very sensitive to their own dynamics as predictor. Nevertheless, by permuting single predictors independently, we only destroy information contained in this predictor. As other variables might hold similar information, e.g., for the sea-ice area and thickness, the inter-variable importance is likely to be underestimated, and the permutations can lead to unphysical instances. To see the effect of the correlations on the importance, we permute different variable groups and estimate their importance on the nine output variables.

For the sea-ice velocities, their dynamics are clearly the predictors with the biggest impact. However, the absolute values of sea-ice area and thickness have combined a small but considerable impact on the velocity in $y$-direction, probably explainable by their coupling via momentum equation.

The stress components and damage are highly sensitive to their own dynamics if only a single variable is shuffled, as shown for the reference feature importance; however, they are insensitive if the stress components and damage are shuffled as a group. For their correction, the NN seems to rely on features that extract relative combinations of these variables. Shuffling a single variable then creates unphysical instances, which destroys such features, whereas they are kept intact when the stress components and the damage are shuffled together. The same feature importance as for the reference is reached if the velocities and the stress variables are shuffled together. Here, the dynamics are as important as the absolute values. Because the area and thickness have no influence, also the errors of the stress components and damage are driven by the dynamical variables, as in our sea-ice model.

For the area and thickness, if their dynamics are shuffled alone, their importance is higher than shuffling their dynamics at the same time. Additionally, similar differences can be observed, if the stress components are shuffled combined with or without damage. Again, we attribute this to the naturally high correlation in some variables, which leads to unphysical instances, skewing the permutation feature importance. The importance of having physically consistent sample instances manifests one of the downsides of the permutation feature importance for correlated input variables. Nevertheless, this importance shows also that the NN takes groups of input variables and their correlations into account, which could explain the efficiency of the NN.

### D6 Forecasting with differences as network input

For forecasting with the model error correction, we only show results for the NNs with the initial conditions and the forecasts as input, although the NNs with initial conditions and the difference between forecast and initial conditions performs better in the testing dataset. Here, we will shortly discuss the forecasting results of these latter NNs with the initial conditions and the differences as input (Table D4).

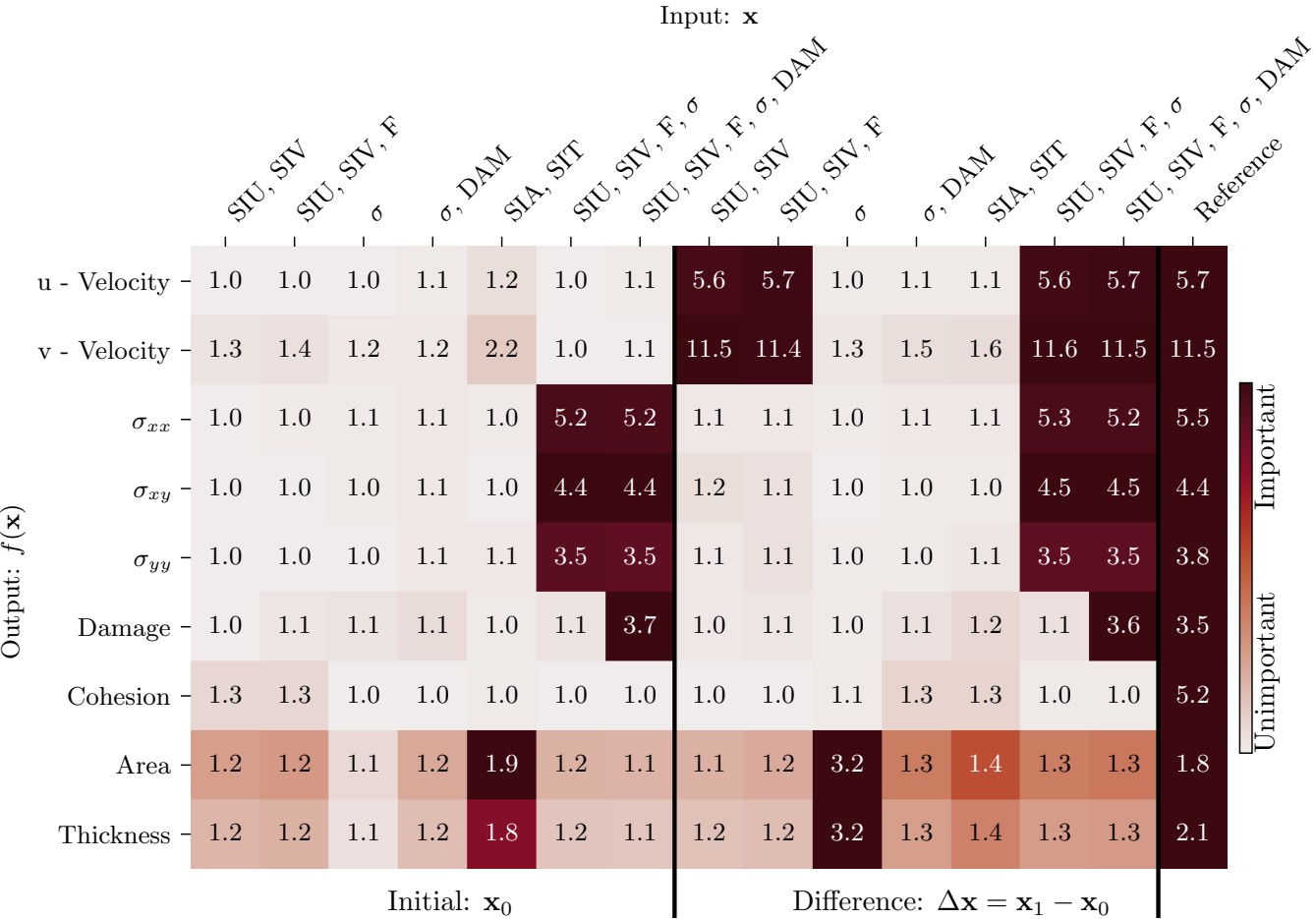

**Table D3.** Permutation feature importance of different variable groups. The colouring is the same as in Table 6 of the original manuscript. $SIU$ stands for velocity in $x$-direction, $SIV$ for velocity in $y$-direction, $F$ for wind forcing, $\sigma$ for all stress variables, $DAM$ for damage, $SIA$ for sea-ice area, and $SIT$ for sea-ice thickness. The reference is the permutation feature importance of the dynamics for a specific variable.

The dynamics are explicitly represented as difference between the forecast and initial conditions. On the one hand, this helps
the NN to extract more information from the dynamics than for the "Initial + Forecast" experiment (see also Table 4). On the other hand, this explicit representation introduces an assumption that the dynamics are additive to the initial conditions. In some sense, the NN can overfit towards the use of the dynamics for a model error correction. Caused by this sort of overfitting, the hybrid model performs worse for the velocity, the stress, and the damage than the hybrid model that uses the raw forecast, but their differences generally remain small. As the hybrid model with the initial conditions and the forecast as input has fewer
assumptions, we present its results with greater detail in Sect. 5.3.

**Table D4.** Normalised RMSE on the test dataset for a lead time of 60 min. The last update in the hybrid models was at a lead time of 50 min and 40 s. The errors are normalised by the expected standard deviation for a lead time of 60 min on the training dataset. The two bottom models correspond to the two hybrid models with different input variables. The symbolic representation of the variables has the same meaning as in Table 6.

| Name | $v$ | $\sigma_{yy}$ | d | A | $\overline{\Sigma}$ |
|------|-----|---------------|---|---|---------------------|
| Persistence | **1.13** | **0.81** | 0.83 | 2.58 | 1.19 |
| Sea-ice model | 1.34 | 0.93 | 1.06 | 0.98 | 1.06 |
| "Initial + Forecast" | 1.16 | 0.95 | **0.68** | 0.46 | **0.81** |
| "Initial + Difference" | 1.20 | 1.00 | 0.71 | **0.41** | 0.82 |

*Code and data availability.* The authors will provide access to the data and weights of the neural networks upon request. The source code for the experiments and the neural networks is publicly available under https://github.com/cerea-daml/hybrid_nn_meb_model. The regional sea-ice model source code will be made available upon request.

*Author contributions.* MB and AC initialised the scientific questions. TSF, CD, AF, and MB refined the scientific questions and prepared an analysis strategy. TSF, YC, and VD advanced the codebase of the regional sea-ice model. TSF performed the experiments. TSF, CD, AF, and MB analysed and discussed the results. TSF wrote and revised the manuscript with CD, AF, MB, YC, AC, and VD reviewing.

*Competing interests.* The authors declare that they have no conflict of interest.

*Acknowledgements.* The authors would like to thank Nils Hutter and an anonymous referee, by reviewing they helped to significantly improve the manuscript. The authors would additionally acknowledge other members of the project SASIP for delightful discussions along the track. Furthermore, the authors received support from the project SASIP (grant no. 353) funded by Schmidt Futures – a philanthropic initiative that seeks to improve societal outcomes through the development of emerging science and technologies. This work was additionally granted access to the HPC resources of IDRIS under the allocation 2021-AD011013069 and 2022-AD011013069R1 made by GENCI.

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
