# Peer review of "Deep learning subgrid-scale parametrisations for short-term forecasting of sea-ice dynamics with a Maxwell-elasto-brittle rheology"

_EGUsphere, 2022_

## Author Comment (AC1)

**Response to Referee 1**
**for "Deep learning of subgrid-scale parametrisations for short-term forecasting of sea-ice dynamics with a Maxwell-Elasto-Brittle rheology"**

Finn, T.S., Durand, C., Farchi, A., Bocquet, M.,
Chen, Y., Carrassi, A., Dansereau, V.

19th April 2023

**RC: Reviewer Comment**, AR: Author Response

**RC:** The authors present an idealised study of sea-ice fracture in a channel due to wind forcing, demonstrating that a neural network (NN) is able to significantly reduce errors of a lower-resolution version of the physical model with respect to a higher-resolution version of the same model for 10-minute forecasts. They conclude that the NN has learned the tendencies from the unresolved scales in the lower-resolution model, and can therefore be used to parameterise these unresolved scales.
I appreciate the originality of the work and the level of detailed analysis it provides. It fits well with current efforts in the community to use machine learning for parameterisation of unresolved scales in geophysical models. However, given the very idealised setup, I have some concerns about the wider applicability of the results. Below, I spell that out in comments which I would like the authors to address before publication:

**AR:** We thank referee 1 for the constructive feedback on our manuscript, especially with respect to the bias correction and a possible overfitting. In the following, we discuss the raised concerns and what we plan to change in our revised manuscript.

**RC:** There are a number of very strong idealisations and restrictions in the setup of this study: a) it is a so-called "perfect model" study, i.e. the performance of the lower-resolution model with/without NN corrections is assessed against a "truth" which is a simulation of that same model at higher resolution, without involving any observations or simulations from a different model; b) The forecast lead times considered are extremely short for most real-life weather and climate applications (only up to 1h); c) spatial domain is a simple rectangular channel; d) no treatment of sea-ice thermodynamics. Given these very strong idealisations and

**restrictions, one would hope for results that are a bit more convincing than the ones presented. I have concerns about whether the methods presented will be useful in a more realistic context, where each of the above assumptions will need to be relaxed. Can the authors please add some in-depth discussion (or even preliminary analysis) about what they think will happen if their methods are applied in a more realistic context?**

AR: Our study is designed to be a proof-of-concept. As sea ice imposes novel challenges for neural networks, it was previously unknown if model error corrections in this form are possible at all for sea-ice modelling. We think this study shows that there is indeed a huge potential for hybrid modelling of sea ice. Given the limited scope of this study, we have decided to apply such simplified settings, far from settings in operational forecasting or projections. For example, we have used the twin experiment setting to prove our points and to cheaply generate a known truth. If realistic model error corrections would be trained with twin experiments, the neural network would learn to emulate the fields from the higher resolution, so, instantiations of already known processes. Consequently, we believe that the true potential lies in the possible learning of model error corrections from observations, which is beyond the scope of our proof-of-concept. Furthermore, the model error correction is designed to correct model errors as soon as possible, before they have a too large impact on the forecast. This is why we concentrate on such short forecast lead times of up to one hour, although they might be far from operational settings. In further studies, with more realistic setups, we will investigate the impact of the model error corrections in longer forecast lead times.

To take this concern into account, we will strengthen the proof-of-concept character of the study in the introduction. Additionally, we will discuss more steps what might follow towards more generalised and realistic settings.

RC: **Figure 7 and the corresponding text makes me wonder how much of the error reduction achieved by the NN is actually due to correcting the bias (i.e. mean error) of the low-resolution simulation w.r.t. the high-resolution simulation. Can the authors please provide some analysis to quantify the contribution of bias to the overall errors, with and without the NN corrections? For instance, one could just decompose the mean squared errors shown in the manuscript into squared bias and variance of the errors. I am asking this because there is a range of other methods to treat biases (e.g. a-priori by tuning the model, and a-posteriori by subtracting them from the forecast before further analysis). These methods are often simpler than the machine-learning approach and are in wide use in the weather and climate community. Utilising a complex and costly machine-learning approach only pays off it is clearly superior to other available methods.**

AR: As shown in Appendix B, Table B1, we made tests where we simply correct the

constant model bias. The performance of this bias corrected model is almost the same as for the raw model without any correction. Most of the model error is hence temporally and spatially variable and cannot be corrected by a simple bias correction. Additionally, we made the tests with a rather small neural network, where only one multi-scale convolutional layer is used. We expect that even simpler methods, e.g., a linear regression, perform worse than this small neural network. Additionally, there are many possibilities with neural networks that we have not taken into account, e.g., generating stochastic parametrisations with correlations learned from data. Therefore, we believe that this study indicates the potential for hybrid modelling, which would be otherwise unachievable.

In the discussion part, we have tried to clarify the possibilities with neural networks. In the revised manuscript, we may further clarify some of these possibilities.

**RC:** **Following up on the previous comment, I would like the authors to comment on potential overfitting of the NN in their methods. If I did the maths correctly, there are about 4500 degrees of freedom in the lower-resolution physical model (9 variables times 500 grid points). As stated on line 197, the NN has 1.2 million trainable parameters. So one could argue the NN has orders of magnitude more degrees of freedom than features it is learning from or results it is predicting. I am not an expert on machine learning, but that strikes me as odd - could the authors please comment on that? I would also like to see some quantitative analysis on the risk of overfitting.**

AR: We agree that a single low-resolution field has only 2558 degrees-of-freedom (DOFs). Compared to this number, the number of parameters in the neural network ($1.2 \times 10^6$) seems to be very high. However, the training dataset has $2558 \times$ number of samples DOFs. In the end, this sums up to around $12.3 \times 10^6$ million DOFs, an order of magnitude larger than the parameters in the neural network. During training, the scores in the validation dataset have been smoothly improving without a sign of overfitting. Furthermore, we have made new experiments (Fig. 1), where we have only used a fraction of the data for training. Even in our most extreme case with only 10 % of the data (480 samples), the model is not overfitting with smoothly decreasing MSE and MAE, and we achieve a nice scaling of the performance with the number of samples. We attribute this behaviour to the projection into Cartesian space and to the strategy of learning a correction for all variables at the same time. Caused by the projection from Cartesian to triangular space, the information content of the extracted features is reduced. Additionally, learning for multiple outputs at the same time acts as a type of regularisation for a single output, the network has to find features that suits all outputs.

We will introduce a new subsection in the additional results part of the Appendix with the discussion of Fig. 1 that indicates no overfitting during the training of the neural networks.

**RC:** **Please revise the presentation of the methods, this is not sufficient in**

**some places, and difficult to follow in others. See technical comments.**

AR: Based on this raised concern and the other review, we have decided to rearrange the presentation of the methods. The sea-ice model and the neural network will be discussed in a hopefully easier language, explaining rather the reasoning behind the different components, instead of the technical details. The more technical explanations will be moved into an Appendix. We will replace Section 2 with a Section where we first introduce the model error correction problem from a mathematical standpoint (previously Section 3.1), then a shortened explanation of the sea-ice model and, finally, an introduction into the used twin experiments. Section 3 about the neural network will be more concise and will focus more on the reasoning behind the different components.

RC: **I am afraid I do not quite understand the motivation why a projection to a Cartesian grid is needed (Section 3.2). It seems to complicate the methods unnecessarily. Can the authors please clarify the motivation for doing this, and what the feasibility/implications would be of doing the analysis on the original triangular grid? Is this just a reflection of the fact that the standard machine-learning libraries for spatial analysis cannot deal with non-Cartesian grids?**

AR: Compared to more "classical" neural networks, so-called multilayer perceptrons, by construction, convolutional neural networks (CNNs) are biased towards localised features, motif extraction across all grid points, and a directional dependency. Additionally, the backend libraries e.g., TensorFlow and PyTorch, are optimised for image processing. CNNs are hence especially efficiently implemented for Cartesian spaces. Although there are different convolutional architectures better suited for unstructured grids, e.g., graph neural networks, they are usually more computationally heavy and more difficult to implement. Given the limited scope of our proof-of-concept, we have thus decided to make use of "normal" convolutional neural networks. Additionally, the variables are different at different positions on the triangles. Consequently, we interpolate from triangular space to a common Cartesian space where the features are extracted. Combined with projecting the features back into triangular space and linearly combining them therein, this architecture turns out to be very efficient and can act as a baseline approach for further studies.

In our new Section 3, we will explain our reasoning behind the feature extraction in Cartesian spaces more than in the former neural network Section.

RC: **Figure 1: Please specify which physical variable that is displayed (damage?). Could the authors find a more convincing "showcase" example? By visual inspection, it looks to me like there is still substantial errors in the "hybrid" field, which seems at odds with the claim of an 75 % error reduction. Please quantify the error reduction for the case shown.**

AR: There was a technical issue to correctly render the file on Copernicus' side. This should be fixed now, and the missing labels etc. should be there now. The shown damage fields are for a lead time of 60 minutes. Given your feedback, we have decided to change the snapshot of the sea-ice damage to a more representative case (Fig. 2). There, the improvement is 62 %, and the hybrid model is able to correctly represent the damaging processes such that not too much damage is produced as in the low-resolution forecast without correction.

RC: l. 35f. (and elsewhere): I am not sure what "wave-like" and "channel-like" - please be more precise.

AR: In the revised manuscript, we will be more specific and will avoid 'wave-like' and 'channel-like'.

RC: Line 76 & 89: a 10 minute (or even 1 hour) forecast is extremely short both for main-stream earth system models and real-world applications. Can you please comment on that and justify looking at these very short time scales?

AR: On the one hand, we want to correct the model error after each integration step, which would be in our case 16 seconds. On the other hand, the neural network is not perfect, and the more signal during training, the better. Furthermore, the neural network is trained without taking interactions with the sea-ice model into account. The missing interactions lead to a distribution shift during the application of the model error correction. Consequently, using a correction time of 10 minutes is already a compromise. Given the limited scope of this proof-of-concept and an already visible distribution shift (Table 8), we have restricted the forecast time to one hour.

RC: The introduction in ll. 80-92 already gives too much technical detail about the methods. This belongs elsewhere.

AR: We will reduce the amount of technical information in the introduction.

RC: In Figure 2 and the corresponding text, the authors need to help the reader to get a physical understanding of the situation that causes the ice to fracture. Please add arrows indicating the wind field, and refer to Equation (1). Please specify which direction is x and which is y. Also Figure 2: Please use other colours than black and red to indicate the two grids, otherwise it is difficult to see for color-blind people.

AR: We will add the forcing field with arrows for this specific case in Figure 2, as shown in Fig. 3. Additionally we will change the colour of the coarse grid in Figure 2 to a lighter blue tone, which should make the figure easier to read.

RC: Line 134: I do not know what "wave-like" means. Please be more precise, and provide the equation with the wind forcing at the earliest possible place in the text.

AR: We will introduce the equation for the wind forcing in the new shortened description of the regional sea-ice model and avoid the term 'wave-like'.

RC: **Figure 6: I much appreciate the sensitivity testing in Section 5.2, very good! However, I am puzzled by the very weak cross-variable coupling in the permutation feature importance. It seems contradictory to your claim that the NN has "learned the dynamics" of the physical model. For instance, for damage as an output variable, it seems that the NN only extracts information from the damage itself, all other input variables are unimportant! Could you please provide some more explanation/clarification/analysis on this?**

AR: The input variables are naturally correlated to each other, e.g., for the forcing and the velocities or the area and thickness. Therefore, by destroying the information of a single variable, almost the same information is still available to the neural network by another variable. With compound effects, the picture changes (Tab. 1), e.g., the dynamics of all stress components have combined a large impact on the sea-ice area and thickness, or the absolute values of the dynamical variables combined strongly influence the stress components. However, our result shows in fact that the dynamical behaviour of a variable is the single most important predictor, if no compound effects are taken into account. We will add the Table 1 to the additional results in the Appendix, showing what happens if the information of a complete group is destroyed, which includes compound effects.

RC: **Figure 7: It is striking that the low-resolution model is much worse than simple persistence. This makes me wonder whether the NN is just correcting biases (see general comment #2). Please provide some discussion on this.**

AR: Dynamical processes of below 8 km are in the subgrid-scale of the low-resolution model setup and parametrised with the damaging process. These processes are nevertheless included in the truth fields on which basis the low-resolution forecasts are initialised. The mismatch between resolved and parameterised processes results into a strong drift for the dynamical variables (velocities, stresses, damage) within the first minutes of forecast. This drift is not a simple bias but a dynamical process, because otherwise the bias-corrected forecast would perform better. Contrary to the low-resolution model, the persistence forecast has no drift, and for the dynamical variables a better score for the shown lead time of up to one hour. We will add some sentences that explain the absence of connection to a bias and will refer for this to the Appendix with the results for additional architectures.

RC: **Lines 516 - 519: This is a good start, but a much more in-depth discussion is needed here of the implications and wider applicability of the work presented (see general comment #1).**

AR: As written for comment #1, we will make the proof-of-concept character of the study stronger in the introduction, and we will add some discussion about steps towards the generalization.

[Figure]

Figure 1: The negative log-likelihood for a Laplace approximation, proportional to the mean absolute error (MAE), with a fixed weighting in the validation dataset as function of epochs for different fractions of training data, the brighter the color, the less training data is used. The smaller Figure shows the averaged MAE in the test dataset as function of the fraction of training data. The performance of the neural networks is averaged across ten different random seeds.

[Figure]

Figure 2: Snapshot of sea-ice damage after a one-hour forecast with the here-used sea-ice dynamics only-model. Shown are the high-resolution truth (a, 4 km resolution) and low-resolution forecasts (b, c). To initialise the low-resolution forecasts, the initial conditions of the high-resolution are projected into a low-resolution space with 8 km resolution. Started from these projected initial conditions, the low-resolution forecast (b) generates too much damage compared to the high-resolution field. Running the low-resolution model together with the learned model error correction (c) leads to a better representation of the damaging process, which improves the forecast by 62 % in this example.

[Figure]

Figure 3: (a) The model domain with the high- (red) and low-resolution (blue) grid; (b) snapshot of the forcing wind velocity in $y$-direction in $\mathrm{m\,s^{-1}}$, the arrows indicate the movement direction; (c) snapshot of the stress, $\sigma_{xy}$ in Pa, where the arrows correspond to von Neumann boundary conditions on all four sides; (d) snapshot of the damage, where the arrows correspond to an inflow of undamaged sea ice on all four sides. The three snapshots are taken at an arbitrary time and represent a commonly encountered case in our dataset.

Input: **x**

[Figure]

Table 1: Permutation feature importance of different variable groups. The colouring is the same as in Table 6 of the original manuscript. $SIU$ stands for velocity in $x$-direction, $SIV$ for velocity in $y$-direction, $F$ for wind forcing, $\sigma$ for all stress variables, $DAM$ for damage, $SIA$ for sea-ice area, and $SIT$ for sea-ice thickness.

---

## Author Comment (AC2)

**Response to Referee 2**
**for "Deep learning of subgrid-scale parametrisations for short-term forecasting of sea-ice dynamics with a Maxwell-Elasto-Brittle rheology"**

Finn, T.S., Durand, C., Farchi, A., Bocquet, M.,
Chen, Y., Carrassi, A., Dansereau, V.

19th April 2023

**RC: Reviewer Comment**, AR: Author Response

RC:  In this manuscript, the authors present a novel machine-learning method to correct unresolved sea-ice dynamics in simulations with low resolution. From the comparison of high and low-resolution simulations in an idealized domain neural networks are trained to predict the residual between both simulations at a certain lead time, which demonstrate promising performance. This approach aligns with recent developments in climate research, where machine learning is used to parameterize unresolved processes in low-resolution simulations. The study presents several innovative approaches to sea-ice science and provides a thorough evaluation of the performance of the trained ML algorithms. To the best of my knowledge, this paper is one of the most advanced works employing machine learning in the field of sea-ice dynamics. The presented analysis is sound and requires only a few modifications that I list below. The authors need, however, to improve the paper's presentation of the paper as it can be difficult to follow in major parts. The manuscript is overly packed with information and details that can be challenging to grasp, even with a background in sea-ice dynamics and machine learning. I strongly recommend the manuscript for publication in The Cryosphere after the authors have addressed the issues mentioned and detailed below.

AR: We thank you for the constructive feedback on our manuscript, especially on the presentation of the methods and the many detailed specific comments. In the following we will discuss the raised concerns and what we plan to change in our revised manuscript.

RC: **Target audience: I think the authors should keep two audiences in mind that will be interested in this work: sea-ice scientists and ML experts. The manuscript in its current form describes the ML part, network design, and thorough evaluation of the performance of the NN in great detail. I appreciate this for reproducible science, but am also afraid that the amount of detail makes the manuscript hard to follow for readers with a sea-ice background and limited knowledge of ML. This could be addressed by shortly introducing the many ML concepts before discussing them in length and/or reducing or reorganizing the information content of the paper (which I will explain in the next point). I highly recommend reading and editing the paper through the lenses of both audiences.**

AR: Exactly as you have proposed, we have both audiences in mind. Based on your review and comments of referee 1, we have seen that we might have missed the fine line between both communities by giving too much technical details. To improve the presentation for both communities, we will move the technical parts into an Appendix and explain the sea-ice model and the neural network more concisely and by rather discussing the reasoning behind different choices and components. We hope with these changes, we can satisfy readers from both communities.

RC: **Readability: I had a hard time following the first half of the paper on my first read. After reading the entire manuscript and knowing the subject, I could follow it better on a second read. Therefore, I would suggest editing and maybe restructuring this part of the manuscript thoroughly. In general, the manuscript holds a lot of information in part to describe the set-up, analysis, and results in detail, but also information that is only linked but not strictly necessary for the understanding or interpretation of the paper. Especially the latter makes it hard to stay focused on the storyline. I recommend going through the paper and reconsidering which information is necessarily required. This would also give the authors more space to explain important concepts in more detail. Section 3.1 helped me a lot to understand what you are after and I definitely recommend moving it further up in the manuscript, maybe even into the introduction. I would also consider moving the description of the data generation (Section 4) before the description of the ML, which would help to understand the network design etc. Section 2 is rather long and I would consider shortening it and eventually merging it with Section 4 as both discuss the sea ice model and the simulations. Up to Section 5, I had a hard time finding a storyline to follow. Please try to emphasize your storyline there stronger and try to guide readers better.**

AR: To improve the readability of the manuscript, and as you have proposed, we have decided to change the structure in presenting the methods. We will replace Section

2 by a new Section, where we explain first the problem that we try to solve in mathematical terms (former Section 3.1), second rather shortly the regional sea-ice model and the used forcing, and third the twin experiments. In Section 3, instead of a technical description, we will explain rather superficially the neural network and its reasoning. As we have moved the explanation of the forcing and of the twin experiments into Section 2, Section 4 will be more disentangled and more restricted to specific parameters etc. used to generate the data and to train the neural networks. Additionally, we will be more consistent, in the use of 'subgrid-scale parameterisation' and 'model error correction'. We believe that these changes will increase the readability of the manuscript, especially for non-specialists. Thanks for this suggestion.

RC: **Lead time for update: The authors use a lead time of 10 min 8 s to update the coarse resolution model. While all other design choices have been explained in detail, this is not the case for the lead time. Why did you choose this lead time? Wouldn't you expect a shorter lead time to improve the results? With the existing twin simulations, it is straightforward to extract the residual between the truth and forecast model also at other lead times. Therefore, I strongly suggest studying also the effect of different lead times here. I would be especially interested to see if shorter lead times improve the seesaw patterns of the trajectories of the hybrid model in Figure 7.**

AR: Originally, we have selected the update time quite early in the research based on considerations about the signal-to-noise ratio. In the best case, the neural network corrects model error before they have a too large influence on the forecast, which would be after each integration step. However, the neural network is not trained to take interactions between correction and sea-ice model into account. Furthermore, the predictable error is corrected by the neural network, such that the unpredictable error accumulates over time. As the network is only trained on the first correction step, this leads to a distribution shift (cf. Tab. 8 of the manuscript).

This distribution shift becomes important the more update steps we take. To underline these pros and cons, we have now made tests where we vary the update time (16 s, 80 s, 20 min), as can be seen in Figure 1 of this response (the results for 16 s are not shown, as they are even worse than for 80 s). The seesaw pattern gets hidden behind more update steps. Nevertheless, the distribution shift outweighs that model errors are corrected earlier, which makes the forecasts less performant for smaller update times. This might be a specific concern in our model setup, but is likely to be an issue related to hybrid modelling in general. As these results underline the importance of the distribution shift, we will add a Subsection in the additional results part of the Appendix where we show and shortly discuss Figure 1.

RC: **Generalization: The neural networks presented in the paper are trained on a specific (idealized) model configuration, which is also a good choice**

**for this proof of concept. There is, however, only limited discussion of what steps are needed to use the same approach in other model configurations, especially realistic ones: do users need to train different NNs for each new model configuration, which will get very expensive as high-resolution truth simulations are required? Or can the trained weights of the kernel be applied also to different grid geometries in different configurations or could be used as starting weights to reduce the amount of training data? A discussion of these considerations would be helpful to get an impression of how feasible and flexible this approach can be applied in other model set-ups.**

AR: This study has a very limited scope of giving a proof-of-concept. For us, it is evident that the way towards operational settings is rather long, given the issues of missing interactions and stochasticity. Additionally, if neural networks are trained with twin experiments, they "only" learn to imitate the model that was used to generate the truth. We rather see the potential for model error corrections coming from the inclusion of observations into the learning process, which complicates model error correction even more. To give some hope, especially related to the question of the reusability, we might be lucky in sea-ice modelling: as sea-ice exhibits multifractality/self-similarity, the same model error correction might be usable across spatial scales, or at least a good starting point for retraining on a different resolution or domain.

In the revised version of the manuscript, we will provide some more discussions upon the next steps and what is missing towards an operational setting.

RC: **L1: "of"**
**Remove "of" as "subgrid-scale" is an adjective.**

AR: We will remove "of" from the title.

RC: **Abstract: I would consider rearranging the abstract, maybe shortening sentences. Might be a matter of taste, but I had a hard time following it reading it the first time.**

AR: We will simplify the abstract and its language wherever we can.

RC: **L5: includes important inductive biases needed for sea-ice dynamics. Unclear what is meant by these biases.**

AR: We will remove this subordinate clause from the abstract.

RC: **L7: we cast the subgrid-scale parametrisation as model error correction. Unclear, please rephrase.**

AR: We will reformulate to "Instead of parameterising single processes, our goal is to correct all model variables at the same time."

**RC: L11: cycling**
**What do you mean by cycling?**

AR: We will be more specific by stating: "Correcting the forecasts every 10 minutes, the neural network can be run together with the sea-ice model, which improves the short-term forecast up to one hour."

**RC: L11: physically-explainable input-to-output relation**
**It is not clear what is meant by "physically-explainable", please clarify.**

AR: We will be more specific in what we mean by physically explainable.

**RC: L16: dynamics of sea ice at an unprecedented resolution and accuracy.**
**Please clarify what unprecedented means with respect to the resolution.**
**All three papers use simulations with a resolution of 10km or lower,**
**while much higher resolution sea-ice simulations have been presented.**
**Do you mean unprecedented accuracy at the given resolution?**

AR: We will rephrase to: "at an unprecedented accuracy for Arctic-wide simulations in the mesoscale with horizontal resolutions of around 10 km."

**RC: L16: Elasto-Brittle. Why capitalized? Here and elsewhere**

AR: We will write elasto-brittle in lowercase letters.

**RC: L17: represent**
**Reproduce?**

AR: We will change to "reproduce" as this is indeed a better wording. Thanks for the suggestion.

**RC: L19: single grid cell at the mesoscale. What is meant by mesoscale here?**
**Please clarify**

AR: In the revised manuscript, we will define mesoscale in the first sentence of the introduction.

**RC: L31: the mesoscale. See comment above**
**please define the length scale mesoscale refers to here.**

AR: Same as before, mesoscale is now defined in the introduction.

**RC: Figure 1. Please clarify that (a) shows the high-resolution initial conditions, but (b) and (c) the low-resolution forecasts one hour later. Why not show for all the damage after 1h forecast, so that the reader actually gets an impression if the hybrid model in (c) is closer to the high-resolution "truth" or not?**

**AR:** The Figure shows the field for all simulations after a lead time of one hour, even in the high-resolution case. As this was not clear in the caption, we will clarify it, as shown in Figure 2 of this response. Additionally, given the feedback from Referee 1, this Figure will be changed to a sample more representative for typical situations in the dataset.

**RC: L37: possibly projected**
**What is meant by this? Please clarify**

**AR:** We wanted to state that we cannot use the same initial conditions across different resolutions, so we must project them. We will rephrase this sentence to clarify it.

**RC: L39-40: Here, the low-resolution simulation 40 (b) misses the rapidly developed opening of sea ice in the high-resolution simulation (a).**
**Does this refer to the upper or the lower opening in the figure? Please clarify in the text.**

**AR:** As the shown snapshot will change, see Figure 2 of this response, also the text will be different, and we will be more specific.

**RC: L54-59: paragraph about marginal ice zone:**
**Does your regional model include the MIZ? To me, it looks more like pack ice with cracks. Also along leads there are sharp transitions that the NN needs to handle, so I think it is justified to present this issue here. However, please frame it in a way that fits your problem at hand.**

**AR:** You are right that our problem is more about cracks/leads with such sharp transitions. As the model is unable to represent ice-free cases, it cannot simulate "real" marginal ice zones. We will change the description here and elsewhere in the text, being more consistent with the actual problem represented by our model.

**RC: L56: jump**
**Step function?**

**AR:** Step function is indeed the more accurate wording, thanks.

**RC: L98: as well**
**Remove?**

**AR:** "As well" will be removed.

**RC: L123-124: As the nodes are shared in the first-order elements, there are more grid points for all variables that are defined as zeroth-order elements than for the velocity and forcing components.**
**What is the relevance of this? Could you elaborate if this is an important point needed to be considered to interpret the presented results?**

AR: This reference to more grid points for variables defined by a zeroth-order discontinuous Galerkin discretisation is relevant to explain the need of a common space for the neural network. As this Section will be rewritten in the revised manuscript, and this more technical description will be moved into an Appendix, we will remove the sentence.

RC: **Section 3: A deep learning based subgrid-scale parametrisation**
**The presentation of the machine learning tools is done very thoroughly, which I appreciate and see as valuable for reproductivity. Given that ML applications in this field of science are just emerging and there are many geophysical and climate scientists interested in advancing in this field, I am afraid that the description is presented too high level for an audience with limited knowledge of ML. To also target this part of the scientific community and broaden the audience for this paper, I recommend summarising the main parts and ideas behind it more comprehensively for readers with limited ML background at the beginning of this section. While I see this recommendation as optional, as all necessary information is given in the current draft, I want to emphasize the large beneficial value I see in adding a summary like this.**

AR: As previously written, we will rewrite this Section and be more concise with the technical description in an Appendix. Thank you very much for this constructive comment.

RC: **L147-149: There, linear functions combine pixel-wise (i.e., processing each element defining grid point independently) the extracted features. Each linear function is shared across all grid points for each predicted residual variable.**
**The linear transformation from features to residuals is not clear to me. Does this involve combining different features for each grid point, where the weights of these combinations are learned in the training? Or is it a fixed combination? Please clarify the text accordingly.**

AR: We will be more specific what we mean by linear functions, as they are learnable.

RC: **Figure 3.**
**Does the red, blue, and grey color code for arrows, boxes, and labels have a specific meaning (trainable vs fixed or similar)? If so please give some explanation.**

AR: We will revise the figure, to mark learnable parts by the same colour, and we will specify our colouring code.

RC: **L154: 3.1 Problem formulation**
**This section helps to understand our approach's goal, and I strongly suggest moving it further up (maybe even the introduction) to give the**

**reader a better understanding of what you try to achieve before going into the details of the model or the pipeline.**

AR: This will be the first part of the new Section in the beginning of the methods.

RC: **L180-182: Note, for coarse Cartesian spaces, the mapping from Cartesian space to triangular space can be non-surjective, meaning that not all triangular elements are covered by at least one Cartesian element: the pseudo-inverse is in this case rank deficient.**
**This is unclear to me: why should the bigger triangles of the coarse resolution simulations not be covered by the higher resolution Cartesian elements? If at all, I imagine that should be an issue of the high-resolution grid with smaller triangles. Please clarify.**

AR: As the Cartesian elements are rectangular, there are cases where not every triangular element has an associated Cartesian element in the forward interpolation. Consequently, taking the pseudo-inverse of the interpolation operator for the projection, there are some triangular points that get no information, even if the triangular resolution is around 8 km and the Cartesian resolution around 4 km. This is specific to our choice of taking the pseudo-inverse instead of defining a new projection operator. This technical detail will be moved to an Appendix.

RC: **L196: complete U-net architecture**
**Do I understand the architecture correctly that you downscale only once in your U-Net? If that is the case, the illustration in Figure 3 is misleading, as 4 down scaling steps are shown. Please clarify this and adapt the figure potentially.**

AR: Yes, indeed in the shown architecture there is only one downscaling operation, we will adapt the figure.

RC: **Section 4: Experimental setup**
**This section describes how data to train the NN is created. Consider renaming this section to e.g. "training data generation" or similar. I also would consider moving this section before the details on the ML algorithms as I feel it helps to know the data before getting introduced to the detailed methods.**

AR: We will rename the section to "Data generation". By moving the twin experiment explanation part before the neural network, the ordering should be clearer, and the section more disentangled.

RC: **L305: their expectation**
**their expected value?**

AR: We will change to expected value, as this seems to be a more common name.

**RC:  Table 3 - Caption: MAE**
**Is the MAE computed at high or low resolution?**
**A score of one would correspond to the performance of the geophysical**
**model forecast in the training dataset.**
**Do you mean the coarse resolution geophysical model forecasting the**
**high resolution run? Please clarify.**

AR: We will be more specific about low- and high-resolution in the caption.

**RC:  Table 3 - Caption: the afterwards used architecture and the best scores**
**If this refers to the hybrid model or online use of the correction, please**
**write this. Also, consider writing "that shows" instead "and".**

AR: We will specify that the architecture is used for all experiments afterwards. The
"and" is chosen as the bold font is chosen independently for each score. It is by
chance that all bold numbers are in-line with the hybrid model.

**RC:  L320: persistence forecast performs**
**Could you for clarity once define what you use as a persistence forecast?**
**It might be obvious to you, but for readers outside the field, it will be**
**helpful.**

AR: We will be more specific about the persistence forecast and will define it in the
beginning of the results section.

**RC:  L335: Such localised features**
**Consider adding the length scale in km if you think this finding is**
**generalizable or holds valuable information for other processes related**
**to sea-ice deformation.**

AR: In our case, we know that the missing processes lie between the 8 km and the
4 km, but if learned from observations, it might be indeed interesting to see the
length-scale of the learned model error correction. For the specific case, shown
in Fig. 5, we will add one or two sentences about the features in the different
resolutions.

**RC:  L340: a generally smoother background pattern**
**What is meant by this? Please clarify and rephrase.**

AR: We will clarify what we mean by a generally smoother background pattern.

**RC:  Figure 6**
**I) What colormap uses a)?**
**II) I am wondering if also the concentration maps in the initial and**
**forecast step would be helpful here to interpret the gradients?**

AR: The colormap of (a) is somewhat arbitrary as the normalized field like used for the input into the NN is shown. Instead, we will change to the unnormalized area and show a colorbar.

To make the Figure as simple as possible, we have restricted it to only the information needed to make the point clear. We will see if we add more fields to make the gradient more explainable.

**RC: L378: either**
**Isn't it to both instead of either or?**

AR: Will be changed to "both" and "and".

**RC: L380: Table ??**
**correct reference**

AR: The reference will be corrected.

**RC: L382-383: Additionally, the sensitivity is directional dependent, Fig. 6g, and exhibits localised features, Fig. 6c and i**
**Could you discuss these results also in the light of physical understanding that we can gain from the gradients? From both the gradients along initial and difference, we can learn about the shortcomings of the coarse resolution simulations that the NN tries to compensate for.**

AR: Thank you for this nice suggestion. Our goal was to prove the point of localised features and representation of anisotropy, neglecting its physical meaning. We will add some discussion about the physical meaning along the suggested lines.

**RC: L391: $-1 \times 10^{-3}$ and $1 \times 10^{-3}$**
**Units?**

AR: This sentence is a remnant of an older version of the manuscript. As there is no longer a restriction to the correction of the sea-ice thickness, the sentence will be removed. Thanks for spotting this typo.

**RC: L392: Related to optimal control theory in dynamical systems,**
**This is not very helpful for readers with limited background knowledge. Please elaborate more or rephrase.**

AR: We will change the sentence to "As more commonly used to evaluate forecast performances, we will ..."

**RC: L405-408: Additionally, for the velocity, stress, and damage, the drift towards ... the "Initial + Forecast" experiment in these variables and averaged over all nine model variables.**
**Unclear, please rephrase.**

AR: We will rephrase the sentence to make clearer what we meant.

**RC: L412: As the initial condition error increases with each update, the network corrects less and less forecast errors.**
**Could this effect be dampened by updating at higher frequencies?**

AR: As discussed for your comment # 3, there is a trade-off. It seems that the cons outweigh the pros, and more updates even hurt the performance. As we will add the Figure to the additional results Section in the Appendix, we will refer to these results in L412.

**RC: L413-419:"To show the effect of this error distribution shift, ... An averaged correlation of 1 would indicate a perfect pattern correlation."**
**This paragraph is hardly understandable with no background knowledge of the method "centred spatial pattern correlation". I suggest describing the principle of the method and its interpretation at the beginning of the paragraph in 1-2 sentences, before describing its specifics.**

AR: We will add a short explanation of the spatial pattern correlation.

**RC: L422: especially for the divergent stress**
**From the previous paragraph, it sounds as if a value close to 1 is favorable, but this statement reads as if a high value close to 1 for divergent stress shows a weakness of the NN. Please clarify.**

AR: Considering the results for the sea-ice area, we will clarify that this shows a deficiency of the neural network but of the forecast model.

**RC: L434-435: However, the parametrisation misses the development of new strains and positions the main strain at the wrong place.**
**This suggests that the corrections of the NN violate the brittle model physics, as highly damaged areas are usually linked to high deformation rates. Is this correct? If so, please comment on this also in the text, and if there is a way to design a network that computes corrections in accordance with the physical laws of the model.**

AR: In the hybrid model, the field is damaged beyond a given deformation threshold, shown for the damaging process in the south, as in the high-resolution simulation. Only, the high deformation rates in already damaged areas are striking in the case of the hybrid model. As such high rates are unobserved for the high resolution simulation, it remains unknown if this is an unphysical behaviour or not. Because the last correction of the neural network is already more than 9 minutes ago, we would speculate and say that the model would have had time to adapt to the new situation such that the rates are physical explainable.

**RC: L456-457: Therefore, using such a mapping into Cartesian space, we can apply CNNs, which can efficiently scale to larger, Arctic-wide, models. Are you talking about Arctic-wide models on unstructured grids? Or why is the mapping needed? Please clarify the text.**

AR: We will clarify that we mean Arctic-wide simulations on unstructured grids, e.g., neXtSIM.

RC: **L462-463: As processes have no discretized resolution in realworld, we would have difficulties to find the right resolution for the projection in such cases.**
**Isn't that only an issue if you would aim to train a correction with observations? If it is a model, then you would always know the resolution of resolved scales. Please clarify**

AR: Indeed, this is only an issue if we train with observations. Since in the ideal world, we learn from observations, we raise this issue here. We will disentangle the sentences and make our points clearer, also with respect to the higher-resolved truth.

RC: **L462: truth**
**Please clarify what is meant by truth: the high-resolution simulation or something different**

AR: See previous comment.

RC: **L464: this argument**
**What argument?**

AR: The argument that the optimal resolution is linked to the resolution of the processes in the "truth", either in high-resolution simulations or observations. We will clarify the sentence.

RC: **L503: The only way is therefore to improve the forecast model, thereby changing its attractor.**
**What about updating the forecasting model at higher frequencies? Please comment.**

AR: In the most extreme case, we would correct after each integration time step, which could be seen as integrated form of a subgrid-scale parameterisation. In this case, we would change the attractor, as each forecast is now a corrected forecast. The problem is rather to define the attractor of the hybrid model; one could define only the corrected states as attractor. We will make such points clearer here.

RC: **L530-532: Mapping the input data into a Cartesian space that has a higher resolution than the original space, such scalable convolutional neural networks can be applied for feature extraction in sea-ice models defined on a triangular or unstructured grid.**
**Something is wrong with this sentence, please correct it.**

AR: This sentence does make indeed no sense, we will rephrase it.

**RC: L543: total deformation**
**On which Figure or result is the statement that the total deformation is improved? From Fig. 8 ,I would agree that the damage in the hybrid model looks closer to the high-resolution run than the uncorrected low-resolution simulation, but for total deformation, it is the other way around in my eyes.**

AR: We agree that the point-wise error for the hybrid forecast in the total deformation is higher. Hence, we will rephrase it into: "... to a better representation of the damaging processes than in the forecast model without parametrisation".

**RC: L552: Appendix**
**Table?**

AR: In the current online version of the manuscript, the Table is correctly referenced?

**RC: L562-564: Caused by their limited capacity, the NNs have to focus on some variables, creating an imbalance between variables, which harms the performance for other variables, like the stress in the case of "Conv (×5)".**
**Have you tried to train individual networks for each variable, which could balance this effect?**

AR: In the initial phase of the research, we have seen that one big neural network with shared parameters performs better than nine small neural networks. Consequently, we have concentrated the research on a single big neural network. As one big network shares the features for all variables, it has to learn more general features than networks for single variables. The shared features act as regularisation which can help to reduce the overfitting. Hence, learning one network for all variables can enable the use of larger neural networks. We currently run a test for the "Conv (×1)" architecture with multiple networks and add the results to Table B1. As preliminary results, we see that learning single "Conv (×1)" networks per variable has better scores than the shared "Conv (×1)" architecture. However, the single networks have already a larger error compared to the shared "Conv (×5)" architecture, although they have in total more parameters, and they are slower to train. Additionally, the single networks have the same imbalances as evident for the shared network. Consequently, this confirms our initial results that one single big neural network for all variables performs better than one small network per variable.

**RC: L565: fast NN**
**Is the speed of the trained U-NeXt NN an issue compared to the computational costs of the geophysical model? Or does this refers to training speeds?**

AR: Compared to the geophysical model, the neural network is fast, if correctly implemented into the model, e.g., using the C++ libraries of PyTorch or TensorFlow.

However, such kind of model error correction sits on top of a geophysical model, its runtime is additive to the runtime of the geophysical model. Consequently, the faster and lightweight the model, the better. So, it is in some sense a trade-off between additional runtime and additional gain, and, for some purposes, the Conv (×1) architecture might be enough.

[Figure]

Figure 1: Normalised RMSE for (a) the velocity in $y$-direction, (b) the divergent stress in $y$-direction, (c) the damage, and (d) the sea-ice area as function of lead time on the test dataset, normalised by the expected RMSE on the training dataset for a lead time of 10 min and 8 s. In the hybrid models, the forecast is corrected after each specified lead time in brackets, and the performance is averaged over all ten random seeds

[Figure]

Figure 2: Snapshot of sea-ice damage after a one-hour forecast with the here-used sea-ice dynamics only-model. Shown are the high-resolution truth (a, 4 km resolution) and low-resolution forecasts (b, c). To initialise the low-resolution forecasts, the initial conditions of the high-resolution are projected into a low-resolution space with 8 km resolution. Started from these projected initial conditions, the low-resolution forecast (b) generates too much damage compared to the high-resolution field. Running the low-resolution model together with the learned model error correction (c) leads to a better representation of the damaging process, which improves the forecast by 62 % in this example.

---

## Author Response (AR1)

**Author's response**

**for "Deep learning subgrid-scale parametrisations for short-term forecasting of sea-ice dynamics with a Maxwell-elasto-brittle rheology"**

Finn, T.S., Durand, C., Farchi, A., Bocquet, M., Chen, Y., Carrassi, A., Dansereau, V.

5th May 2023

RC: Reviewer Comment, AR: Author Response

**1** Response to Referee **1**

RC: The authors present an idealised study of sea-ice fracture in a channel due to wind forcing, demonstrating that a neural network (NN) is able to significantly reduce errors of a lower-resolution version of the physical model with respect to a higher-resolution version of the same model for 10-minute forecasts. They conclude that the NN has learned the tendencies from the unresolved scales in the lower-resolution model, and can therefore be used to parameterise these unresolved scales. I appreciate the originality of the work and the level of detailed analysis

it provides. It fits well with current efforts in the community to use machine learning for parameterisation of unresolved scales in geophysical models. However, given the very idealised setup, I have some concerns about the wider applicability of the results. Below, I spell that out in comments which I would like the authors to address before publication:

- AR: We thank referee 1 for the constructive feedback on our manuscript, especially with respect to the bias correction and a possible overfitting. In the following, we discuss the raised concerns and what we have changed in our revised manuscript.
- RC: There are a number of very strong idealisations and restrictions in the setup of this study: a) it is a so-called "perfect model" study, i.e. the performance of the lower-resolution model with/without NN corrections is assessed against a "truth" which is a simulation of that same model at higher resolution, without involving any observations or simulations from a different model; b) The forecast lead times considered are extremely

short for most real-life weather and climate applications (only up to 1h); c) spatial domain is a simple rectangular channel; d) no treatment of sea-ice thermodynamics. Given these very strong idealisations and restrictions, one would hope for results that are a bit more convincing than the ones presented. I have concerns about whether the methods presented will be useful in a more realistic context, where each of the above assumptions will need to be relaxed. Can the authors please add some in-depth discussion (or even preliminary analysis) about what they think will happen if their methods are applied in a more realistic context?

- AR: Our study is designed to be a proof-of-concept. As sea ice imposes novel challenges for neural networks, it was previously unknown if model error corrections in this form are possible at all for sea-ice modelling. We think this study shows that there is indeed a huge potential for hybrid modelling of sea ice. Given the limited scope of this study, we have decided to apply such simplified settings, far from settings in operational forecasting or projections. For example, we have used the twin experiment setting to prove our points and to cheaply generate a known truth. If realistic model error corrections would be trained with twin experiments, the neural network would learn to emulate the fields from the higher resolution, so, instantiations of already known processes. Consequently, we believe that the true potential lies in the possible learning of model error corrections from observations, which is beyond the scope of our proof-of-concept. Furthermore, the model error correction is designed to correct model errors as soon as possible, before they have a too large impact on the forecast. This is why we concentrate on such short forecast lead times of up to one hour, although they might be far from operational settings. In further studies, with more realistic setups, we will investigate the impact of the model error corrections in longer forecast lead times in more realistic settings. To take this concern into account, we have strengthened the proof-of-concept character of the study in the introduction. Additionally, we have extended the section "Summary and Discussion" by few paragraphs about next steps that might follow towards more generalised and realistic settings. We added more discussion about the apparent model drift, about a possible application to other resolutions
- RC: Figure 7 and the corresponding text makes me wonder how much of the error reduction achieved by the NN is actually due to correcting the bias (i.e. mean error) of the low-resolution simulation w.r.t. the highresolution simulation. Can the authors please provide some analysis to quantify the contribution of bias to the overall errors, with and without the NN corrections? For instance, one could just decompose the mean squared errors shown in the manuscript into squared bias and variance of the errors. I am asking this because there is a range of other methods

operational settings.

and model settings, and about applying twin experiments for learning in more

to treat biases (e.g. a-priori by tuning the model, and a-posteriori by subtracting them from the forecast before further analysis). These methods are often simpler than the machine-learning approach and are in wide use in the weather and climate community. Utilising a complex and costly machine-learning approach only pays off it is clearly superior to other available methods.

AR: As shown in Appendix B, Table B1, we made tests where we simply correct the constant model bias. The performance of this bias corrected model is almost the same as for the raw model without any correction. Most of the model error is hence temporally and spatially variable and cannot be corrected by a simple bias correction. Additionally, we made the tests with a rather small neural network, where only one multi-scale convolutional layer is used. We expect that even simpler methods, e.g., a linear regression, perform worse than this small neural network. Additionally, there are many possibilities with neural networks that we have not taken into account, e.g., generating stochastic parametrisations with correlations learned from data. Therefore, we believe that this study indicates the potential for hybrid modelling, which would be otherwise unachievable.

In Sect. 5.1, "Performance on the test dataset", we have added a reference to the appendix, showing that correcting the bias has almost no impact on the performance on the test dataset. In Sect. 6, "Summary and Discussion", we have partially streamlined the language to clarify further possibilities with neural networks, which makes our point of using neural networks hopefully clearer.

- RC: Following up on the previous comment, I would like the authors to comment on potential overfitting of the NN in their methods. If I did the maths correctly, there are about 4500 degrees of freedom in the lower-resolution physical model (9 variables times 500 grid points). As stated on line 197, the NN has 1.2 million trainable parameters. So one could argue the NN has orders of magnitude more degrees of freedom than features it is learning from or results it is predicting. I am not an expert on machine learning, but that strikes me as odd could the authors please comment on that? I would also like to see some quantitative analysis on the risk of overfitting.
- AR: We agree that a single low-resolution field has only 2558 degrees-of-freedom (DOFs). Compared to this number, the number of parameters in the neural network  $(1.2 \times 10^6)$  seems to be very high. However, the training dataset has  $2558 \times \text{number of samples}$  DOFs. In the end, this sums up to around  $12.3 \times 10^6$  million DOFs, an order of magnitude larger than the parameters in the neural network. During training, the scores in the validation dataset have been smoothly improving without a sign of overfitting. Furthermore, we have made new experiments (Fig. 1), where we have only used a fraction of the data for training. Even in our most extreme case with only 10 % of the data (480 samples), the model is not overfitting with smoothly decreasing MSE and MAE, and we achieve a nice scaling of the performance with

the number of samples. We attribute this behaviour to the projection into Cartesian space and to the strategy of learning a correction for all variables at the same time. Caused by the projection from Cartesian to triangular space, the information content of the extracted features is reduced. Additionally, learning for multiple outputs at the same time acts as a type of regularisation for a single output, the network has to find features that suits all outputs.

In Sect. 4 "Data generation and training", we have introduced a paragraph where we discuss the number of training samples and a possible overfitting, caused by the degrees-of-freedom in the neural network. Additionally, we have introduced a new subsection (Appendix D1), where we discuss Fig. 1, which shows the influence of the number of training samples on the results in the testing dataset.

**RC: Please revise the presentation of the methods, this is not sufficient in some places, and difficult to follow in others. See technical comments.**

- AR: Based on this raised concern and the other review, we have decided to rearrange the presentation of the methods. The sea-ice model and the neural network is discussed in a hopefully easier language, explaining rather the reasoning behind different components, instead of the technical details. The more technical explanations are moved into Appendix A for the sea-ice model and Appendix B for the neural network. We have introduced a new Sect. 2, where we introduce the model error correction problem from a mathematical standpoint (previously Section 3.1), a shortened explanation of the sea-ice model, and, finally, an introduction to twin experiments. Furthermore, we have streamlined Sect. 3 about the neural network and focussed more on the reasoning behind the different components in the pipeline.
- RC: I am afraid I do not quite understand the motivation why a projection to a Cartesian grid is needed (Section 3.2). It seems to complicate the methods unnecessarily. Can the authors please clarify the motivation for doing this, and what the feasibility/implications would be of doing the analysis on the original triangular grid? Is this just a reflection of the fact that the standard machine-learning libraries for spatial analysis cannot deal with non-Cartesian grids?
- AR: Compared to more "classical" neural networks, so-called multilayer perceptrons, by construction, convolutional neural networks (CNNs) are biased towards localised features, motif extraction across all grid points, and a directional dependency. Additionally, the backend libraries e.g., TensorFlow and PyTorch, are optimised for image processing. CNNs are hence especially efficiently implemented for Cartesian spaces. Although there are different convolutional architectures better suited for unstructured grids, e.g., graph neural networks, they are usually more computationally heavy and more difficult to implement. Given the limited scope of our proof-of-concept, we have thus decided to make use of "normal" convolutional neural networks. Additionally, the variables are different at different positions on the triangles. Consequently, we interpolate from triangular space to a common

Cartesian space where the features are extracted. Combined with projecting the features back into triangular space and linearly combining them therein, this architecture turns out to be very efficient and can act as a baseline approach for further studies.

In our rewritten Sect. 3, we have dedicated a full paragraph to the use of why a projection step is necessary for convolutional neural networks. As motivation, we have specifically written: "Convolutional NNs are optimised for their use on Cartesian spaces, where they can easily exploit spatial autocorrelations. The model variables are additionally defined on different positions at the triangles: the velocities are defined on the nodes of the triangles, whereas all other variables are constant across a triangle".

- RC: Figure 1: Please specify which physical variable that is displayed (damage?). Could the authors find a more convincing "showcase" example? By visual inspection, it looks to me like there is still substantial errors in the "hybrid" field, which seems at odds with the claim of an 75 % error reduction. Please quantify the error reduction for the case shown.
- AR: There was a technical issue to correctly render the file on Copernicus' side. This should be fixed now, and the missing labels etc. should be there now. The shown damage fields are for a lead time of 60 minutes. Given your feedback, we have decided to change the snapshot of the sea-ice damage (Fig. 1 in the manuscript) to a more representative case (Fig. 2). There, the improvement is 62 %, and the hybrid model is able to correctly represent the fracturing processes such that not too much damage is produced as in the low-resolution forecast without correction.
- RC: l. 35f. (and elsewhere): I am not sure what "wave-like" and "channellike" - please be more precise.
- AR: In the revised manuscript, we are more specific and have replaced 'channel-like' by specifying the domain dimensions and 'wave-like' by "an external wind forcing with a sinusoidal velocity in *y*-direction".
- RC: Line 76 & 89: a 10 minute (or even 1 hour) forecast is extremely short both for main-stream earth system models and real-world applications. Can you please comment on that and justify looking at these very short time scales?
- AR: On the one hand, we want to correct the model error after each integration step, which would be in our case 16 seconds. On the other hand, the neural network is not perfect, and the more signal during training, the better. Furthermore, the neural network is trained without taking interactions with the sea-ice model into account. The missing interactions lead to a distribution shift during the application of the model error correction. Consequently, using a correction time of 10 minutes is already a compromise. Given the limited scope of this proof-of-concept and an

already visible distribution shift (Table 8 in the manuscript), we have restricted the forecast time to one hour.

- RC: The introduction in ll. 80-92 already gives too much technical detail about the methods. This belongs elsewhere.
- AR: We have reduced the amount of technical information in the introduction, especially at its end.
- RC: In Figure 2 and the corresponding text, the authors need to help the reader to get a physical understanding of the situation that causes the ice to fracture. Please add arrows indicating the wind field, and refer to Equation (1). Please specify which direction is x and which is y. Also Figure 2: Please use other colours than black and red to indicate the two grids, otherwise it is difficult to see for color-blind people.
- AR: We have added the forcing field with arrows for this specific case in Figure 2b, as shown in Fig. 3. Furthermore, we have changed the colour of the coarse grid in Figure 2a to a lighter blue tone, which should make the figure easier to read.
- RC: Line 134: I do not know what "wave-like" means. Please be more precise, and provide the equation with the wind forcing at the earliest possible place in the text.
- AR: We have introduced the equation for the wind forcing in the new shortened description of the regional sea-ice model in Sect. 2.2, and have avoided the term 'wave-like'.
- RC: Figure 6: I much appreciate the sensitivity testing in Section 5.2, very good! However, I am puzzled by the very weak cross-variable coupling in the permutation feature importance. It seems contradictory to your claim that the NN has "learned the dynamics" of the physical model. For instance, for damage as an output variable, it seems that the NN only extracts information from the damage itself, all other input variables are unimportant! Could you please provide some more explanation/clarification/analysis on this?
- AR: The input variables are naturally coupled to each other, e.g., the forcing and the velocities, the area and thickness, or the stress components. Therefore, by destroying the information of a single variable, almost the same information might be still available to the neural network by another variable. Additionally, the shuffling of a single variable can lead to unphysical instances. Consequently, we have introduced a new subsection, Appendix D6, where we discuss the impact of permuting variable groups.

By permuting variable groups (Table 1), we find, e.g., a sensitivity of the sea-ice area and thickness on the stress components. Such additional sensitivities indicate an underestimation of the feature importance across different variables. Furthermore, we show that the generation of unphysical instances has a large impact on the feature importance. If they are shuffled alone, the stress components have a larger importance on the predictions of the stress components than shuffled together. The neural network works with relative features that take the difference between different variables, which is artificially large if only a single variable is shuffled. Nevertheless, the permutation feature importance indicates clearly that the neural network uses the dynamics of the output variables as basis for such relative features.

- RC: Figure 7: It is striking that the low-resolution model is much worse than simple persistence. This makes me wonder whether the NN is just correcting biases (see general comment #2). Please provide some discussion on this.
- AR: Dynamical processes of below 8 km are in the subgrid-scale of the low-resolution model setup and parametrised with the damaging process. These processes are nevertheless included in the truth fields on which basis the low-resolution forecasts are initialised. The mismatch between resolved and parameterised processes results into a strong drift for the dynamical variables (velocities, stresses, damage) within the first minutes of forecast. This drift is not a simple bias but a dynamical process, because otherwise the bias-corrected forecast would perform better. Contrary to the low-resolution model, the persistence forecast has no drift, and for the dynamical variables a better score for the shown lead time of up to one hour.

We have added to the discussion of the results in Sect. 5.3, "As correcting the bias has almost no impact on the performance in the test dataset (Appendix C), this drift is not caused by model biases"..

**RC: Lines 516 - 519: This is a good start, but a much more in-depth discussion is needed here of the implications and wider applicability of the work presented (see general comment #1).**

AR: As written for comment #1, we emphasised the proof-of-concept character of the study, and we added paragraphs about next steps towards a generalization.

**2 Response to Referee 2**

- RC: In this manuscript, the authors present a novel machine-learning method to correct unresolved sea-ice dynamics in simulations with low resolution. From the comparison of high and low-resolution simulations in an idealized domain neural networks are trained to predict the residual between both simulations at a certain lead time, which demonstrate promising performance. This approach aligns with recent developments in climate research, where machine learning is used to parameterize unresolved processes in low-resolution simulations. The study presents several innovative approaches to sea-ice science and provides a thorough evaluation of the performance of the trained ML algorithms. To the best of my knowledge, this paper is one of the most advanced works employing machine learning in the field of sea-ice dynamics. The presented analysis is sound and requires only a few modifications that I list below. The authors need, however, to improve the paper's presentation of the paper as it can be difficult to follow in major parts. The manuscript is overly packed with information and details that can be challenging to grasp, even with a background in sea-ice dynamics and machine learning. I strongly recommend the manuscript for publication in The Cryosphere after the authors have addressed the issues mentioned and detailed below.
- AR: We thank you for the constructive feedback on our manuscript, especially on the presentation of the methods and the many detailed specific comments. In the following we will discuss the raised concerns and what we have changed in our revised manuscript.
- RC: Target audience: I think the authors should keep two audiences in mind that will be interested in this work: sea-ice scientists and ML experts. The manuscript in its current form describes the ML part, network design, and thorough evaluation of the performance of the NN in great detail. I appreciate this for reproducible science, but am also afraid that the amount of detail makes the manuscript hard to follow for readers with a sea-ice background and limited knowledge of ML. This could be addressed by shortly introducing the many ML concepts before discussing them in length and/or reducing or reorganizing the information content of the paper (which I will explain in the next point). I highly recommend reading and editing the paper through the lenses of both audiences.
- AR: Exactly as you have proposed, we have both audiences in mind. Based on your review and comments of referee 1, we have seen that we might have missed the fine line between both communities by giving too much technical details. To improve the presentation for both communities, we have moved the technical parts into

Appendix A for the sea-ice model, and into Appendix B for the neural network. We have introduced more concise explanation of the sea-ice model in Sect. 2.2 and of the neural network in Sect. 3. There, we rather discuss the reasoning behind different components of the model and neural network pipeline. We hope that these changes make the manuscript more accessible for a more general audience.

- RC: Readability: I had a hard time following the first half of the paper on my first read. After reading the entire manuscript and knowing the subject, I could follow it better on a second read. Therefore, I would suggest editing and maybe restructuring this part of the manuscript thoroughly. In general, the manuscript holds a lot of information in part to describe the set-up, analysis, and results in detail, but also information that is only linked but not strictly necessary for the understanding or interpretation of the paper. Especially the latter makes it hard to stay focused on the storyline. I recommend going through the paper and reconsidering which information is necessarily required. This would also give the authors more space to explain important concepts in more detail. Section 3.1 helped me a lot to understand what you are after and I definitely recommend moving it further up in the manuscript, maybe even into the introduction. I would also consider moving the description of the data generation (Section 4) before the description of the ML, which would help to understand the network design etc. Section 2 is rather long and I would consider shortening it and eventually merging it with Section 4 as both discuss the sea ice model and the simulations. Up to Section 5, I had a hard time finding a storyline to follow. Please try to emphasize your storyline there stronger and try to guide readers better.
- AR: To improve the readability of the manuscript, and as you have proposed, we have decided to change the structure in presenting the methods. We have introduced a new Sect. 2, "Twin experiments for deep learning a model error correction". First, we formulate our problem in mathematical terms (former Sect. 3.1). Secondly, we rather shortly introduce the regional sea-ice model and the used forcing. Thirdly, we explain the twin experiments and how they are used to learn the model error correction. In the rewritten Sect. 3, "A convolutional U-Net baseline", we have rather superficially explained the neural network. As the explanation of the forcing and twin experiments is moved into Sect. 2, Sect. 4 is more disentangled and more restricted to data generation and the training of the neural networks. Additionally, we have tried to be more consistent in the use of 'subgrid-scale parameterisation' and 'model error correction'. We believe that these changes have significantly increased the readability of the manuscript, especially for non-specialists. Thank you for this suggestion.
- RC: Lead time for update: The authors use a lead time of 10 min 8 s to update the coarse resolution model. While all other design choices

have been explained in detail, this is not the case for the lead time. Why did you choose this lead time? Wouldn't you expect a shorter lead time to improve the results? With the existing twin simulations, it is straightforward to extract the residual between the truth and forecast model also at other lead times. Therefore, I strongly suggest studying also the effect of different lead times here. I would be especially interested to see if shorter lead times improve the seesaw patterns of the trajectories of the hybrid model in Figure 7.

AR: Originally, we have selected the update time quite early in the research based on considerations about the signal-to-noise ratio. In the best case, the neural network corrects model error before they have a too large influence on the forecast, which would be after each integration step. However, the neural network is not trained to take interactions between correction and sea-ice model into account. Furthermore, the predictable error is corrected by the neural network, such that the unpredictable error accumulates over time. As the network is only trained on the first correction step, this leads to a distribution shift (cf. Tab. 8 of the manuscript).

This distribution shift becomes important the more update steps we take. To underline these pros and cons, we have now made tests where we vary the update time (16 s, 80 s, 20 min), as can be seen in Figure 4 of this response (the results for 16 s are not shown, as they are even worse than for 80 s). The seesaw pattern gets hidden behind more update steps. Nevertheless, the distribution shift outweighs that model errors are corrected earlier, which makes the forecasts less performant for smaller update times. This might be a specific concern in our model setup, but is likely to be an issue related to hybrid modelling in general. As these results underline the limitations of the NN caused by the distribution shift, we have added a Subsection in Appendix D2, where we show and shortly discuss Figure 4.

- RC: Generalization: The neural networks presented in the paper are trained on a specific (idealized) model configuration, which is also a good choice for this proof of concept. There is, however, only limited discussion of what steps are needed to use the same approach in other model configurations, especially realistic ones: do users need to train different NNs for each new model configuration, which will get very expensive as high-resolution truth simulations are required? Or can the trained weights of the kernel be applied also to different grid geometries in different configurations or could be used as starting weights to reduce the amount of training data? A discussion of these considerations would be helpful to get an impression of how feasible and flexible this approach can be applied in other model set-ups.
- AR: This study has a very limited scope of giving a proof-of-concept. For us, it is evident that the way towards operational settings is rather long, given the issues of the distribution shift and stochasticity. Additionally, if neural networks are trained with twin experiments, they "only" learn to imitate the model that was used to

generate the truth. We rather see the potential for model error corrections coming from the inclusion of observations into the learning process, which complicates model error correction even more. To give some hope, especially related to the question of the reusability, we might be lucky in sea-ice modelling: as sea-ice exhibits multifractality/self-similarity, the same model error correction might be usable across spatial scales, or at least a good starting point for retraining on a different resolution or domain.

In the revised version of the manuscript, we have extended Sect. 6, "Summary and Discussion", by few paragraphs about next steps that might follow towards more generalised and realistic settings. We added more discussion about the apparent model drift, about a possible application to other resolutions and model settings, and about applying twin experiments for learning in more operational settings.

**RC: L1: "of" Remove "of" as "subgrid-scale" is an adjective.**

- AR: We have removed "of" from the title, now "Deep learning subgrid-scale parametrisations for short-term forecasting of sea-ice dynamics with a Maxwell-elasto-brittle rheology".
- RC: Abstract: I would consider rearranging the abstract, maybe shortening sentences. Might be a matter of taste, but I had a hard time following it reading it the first time.
- AR: We have rewritten the abstract, trying to simplify the language and shorten the sentences. We hope that the abstract is easier to read and follow now.
- RC: L5: includes important inductive biases needed for sea-ice dynamics. Unclear what is meant by these biases.
- AR: As the abstract is rewritten, we have removed this subordinate clause from the abstract.
- RC: L7: we cast the subgrid-scale parametrisation as model error correction. Unclear, please rephrase.
- AR: In the revised abstract, we have moved this sentence up and have rewritten it to: "Instead of parameterising single processes, a single neural network is trained to correct all model variables at the same time".

**RC: L11: cycling What do you mean by cycling?**

AR: In the revised abstract, we have reformulated the sentence to: "Applied to correct the forecasts every 10 minutes, the neural network is run together with the sea-ice model.". Additionally, we have avoided the word cycling in the whole manuscript now.

**RC: L11: physically-explainable input-to-output relation It is not clear what is meant by "physically-explainable", please clarify.**

- AR: In the revised abstract, we have avoided the use of "physically-explainable" and have rewritten the sentence to: "Furthermore, the neural network extracts localised and directional dependent features, which points towards the shortcomings of the low-resolution simulations.", integrating one of your suggestions to analyse the physical meaning of the learned features.
- RC: L16: dynamics of sea ice at an unprecedented resolution and accuracy. Please clarify what unprecedented means with respect to the resolution. All three papers use simulations with a resolution of 10km or lower, while much higher resolution sea-ice simulations have been presented. Do you mean unprecedented accuracy at the given resolution?
- AR: We have rephrased the sentence to: "with an unprecedented accuracy for Arctic-wide simulations in the mesoscale with horizontal resolutions of around 10 km".

**RC: L16: Elasto-Brittle. Why capitalized? Here and elsewhere**

AR: We have rewritten elasto-brittle in lowercase letters everywhere.

**RC: L17: represent Reproduce?**

- AR: We have changed to "reproduce" as this is indeed a better wording. Thank you for the suggestion.
- RC: L19: single grid cell at the mesoscale. What is meant by mesoscale here? Please clarify
- AR: In the revised manuscript, we have defined mesoscale in the first sentence by specifying it to resolutions of around 10 km.
- RC: L31: the mesoscale. See comment above please define the length scale mesoscale refers to here.
- AR: Same as before, mesoscale is now defined in the introduction.
- RC: Figure 1. Please clarify that (a) shows the high-resolution initial conditions, but (b) and (c) the low-resolution forecasts one hour later. Why not show for all the damage after 1h forecast, so that the reader actually gets an impression if the hybrid model in (c) is closer to the high-resolution "truth" or not?
- AR: The Figure shows the field for all simulations after a lead time of one hour, even in the high-resolution case. As this was not clear in the caption, we have clarified it, as shown in Fig. 2. Additionally, given the feedback from Referee 1, this Figure has been changed to a different sample from the test dataset.

**RC: L37: possibly projected What is meant by this? Please clarify**

- AR: We wanted to state that we cannot use the same initial conditions across different resolutions, so we must project them. We have rephrased the sentence to: "Initialised with the same but projected initial conditions, a simulation at a 8 km horizontal resolution leads to a different trajectory".
- RC: L39-40: Here, the low-resolution simulation 40 (b) misses the rapidly developed opening of sea ice in the high-resolution simulation (a). Does this refer to the upper or the lower opening in the figure? Please clarify in the text.
- AR: The shown snapshot has been changed. Consequently, the text is changed to: "in the transition zones, the low-resolution simulation fractures the sea ice too strongly compared to the high-resolution".
- RC: L54-59: paragraph about marginal ice zone: Does your regional model include the MIZ? To me, it looks more like pack ice with cracks. Also along leads there are sharp transitions that the NN needs to handle, so I think it is justified to present this issue here. However, please frame it in a way that fits your problem at hand.
- AR: You are right that our problem is more about cracks/leads with such sharp transitions. As the model is unable to represent ice-free cases, it cannot simulate "real" marginal ice zones. We have changed the description here and elsewhere in the text to ?"leads", being more consistent with the actual problem represented by our model.

**RC: L56: jump Step function?**

AR: Step function is indeed the more accurate wording, thanks.

**RC: L98: as well Remove?**

- AR: "As well" has been removed.
- RC: L123-124: As the nodes are shared in the first-order elements, there are more grid points for all variables that are defined as zeroth-order elements than for the velocity and forcing components.What is the relevance of this? Could you elaborate if this is an important point needed to be considered to interpret the presented results?
- AR: This reference to more grid points for variables defined by a zeroth-order discontinuous Galerkin discretisation is relevant to explain the need of a common space for the neural network. As this Section have been rewritten in the revised manuscript,

and this more technical description have been moved into an Appendix, we have removed the sentence.

- RC: Section 3: A deep learning based subgrid-scale parametrisation The presentation of the machine learning tools is done very thoroughly, which I appreciate and see as valuable for reproductivity. Given that ML applications in this field of science are just emerging and there are many geophysical and climate scientists interested in advancing in this field, I am afraid that the description is presented too high level for an audience with limited knowledge of ML. To also target this part of the scientific community and broaden the audience for this paper, I recommend summarising the main parts and ideas behind it more comprehensively for readers with limited ML background at the beginning of this section. While I see this recommendation as optional, as all necessary information is given in the current draft, I want to emphasize the large beneficial value I see in adding a summary like this.
- AR: As previously written, we have rewritten this Section in the main manuscript and moved the more technical description to Appendix B. Thank you very much for this constructive comment.
- RC: L147-149: There, linear functions combine pixel-wise (i.e., processing each element defining grid point independently) the extracted features. Each linear function is shared across all grid points for each predicted residual variable.

The linear transformation from features to residuals is not clear to me. Does this involve combining different features for each grid point, where the weights of these combinations are learned in the training? Or is it a fixed combination? Please clarify the text accordingly.

- AR: We have been more specific what we mean by linear functions, as they are learnable: "Back in the triangular space, the extracted features are combined by learnable linear functions. These linear functions process each element-defining grid point independently but using the same weights across all grid points. To estimate their own model error correction out of the features, each of the nine model variables has its own linear function".
- RC: Figure 3. Does the red, blue, and grey color code for arrows, boxes, and labels have a specific meaning (trainable vs fixed or similar)? If so please give some explanation.
- AR: We have revised the figure and have marked the learnable and fixed parts by consistent colours, as shown in Fig. 5.

**RC: L154: 3.1 Problem formulation This section helps to understand our approach's goal, and I strongly**

suggest moving it further up (maybe even the introduction) to give the reader a better understanding of what you try to achieve before going into the details of the model or the pipeline.

- AR: Thank you for the suggestion. We have moved this part to the newly introduced Sect. 2.1, before explaining any other method.
- RC: L180-182: Note, for coarse Cartesian spaces, the mapping from Cartesian space to triangular space can be non-surjective, meaning that not all triangular elements are covered by at least one Cartesian element: the pseudo-inverse is in this case rank deficient. This is unclear to me: why should the bigger triangles of the coarse resolution simulations not be covered by the higher resolution Cartesian elements? If at all, I imagine that should be an issue of the high-resolution

grid with smaller triangles. Please clarify.

- AR: As the Cartesian elements are rectangular, there are cases where not every triangular element has an associated Cartesian element in the forward interpolation. Consequently, taking the pseudo-inverse of the interpolation operator for the projection, there are some triangular points that get no information, even if the triangular resolution is around 8 km and the Cartesian resolution around 4 km. This is specific to our choice of taking the pseudo-inverse instead of defining a new projection operator. This technical detail has been moved to an Appendix.
- RC: L196: complete U-net architecture Do I understand the architecture correctly that you downscale only once in your U-Net? If that is the case, the illustration in Figure 3 is misleading, as 4 down scaling steps are shown. Please clarify this and adapt the figure potentially.
- AR: Yes, indeed in the shown architecture there is only one downscaling operation, we have adapted the figure, as shown in Fig. 5.

**RC:** Section 4: Experimental setup**

- This section describes how data to train the NN is created. Consider renaming this section to e.g. "training data generation" or similar. I also would consider moving this section before the details on the ML algorithms as I feel it helps to know the data before getting introduced to the detailed methods.
- AR: We have renamed the section to "Data generation and training". We have moved the explanation of the twin experiments into the newly introduced Sect. 2.3. The ordering should be clearer now and Sect. 4 more disentangled.
- RC: L305: their expectation their expected value?

- AR: We have changed to "expected value", as this seems to be a more common name.
- RC: Table 3 Caption: MAE
  Is the MAE computed at high or low resolution?
  A score of one would correspond to the performance of the geophysical model forecast in the training dataset.
  Do you mean the coarse resolution geophysical model forecasting the high resolution run? Please clarify.
- AR: We have been more specific about low- and high-resolution in the caption and written: "Normalised MAE on the test dataset, estimated in low-resolution, and averaged over ten NNs trained with different seeds".
- RC: Table 3 Caption: the afterwards used architecture and the best scores If this refers to the hybrid model or online use of the correction, please write this. Also, consider writing "that shows" instead "and".
- AR: We have clarified the caption and written: "Bold scores are the best scores in a column".
- RC: L320: persistence forecast performs Could you for clarity once define what you use as a persistence forecast? It might be obvious to you, but for readers outside the field, it will be helpful.
- AR: We have been more specific about the persistence forecast and have defined it in the beginning of the results section by: "As baseline method, we use a persistence forecast with the initial conditions as constant prediction".

**RC: L335: Such localised features Consider adding the length scale in km if you think this finding is generalizable or holds valuable information for other processes related to sea-ice deformation.**

AR: In our case, we know that the missing processes lie between the 8 km and the 4 km, but if learned from observations, it might be indeed interesting to see the length-scale of the learned model error correction. We have tried to extract specific length-scales but had problems to properly define them. Instead, we have tried to be more specific.

**RC: L340: a generally smoother background pattern What is meant by this? Please clarify and rephrase.**

AR: We have clarified the discussion of the feature map.

**RC: Figure 6**

I) What colormap uses a)?

**II) I am wondering if also the concentration maps in the initial and forecast step would be helpful here to interpret the gradients?**

AR: The colormap of (a) is somewhat arbitrary as the normalized field like used for the input into the NN is shown. Instead, we will change to the unnormalized area and show a colorbar.

We have redesigned the Figure, as shown in Fig. 6, and added colormaps, and more useful information about the sea-ice area fields.

**RC: L378: either Isn't it to both instead of either or?**

- AR: Has been changed to "and".
- RC: L380: Table ?? correct reference
- AR: The reference has been corrected.
- RC: L382-383: Additionally, the sensitivity is directional dependent, Fig. 6g, and exhibits localised features, Fig. 6c and i Could you discuss these results also in the light of physical understanding that we can gain from the gradients? From both the gradients along initial and difference, we can learn about the shortcomings of the coarse resolution simulations that the NN tries to compensate for.
- AR: Thank you for this nice suggestion. Our goal was to prove the point of localised features and representation of anisotropy, neglecting its physical meaning. We have added a paragraph, discussing the physical meaning along the suggested lines.
- **RC:** L391:  $-1 \times 10^{-3}$  and  $1 \times 10^{-3}$ Units?
- AR: This sentence was a remnant of an older version of the manuscript. As there is no longer a restriction to the correction of the sea-ice thickness, the sentence has been removed. Thank you for spotting this typo.
- RC: L392: Related to optimal control theory in dynamical systems, This is not very helpful for readers with limited background knowledge. Please elaborate more or rephrase.
- AR: We have changed the sentence to "We change the performance metric to be the RMSE, a commonly used metric to evaluate forecast performances".
- RC: L405-408: Additionally, for the velocity, stress, and damage, the drift towards ... the "Initial + Forecast" experiment in these variables and averaged over all nine model variables. Unclear, please rephrase.

AR: We have simplified Sect. 5.3 and moved the results and discussion for the "Initial+Difference" experiment to Appendix D6. There, we have simplified this sentence.

RC: L412: As the initial condition error increases with each update, the network corrects less and less forecast errors. Could this effect be dampened by updating at higher frequencies?

- AR: As discussed for your comment # 3, there is a trade-off. It seems that the cons outweigh the pros, and more updates even hurt the performance. As we have added the Figure to Appendix D2, we have referenced to these results: "This effect has an larger impact on the forecast if the lead time between two corrections with the NN is further reduced (Appendix D2)".
- RC: L413-419:"To show the effect of this error distribution shift, ... An averaged correlation of 1 would indicate a perfect pattern correlation." This paragraph is hardly understandable with no background knowledge of the method "centred spatial pattern correlation". I suggest describing the principle of the method and its interpretation at the beginning of the paragraph in 1-2 sentences, before describing its specifics.
- AR: We have added a short description of the centred spatial pattern correlation: "we centre all fields by removing their mean, and estimate Pearson's correlation coefficient between the prediction and the residual in space".
- RC: L422: especially for the divergent stress From the previous paragraph, it sounds as if a value close to 1 is favorable, but this statement reads as if a high value close to 1 for divergent stress shows a weakness of the NN. Please clarify.
- AR: Considering the results for the sea-ice area, we have clarified that this shows a deficiency of the neural network but of the forecast model and written: "Caused by the drift towards the attractor, the sea-ice model forgets parts of the previous error correction for the velocity and divergent stress component, and these forgotten parts get corrected again in the fifth update".
- RC: L434-435: However, the parametrisation misses the development of new strains and positions the main strain at the wrong place. This suggests that the corrections of the NN violate the brittle model physics, as highly damaged areas are usually linked to high deformation rates. Is this correct? If so, please comment on this also in the text, and if there is a way to design a network that computes corrections in accordance with the physical laws of the model.
- AR: In the hybrid model, the field is damaged beyond a given deformation threshold, shown for the damaging process in the south, as in the high-resolution simulation. Only, the high deformation rates in already damaged areas are striking in the case

of the hybrid model. As such high rates are unobserved for the high resolution simulation, it remains unknown if this is an unphysical behaviour or not. Because the last correction of the neural network is already more than 9 minutes ago, we would speculate and say that the model would have had time to adapt to the new situation such that the rates are physical explainable.

- RC: L456-457: Therefore, using such a mapping into Cartesian space, we can apply CNNs, which can efficiently scale to larger, Arctic-wide, models. Are you talking about Arctic-wide models on unstructured grids? Or why is the mapping needed? Please clarify the text.
- AR: We have clarified that we mean Arctic-wide simulations on unstructured grids and written: "Therefore, using a mapping into Cartesian space, we can apply CNNs to Arctic-wide models with unstructured grids, like neXtSIM".
- RC: L462-463: As processes have no discretized resolution in realworld, we would have difficulties to find the right resolution for the projection in such cases.

Isn't that only an issue if you would aim to train a correction with observations? If it is a model, then you would always know the resolution of resolved scales. Please clarify

AR: Indeed, this is only an issue if we train with observations. We have disentangled this paragraph and removed the reference to learning from observations.

**RC: L462: truth Please clarify what is meant by truth: the high-resolution simulation or something different**

AR: We have replaced by "targetted simulations".

**RC: L464: this argument What argument?**

- AR: The argument that the optimal resolution is linked to the resolution of the processes in the targetted simulations. We have clarified that we mean this link.
- RC: L503: The only way is therefore to improve the forecast model, thereby changing its attractor.What about updating the forecasting model at higher frequencies? Please comment.
- AR: In the most extreme case, we would correct after each integration time step, which could be seen as integrated form of a subgrid-scale parameterisation. In this case, we would change the attractor, as each forecast is now a corrected forecast. The problem is rather to define the attractor of the hybrid model; one could define only the corrected states as attractor. We have added: "e.g., by directly parametrising

the subgrid-scale processes with a tendency correction." to make the link to the next paragraph more explicit.

- RC: L530-532: Mapping the input data into a Cartesian space that has a higher resolution than the original space, such scalable convolutional neural networks can be applied for feature extraction in sea-ice models defined on a triangular or unstructured grid. Something is wrong with this sentence, please correct it.
- AR: This sentence does make indeed no sense, we have rephrased it to: "For sea-ice models defined on a triangular or unstructured grid, such scalable convolutional neural networks can be applied for feature extraction by mapping the input data into a Cartesian space that has a higher resolution than the original space".
- RC: L543: total deformation On which Figure or result is the statement that the total deformation is improved? From Fig. 8, I would agree that the damage in the hybrid model looks closer to the high-resolution run than the uncorrected lowresolution simulation, but for total deformation, it is the other way around in my eyes.
- AR: We agree that the point-wise error for the hybrid forecast in the total deformation is higher. Hence, we have rephrases it into: "The deterministic model error correction leads to an improved representation of the fracturing processes".
- RC: L552: Appendix Table?
- AR: The Table number should be fixed.
- RC: L562-564: Caused by their limited capacity, the NNs have to focus on some variables, creating an imbalance between variables, which harms the performance for other variables, like the stress in the case of "Conv  $(\times 5)$ ".

Have you tried to train individual networks for each variable, which could balance this effect?

AR: In the initial phase of the research, we have seen that one big neural network with shared parameters performs better than nine small neural networks. Consequently, we have concentrated the research on a single big neural network. As one big network shares the features for all variables, it has to learn more general features than networks for single variables. The shared features act as regularisation which can help to reduce the overfitting. Hence, learning one network for all variables can enable the use of larger neural networks. We have run a test for the "Conv ( $\times$ 1)" architecture with multiple networks and added the results to Table C1. In Appendix C, we additionally added a discussion for these results.

**RC: L565: fast NN**

**Is the speed of the trained U-NeXt NN an issue compared to the computational costs of the geophysical model? Or does this refers to training speeds?**

AR: Compared to the geophysical model, the neural network is fast, if correctly implemented into the model, e.g., using the C++ libraries of PyTorch or TensorFlow. However, such kind of model error correction sits on top of a geophysical model, its runtime is additive to the runtime of the geophysical model. Consequently, the faster and lightweight the model, the better. So, it is in some sense a trade-off between additional runtime and additional gain, and, for some purposes, the Conv (×1) architecture might be enough. We have added "Such a fast NN can be helpful if the additional latency time impacts its application in a sea-ice model".

**3 General changes in the manuscript**

For the most of our result tables, we have found that we had mistakenly misordered the variable names (column heads). To match the order of the variable names, we have changed the order of the values. Consequently, we have recomputed all numbers. This has only a minor impact on the results and has not changed any of the conclusions.

To streamline the results part for the forecasting with the model error correction, we have removed the results of the "Initial+Difference" experiment from Fig. 7 and Table 6. Instead, we have introduced a new subsection in Appendix D6, where we compare the "Initial+Forecast" and "Initial+Difference" experiment.

During proofreading of the revised manuscript, we have found and fixed some orthographic mistakes. Additionally, we have improved the consistency in the wording.

Figure 1: The negative log-likelihood for a Laplace distribution, proportional to the mean absolute error (MAE), with a fixed weighting in the validation dataset as function of epochs for different fractions of training data, the brighter the colour, the less training data is used. The smaller Figure shows the averaged MAE in the test dataset as function of the fraction of training data. A fraction of 1.0 corresponds to around  $12.3 \times 10^6$  degrees-of-freedom in the training dataset.

Figure 2: Snapshot of sea-ice damage for a one-hour forecast with the here-used regional sea-ice model. Shown are the high-resolution simulations (a, 4 km resolution) and low-resolution forecasts (b, c). To initialise the low-resolution forecasts, the initial conditions of the high-resolution are projected into a low-resolution space with 8 km resolution. Started from these projected initial conditions, the low-resolution forecast (b) generates too much damage compared to the high-resolution field. Running the low-resolution model together with our learned model error correction (c) leads to a better representation of the damaging process, which improves the forecast by 62 % in this example.